# A PRIMAL-DUAL FRAMEWORK FOR TRANSFORMERS AND NEURAL NETWORKS

**Tan M. Nguyen\***
Department of Mathematics
University of California, Los Angeles
`tanmnguyen89@ucla.edu`

**Tam Nguyen\***
Department of ECE
Rice University
`nguyenminhtam9520@gmail.com`

**Nhat Ho**
Department of Statistics & Data Sciences
University of Texas at Austin
`minhnhat@utexas.edu`

**Andrea L. Bertozzi**
Department of Mathematics
University of California, Los Angeles
`bertozzi@math.ucla.edu`

**Richard G. Baraniuk\*\***
Department of ECE
Rice University
`richb@rice.edu`

**Stanley J. Osher\*\***
Department of Mathematics
University of California, Los Angeles
`sjo@math.ucla.edu`

## ABSTRACT

Self-attention is key to the remarkable success of transformers in sequence modeling tasks including many applications in natural language processing and computer vision. Like neural network layers, these attention mechanisms are often developed by heuristics and experience. To provide a principled framework for constructing attention layers in transformers, we show that the self-attention corresponds to the support vector expansion derived from a support vector regression problem, whose primal formulation has the form of a neural network layer. Using our framework, we derive popular attention layers used in practice and propose two new attentions: 1) the Batch Normalized Attention (Attention-BN) derived from the batch normalization layer and 2) the Attention with Scaled Head (Attention-SH) derived from using less training data to fit the SVR model. We empirically demonstrate the advantages of the Attention-BN and Attention-SH in reducing head redundancy, increasing the model's accuracy, and improving the model's efficiency in a variety of practical applications including image and time-series classification.

## 1 INTRODUCTION

Transformer models (Vaswani et al., 2017) have achieved impressive success with state-of-the-art performance in a myriad of sequence processing tasks, including those in computer vision (Dosovitskiy et al., 2021; Liu et al., 2021; Touvron et al., 2020; Ramesh et al., 2021; Radford et al., 2021; Arnab et al., 2021; Liu et al., 2022; Zhao et al., 2021; Guo et al., 2021), natural language processing (Devlin et al., 2018; Al-Rfou et al., 2019; Dai et al., 2019; Child et al., 2019; Raffel et al., 2020; Baevski & Auli, 2019; Brown et al., 2020; Dehghani et al., 2018), reinforcement learning (Chen et al., 2021; Janner et al., 2021), and other important applications (Rives et al., 2021; Jumper et al., 2021; Zhang et al., 2019; Gulati et al., 2020; Wang & Sun, 2022). Transformers can also effectively transfer knowledge from pre-trained models to new tasks with limited supervision (Radford et al., 2018; 2019; Devlin et al., 2018; Yang et al., 2019; Liu et al., 2019). The driving force behind the success of transformers is the self-attention mechanism (Cho et al., 2014; Parikh et al., 2016; Lin et al., 2017), which computes a weighted average of feature representations of the tokens in the sequence with the weights proportional to similarity scores between pairs of representations. The weights calculated by the self-attention determine the relative importance between tokens and thus capture the contextual representations of the sequence (Bahdanau et al., 2014; Vaswani et al., 2017; Kim et al., 2017). It has

---

\* Co-first authors. \*\* Co-last authors. Please correspond to: tanmnguyen89@ucla.edu

been argued that the flexibility in capturing diverse syntactic and semantic relationships is critical for the success of transformers (Tenney et al., 2019; Vig & Belinkov, 2019; Clark et al., 2019).

## 1.1 BACKGROUND: SELF-ATTENTION

For a given input sequence $\mathbf{X} := [\boldsymbol{x}_1, \cdots, \boldsymbol{x}_N]^\top \in \mathbb{R}^{N \times D_x}$ of $N$ feature vectors, self-attention transforms $\mathbf{X}$ into the output sequence $\mathbf{H}$ in the following two steps:

**Step 1.** The input sequence $\mathbf{X}$ is projected into the query matrix $\mathbf{Q}$, the key matrix $\mathbf{K}$, and the value matrix $\mathbf{V}$ via three linear transformations

$$\mathbf{Q} = \mathbf{X}\mathbf{W}_Q^\top; \mathbf{K} = \mathbf{X}\mathbf{W}_K^\top; \mathbf{V} = \mathbf{X}\mathbf{W}_V^\top,$$

where $\mathbf{W}_Q, \mathbf{W}_K \in \mathbb{R}^{D \times D_x}$, and $\mathbf{W}_V \in \mathbb{R}^{D_v \times D_x}$ are the weight matrices. We denote $\boldsymbol{Q} := [\boldsymbol{q}_1, \cdots, \boldsymbol{q}_N]^\top, \mathbf{K} := [\boldsymbol{k}_1, \cdots, \boldsymbol{k}_N]^\top$, and $\mathbf{V} := [\boldsymbol{v}_1, \cdots, \boldsymbol{v}_N]^\top$, where the vectors $\boldsymbol{q}_i, \boldsymbol{k}_i, \boldsymbol{v}_i$ for $i = 1, \cdots, N$ are the query, key, and value vectors, respectively.

**Step 2.** The output sequence $\mathbf{H} := [\boldsymbol{h}_1, \cdots, \boldsymbol{h}_N]^\top$ is then computed as follows

$$\mathbf{H} = \mathrm{softmax}\Big(\mathbf{Q}\mathbf{K}^\top/\sqrt{D}\Big)\mathbf{V} := \mathbf{A}\mathbf{V}, \tag{1}$$

where the softmax function is applied to each row of the matrix $\mathbf{Q}\mathbf{K}^\top/\sqrt{D}$. The matrix $\mathbf{A} := \mathrm{softmax}\Big(\frac{\mathbf{Q}\mathbf{K}^\top}{\sqrt{D}}\Big) \in \mathbb{R}^{N \times N}$ and its component $a_{ij}$ for $i, j = 1, \cdots, N$ are called the attention matrix and attention scores, respectively. For each query vector $\boldsymbol{q}_i$ for $i = 1, \cdots, N$, an equivalent form of Eqn. (1) to compute the output vector $\boldsymbol{h}_i$ is given by

$$\boldsymbol{h}_i = \sum_{j=1}^{N} \mathrm{softmax}\Big(\boldsymbol{q}_i^\top \boldsymbol{k}_j/\sqrt{D}\Big)\boldsymbol{v}_j. \tag{2}$$

The self-attention computed by Eqn. (1) and (2) is called the scaled dot-product or softmax attention. In our paper, we call a transformer that uses this attention the softmax transformer. The structure that the attention matrix $\mathbf{A}$ learns from training determines the ability of the self-attention to capture contextual representation for each token. Additionally, a residual connection can be added to the output of the self-attention layer, $\boldsymbol{h}_i = \boldsymbol{x}_i + \sum_{j=1}^{N} \mathrm{softmax}\Big(\boldsymbol{q}_i^\top \boldsymbol{k}_j/\sqrt{D}\Big)\boldsymbol{v}_j$.

**Multi-head Attention (MHA).** In MHA, multiple heads are concatenated to compute the final output. This MHA mechanism allows transformers to capture more diverse attention patterns and increase the capacity of the model. Let $H$ be the number of heads and $\mathbf{W}_O^{\mathrm{multi}} = \big[\mathbf{W}_O^1, \ldots, \mathbf{W}_O^H\big] \in \mathbb{R}^{D_v \times HD_v}$ be the projection matrix for the output where $\mathbf{W}_O^1, \ldots, \mathbf{W}_O^H \in \mathbb{R}^{D_v \times D_v}$. The MHA is defined as

$$\mathrm{MultiHead}(\{\mathbf{H}\}_{s=1}^H) = \mathrm{Concat}(\mathbf{H}^1, \ldots, \mathbf{H}^H)\mathbf{W}_O^{\mathrm{multi}\top} = \sum_{s=1}^{H} \mathbf{H}^s \mathbf{W}_O^{s\top} = \sum_{s=1}^{H} \mathbf{A}^s \mathbf{V}^s \mathbf{W}_O^{s\top}. \tag{3}$$

Despite their remarkable success, most attention layers are developed based on heuristic approaches, and a coherent principled framework for synthesizing attention layers has remained elusive.

## 1.2 CONTRIBUTION

We derive the self-attention as the support vector expansion of a given support vector regression (SVR) problem. The primal representation of the regression function has the form of a neural network layer. Thus, we establish a primal-dual connection between an attention layer in transformers and a neural network layer in deep neural networks. Our framework suggests a principled approach to developing an attention mechanism: *Starting from a neural network layer and a support vector regression problem, we derive the dual as a support vector expansion to attain the corresponding attention layer.* We then employ this principled approach to invent two novel classes of attentions: the Batch Normalized Attention (Attention-BN) derived from the batch normalization layer in deep neural networks and the Attention with Scaled Heads (Attention-SH) resulting from solving the support vector regression model with less amount of training data. Our contribution is three-fold.

1. We derive self-attention as a support vector expansion that solves a SVR problem, thus providing a principled primal-dual framework to study and develop self-attentions.

2. We re-derive popular attentions, such as the linear attention (Katharopoulos et al., 2020), the sparse attention (Child et al., 2019), and the multi-head attention (Vaswani et al., 2017), from our proposed framework.

3. We develop two new attention mechanism: the Batch Normalized Attention (Attention-BN) and the Attention with Scaled Heads (Attention-SH) using our proposed framework.

We empirically demonstrate that 1) the Attention-BN significantly outperforms the baseline softmax and linear attention and 2) the Attention-SH performs better while being more efficient than the same baselines on a variety of practical tasks including image and time-series classification.

## 2 PRIMAL-DUAL INTERPRETATION OF SELF-ATTENTION

We first provide a primal-dual interpretation of self-attention as a support vector regression problem in Section 2.1. Based on that primal-dual framework, we derive popular attention mechanisms as the support vector expansion in Section 2.2. Finally, we introduce two new attention mechanisms in Section 2.3, the Attention-BN and Attention-SH.

### 2.1 ATTENTION AS A SUPPORT VECTOR REGRESSION MODEL

In this section, we derive self-attention from a support vector regression problem. Suppose we are given a training data $\{(\boldsymbol{k}_1, \boldsymbol{y}_1), \ldots, (\boldsymbol{k}_N, \boldsymbol{y}_N)\} \subset \mathcal{K} \times \mathcal{Y}$, where $\mathcal{K} = \mathbb{R}^D$ and $\mathcal{Y} = \mathbb{R}^{D_v}$. Here, $\boldsymbol{k}_1, \ldots, \boldsymbol{k}_N$ are attention keys in self-attention, and $\boldsymbol{y}_1, \ldots, \boldsymbol{y}_N$ are the training targets. We consider the function $f$, taking the form

$$\boldsymbol{y} = f(\boldsymbol{x}) := \mathbf{W}\frac{\Phi(\boldsymbol{x})}{h(\boldsymbol{x})} + \boldsymbol{b}, \tag{4}$$

where $\boldsymbol{x} \in \mathcal{K} = \mathbb{R}^D$, $\Phi(\boldsymbol{x}) = [\phi_1(\boldsymbol{x}), \ldots, \phi_{D_\phi}(\boldsymbol{x})] \in \mathbb{R}^{D_\phi}$, $\mathbf{W} = [\boldsymbol{w}_1, \ldots, \boldsymbol{w}_{D_v}]^\top \in \mathbb{R}^{D_v \times D_\phi}$, $\boldsymbol{b} \in \mathbb{R}^{D_v}$, and $h(\boldsymbol{x})$ is a vector-scalar function. We fit the function $f$ to the training data $\{(\boldsymbol{k}_1, \boldsymbol{y}_1), \ldots, (\boldsymbol{k}_N, \boldsymbol{y}_N)\}$ with an $\mathbb{L}_2$ regularization on $\mathbf{W}$, i.e., a ridge regression, by solving the following convex optimization problem:

$$\underset{\substack{\mathbf{W} \\ \boldsymbol{\xi}_j, \tilde{\boldsymbol{\xi}}_j, j=1,\ldots,N}}{\text{minimize}} \quad \frac{1}{2}\|\mathbf{W}\|_{\mathrm{F}}^2 + C\sum_{j=1}^N\sum_{d=1}^{D_v}\left(\boldsymbol{\xi}_j(d) + \tilde{\boldsymbol{\xi}}_j(d)\right) = \frac{1}{2}\sum_{d=1}^{D_v}\|\boldsymbol{w}_d\|^2 + C\sum_{j=1}^N\sum_{d=1}^{D_v}\left(\boldsymbol{\xi}_j(d) + \tilde{\boldsymbol{\xi}}_j(d)\right)$$

$$\text{subject to} \quad \begin{cases} \boldsymbol{y}_j(d) - \boldsymbol{w}_d^\top\Phi(\boldsymbol{k}_j)/h(\boldsymbol{k}_j) - \boldsymbol{b}(d) \leq \epsilon + \boldsymbol{\xi}_j(d) \\ \boldsymbol{w}_d^\top\Phi(\boldsymbol{k}_j)/h(\boldsymbol{k}_j) + \boldsymbol{b}(d) - \boldsymbol{y}_j(d) \leq \epsilon + \tilde{\boldsymbol{\xi}}_j(d) \quad, \ j = 1, \ldots, N, \ d = 1, \ldots, D_v. \\ \boldsymbol{\xi}_j(d), \tilde{\boldsymbol{\xi}}_j(d) \geq 0 \end{cases}$$

$$\tag{5}$$

The Eqn. 5 implies that there exists a function $f$ that can approximates all pairs $(\boldsymbol{k}_j, \boldsymbol{y}_j)$ with $\epsilon$ precision. The additional slack variables $\boldsymbol{\xi}_j, \tilde{\boldsymbol{\xi}}_j$ relax this assumption and allows some of the training set data points to have the training error greater than $\epsilon$ just as in the soft-margin SVM (Cortes & Vapnik, 1995; Schölkopf et al., 2002). $C > 0$ is a constant determining the trade between the complexity penalizer $\sum_{d=1}^{D_v}\|\boldsymbol{w}_d\|^2$, i.e., the flatness of $f$, and the amount up to which deviations larger than $\epsilon$ are tolerated.

In order to derive the self-attention from the support vector regression defined by the optimization problem 5, the key idea to construct the Lagrangian from Eqn. 5 and find the representation of the $\boldsymbol{w}_d$, $d = 1, \ldots, D_v$, in terms of the dual variables. We define the Lagrangian function as follows:

$$\mathcal{L} := \frac{1}{2}\sum_{d=1}^{D_v}\|\boldsymbol{w}_d\|^2 + C\sum_{j=1}^N\sum_{d=1}^{D_v}\left(\boldsymbol{\xi}_j(d) + \tilde{\boldsymbol{\xi}}_j(d)\right) - \sum_{j=1}^N\sum_{d=1}^{D_v}\left(\boldsymbol{\eta}_j(d)\boldsymbol{\xi}_j(d) + \tilde{\boldsymbol{\eta}}_j(d)\tilde{\boldsymbol{\xi}}_j(d)\right)$$

$$- \sum_{j=1}^N\sum_{d=1}^{D_v}\boldsymbol{\alpha}_j(d)\left(\epsilon + \boldsymbol{\xi}_j(d) - \boldsymbol{y}_j(d) + \boldsymbol{w}_d^\top\frac{\Phi(\boldsymbol{k}_j)}{h(\boldsymbol{k}_j)} + \boldsymbol{b}(d)\right) \tag{6}$$

$$- \sum_{j=1}^N\sum_{d=1}^{D_v}\tilde{\boldsymbol{\alpha}}_j(d)\left(\epsilon + \tilde{\boldsymbol{\xi}}_j(d) + \boldsymbol{y}_j(d) - \boldsymbol{w}_d^\top\frac{\Phi(\boldsymbol{k}_j)}{h(\boldsymbol{k}_j)} - \boldsymbol{b}(d)\right),$$

where $\boldsymbol{\eta}_j, \tilde{\boldsymbol{\eta}}_j, \boldsymbol{\alpha}_j$ and $\tilde{\boldsymbol{\alpha}}_j$ are Lagrange multipliers. These dual variables have to satisfy positivity constraints, i.e., $\boldsymbol{\eta}_j(d), \tilde{\boldsymbol{\eta}}_j(d), \boldsymbol{\alpha}_j(d), \tilde{\boldsymbol{\alpha}}_j(d) \geq 0, \forall j = 1, \ldots, N, \forall d = 1, \ldots, D_v$. It follows from the saddle point condition that the partial derivatives of the Lagrangian function $\mathcal{L}$ with respect to the primal variables $\left(\boldsymbol{w}_d, \boldsymbol{b}(d), \{\boldsymbol{\xi}_j(d), \tilde{\boldsymbol{\xi}}_j(d)\}_{j=1}^N\right)$, $d = 1, \ldots, D_v$, have to vanish for optimality,

namely, we have:

$$\partial_{\boldsymbol{b}(d)}\mathcal{L} = \sum_{j=1}^{N}(\tilde{\boldsymbol{\alpha}}_j(d) - \boldsymbol{\alpha}_j(d)) = 0 \Rightarrow \sum_{j=1}^{N}(\boldsymbol{\alpha}_j(d) - \tilde{\boldsymbol{\alpha}}_j(d)) = 0, \tag{7}$$

$$\partial_{\boldsymbol{w}_d}\mathcal{L} = \boldsymbol{w}_d - \sum_{j=1}^{N}(\boldsymbol{\alpha}_j(d) - \tilde{\boldsymbol{\alpha}}_j(d))\frac{\Phi(\boldsymbol{k}_j)}{h(\boldsymbol{k}_j)} = 0 \Rightarrow \boldsymbol{w}_d = \sum_{j=1}^{N}(\boldsymbol{\alpha}_j(d) - \tilde{\boldsymbol{\alpha}}_j(d))\frac{\Phi(\boldsymbol{k}_j)}{h(\boldsymbol{k}_j)}, \tag{8}$$

$$\partial_{\boldsymbol{\xi}_j(d)}\mathcal{L} = C - \boldsymbol{\alpha}_j(d) - \boldsymbol{\eta}_j(d) = 0, \ \ \partial_{\tilde{\boldsymbol{\xi}}_j(d)}\mathcal{L} = C - \tilde{\boldsymbol{\alpha}}_j(d) - \tilde{\boldsymbol{\eta}}_j(d) = 0. \tag{9}$$

Let $\boldsymbol{v}_j = [\frac{\boldsymbol{\alpha}_j(1)-\tilde{\boldsymbol{\alpha}}_j(1)}{h(\boldsymbol{k}_j)}, \dots, \frac{\boldsymbol{\alpha}_j(D_v)-\tilde{\boldsymbol{\alpha}}_j(D_v)}{h(\boldsymbol{k}_j)}]^\top$, $j = 1, \dots, N$, and substitute Eqn. 8 into Eqn. 4, we obtain the following support vector expansion of the linear basis function $f$:

$$f(\boldsymbol{x}) = \left[\sum_{j=1}^{N}\frac{\boldsymbol{\alpha}_j(1) - \tilde{\boldsymbol{\alpha}}_j(1)}{h(\boldsymbol{k}_j)}\frac{\Phi(\boldsymbol{x})^\top\Phi(\boldsymbol{k}_j)}{h(\boldsymbol{x})}, \dots, \sum_{j=1}^{N}\frac{\boldsymbol{\alpha}_j(D_v) - \tilde{\boldsymbol{\alpha}}_j(D_v)}{h(\boldsymbol{k}_j)}\frac{\Phi(\boldsymbol{x})^\top\Phi(\boldsymbol{k}_j)}{h(\boldsymbol{x})}\right]^\top + \boldsymbol{b},$$

$$= \sum_{j=1}^{N}\frac{\Phi(\boldsymbol{x})^\top\Phi(\boldsymbol{k}_j)}{h(\boldsymbol{x})}\boldsymbol{v}_j + \boldsymbol{b}. \tag{10}$$

**Remark 1** *Notice that from Eqn. 9 and the conditions $\boldsymbol{\eta}_j(d), \tilde{\boldsymbol{\eta}}_j(d), \boldsymbol{\alpha}_j(d), \tilde{\boldsymbol{\alpha}}_j(d) \geq 0$, we can prove that $\boldsymbol{\alpha}_j(d), \tilde{\boldsymbol{\alpha}}_j(d) \in [0, C]$. Furthermore, we can show that $\boldsymbol{\alpha}_j(d) * \tilde{\boldsymbol{\alpha}}_j(d) = 0$ (Smola & Schölkopf, 2004; Schölkopf et al., 2002). As a result, $\boldsymbol{v}_j(d) \in \left[-\frac{C}{h(\boldsymbol{k}_j)}, \frac{C}{h(\boldsymbol{k}_j)}\right]$, $d = 1, \dots, D_v$.*

**Deriving Softmax Attention.** Choosing the appropriate $h(\boldsymbol{x})$ and $\Phi(\boldsymbol{x})$ allows us to derive the popular softmax attention given in Eqn. 1 and 2. In particular, if we choose $h(\boldsymbol{x}) := \sum_{j}^{N}\Phi(\boldsymbol{x})^T\Phi(\boldsymbol{k}_j)$, Eqn. 10 becomes

$$f(\boldsymbol{x}) = \sum_{j=1}^{N}\frac{\Phi(\boldsymbol{x})^\top\Phi(\boldsymbol{k}_j)}{\sum_{j'}^{N}\Phi(\boldsymbol{x})^T\Phi(\boldsymbol{k}_{j'})}\boldsymbol{v}_j + \boldsymbol{b} = \frac{\sum_{j=1}^{N}\Phi(\boldsymbol{x})^\top\Phi(\boldsymbol{k}_j)\boldsymbol{v}_j}{\sum_{j'}^{N}\Phi(\boldsymbol{x})^T\Phi(\boldsymbol{k}_{j'})} + \boldsymbol{b}. \tag{11}$$

We then select $\Phi(\boldsymbol{x}) = \left(a_{l_0}^{(0)}, a_1^{(1)}, \dots, a_{l_1}^{(1)}, \dots, a_1^{(t)}, \dots, a_{l_t}^{(t)}, \dots\right)$ where $l_t = \binom{D+t-1}{t}$ and

$$a_l^{(t)} = \frac{(x_1/\sqrt[4]{D})^{n_1}\dots(x_D/\sqrt[4]{D})^{n_D}}{\sqrt{n_1!\dots n_D!}} \mid n_1 + \dots + n_D = t, \ 1 \leq l \leq l_t. \tag{12}$$

Since

$$\exp(\boldsymbol{x}^T\boldsymbol{y}) = \sum_{t=0}^{\infty}\frac{(\boldsymbol{x}^T\boldsymbol{y})^t}{t!} = \sum_{t=0}^{\infty}\sum_{n_1+\dots+n_D=t}\left(\frac{x_1^{n_1}\dots x_D^{n_D}}{\sqrt{n_1!\dots n_D!}}\right)\left(\frac{y_1^{n_1}\dots y_D^{n_D}}{\sqrt{n_1!\dots n_D!}}\right), \tag{13}$$

then Eqn. 27 becomes

$$f(\boldsymbol{x}) = \sum_{j=1}^{N}\frac{\exp(\boldsymbol{x}^\top\boldsymbol{k}_j/\sqrt{D})}{\sum_{j'=1}^{N}\exp(\boldsymbol{x}^\top\boldsymbol{k}_{j'}/\sqrt{D})}\boldsymbol{v}_j + \boldsymbol{b} = \sum_{j=1}^{N}\text{softmax}\left(\boldsymbol{x}^\top\boldsymbol{k}_j/\sqrt{D}\right)\boldsymbol{v}_j + \boldsymbol{b}. \tag{14}$$

Let $\boldsymbol{x} = \boldsymbol{q}_i$, $\boldsymbol{b} = 0$ and relax the boundness constraint of $\boldsymbol{v}_j$ in Remark 1. Eqn. 30 becomes Eqn. 2 of the softmax attention (Vaswani et al., 2017). We summarize our results in the following theorem.

**Theorem 1 (Softmax Attention as a Support Vector Expansion)** *Given the function $f$ defined in Eqn. 4 with $h(\boldsymbol{x}) := \sum_{j}^{N}\Phi(\boldsymbol{x})^T\Phi(\boldsymbol{k}_j)$ and the support vector regression problem defined in Eqn. 5, we set $\boldsymbol{b} = 0$, choose $\Phi(\boldsymbol{x})$ as in Eqn. 28, and relax the boundness constraint of the variables $\boldsymbol{v}_j = [\frac{\boldsymbol{\alpha}_j(1)-\tilde{\boldsymbol{\alpha}}_j(1)}{h(\boldsymbol{k}_j)}, \dots, \frac{\boldsymbol{\alpha}_j(D_v)-\tilde{\boldsymbol{\alpha}}_j(D_v)}{h(\boldsymbol{k}_j)}]^\top$, where $\boldsymbol{\alpha}_j$ and $\tilde{\boldsymbol{\alpha}}_j$ are dual variables of Eqn. 5, $j = 1, \dots, N$. Then, the support vector expansion of $f$ derived from Eqn. 5 has the form of a softmax attention*

$$f(\boldsymbol{x}) = \sum_{j=1}^{N}\text{softmax}\left(\boldsymbol{x}^\top\boldsymbol{k}_j/\sqrt{D}\right)\boldsymbol{v}_j. \tag{15}$$

**Remark 2** *Since $\boldsymbol{b}$ is set to 0, the centering constraint of $\boldsymbol{\alpha}_j$ and $\tilde{\boldsymbol{\alpha}}_j$ in Eqn. 7 can be ignored.*

**Remark 3** *Theorem 1 and its derivation can be easily extended to capture the full form of the softmax attention with the residual connection, the query matrix projection $\mathbf{W}_Q$, the key matrix projection $\mathbf{W}_K$, and the value matrix projection $\mathbf{W}_V$. We include this result in Appendix F.*

**Remark 4** *The primal representation of the function $f$ as in Eqn. 4 has the form of a neural network layer where $\mathbf{W}$ is the weight, $\boldsymbol{b}$ is the bias term, $\Phi(\boldsymbol{x})$ is the input, and $h(\boldsymbol{x})$ is the normalization term. Thus, an attention layer and a neural network layer are primal-dual of each other.*

**A principled approach to developing an attention mechanism.** The observation in Remark 4 suggests a principled way to construct an attention layer: *Starting from a neural network layer and a support vector regression problem, we derive the dual as a support vector expansion to attain the corresponding attention layer.* Using this approach, we derive popular attention mechanisms in Section 2.2 and propose our new attention mechanisms in Section 2.3.

## 2.2   DERIVING POPULAR ATTENTION MECHANISMS AS THE SUPPORT VECTOR EXPANSION

In this section, we derive popular attentions such as the linear attention (Katharopoulos et al., 2020), the sparse attention (Child et al., 2019), and the multi-head attention (Vaswani et al., 2017).

### 2.2.1   LINEAR ATTENTION

The Eqn. 27, which is obtained when choosing $h(\boldsymbol{x}) := \sum_j^N \Phi(\boldsymbol{x})^T \Phi(\boldsymbol{k}_j)$, already matches the formula of the linear attention. Here, we can let $\boldsymbol{b} = 0$ as above and select the function $\Phi$ that results in a positive similarity function, e.g. $\Phi(\boldsymbol{x}) = \text{elu}(\boldsymbol{x}) + 1$, as in (Katharopoulos et al., 2020).

### 2.2.2   SPARSE ATTENTION

The sparse attention (Child et al., 2019) can be derived by fitting the function $f$ in Eqn. 4 using a different subset $\{(\boldsymbol{k}_{m_x(1)}, \boldsymbol{y}_{m_x(1)}), \ldots, (\boldsymbol{k}_{m_x(M)}, \boldsymbol{y}_{m_x(M)})\}$ of training data $\{(\boldsymbol{k}_1, \boldsymbol{y}_1), \ldots, (\boldsymbol{k}_N, \boldsymbol{y}_N)\}$ for each input data $\boldsymbol{x}$, where $\mathcal{M}_{\boldsymbol{x}} = \{m_{\boldsymbol{x}}(1), \ldots, m_{\boldsymbol{x}}(M)\} \subset \{1, \ldots, N\}$. The support vector expansion of $f$ is then given by

$$f(\boldsymbol{x}) = \sum_{j=1}^{N} \mathbf{1}_{\mathcal{M}_{\boldsymbol{x}}}(j) \frac{\Phi(\boldsymbol{x})^{\top} \Phi(\boldsymbol{k}_j)}{h(\boldsymbol{x})} \boldsymbol{v}_j + \boldsymbol{b} \tag{16}$$

where $\mathbf{1}_{\mathcal{M}_{\boldsymbol{x}}}(j) = [j \in \mathcal{M}_{\boldsymbol{x}}] := \begin{cases} 1 \text{ if } j \in \mathcal{M}_{\boldsymbol{x}} \\ 0 \text{ otherwise} \end{cases}$ . Note that the subsets $\mathcal{M}_{\boldsymbol{x}}$ are different for different $\boldsymbol{x}$. When letting $\boldsymbol{x} = \boldsymbol{q}_i$ where $\boldsymbol{q}_i, i = 1, \ldots, N$, are the query vectors and choosing $\Phi, h, \boldsymbol{b}$ as in Section 2.1, we can obtain the sparse attention in (Child et al., 2019) where the binary matrix $\mathbf{M} = \left(\mathbf{1}_{\mathcal{M}_{\boldsymbol{q}_i}}(j)\right)_{i,j=1}^{N}$ becomes the sparse masking matrix.

### 2.2.3   MULTI-HEAD ATTENTION (MHA)

The MHA can be derived by solving multiple support vector regression problems and then linearly combining their outputs. In particular, given $H$ training datasets $\{(\boldsymbol{k}_1^1, \boldsymbol{y}_1^1), \ldots, (\boldsymbol{k}_N^1, \boldsymbol{y}_N^1)\}, \ldots, \{(\boldsymbol{k}_1^H, \boldsymbol{y}_1^H), \ldots, (\boldsymbol{k}_N^H, \boldsymbol{y}_N^H)\} \subset \mathcal{K} \times \mathcal{Y}$, where $\mathcal{K} = \mathbb{R}^D$ and $\mathcal{Y} = \mathbb{R}^{D_v}$. We define the function $f$ applied on the input vector $\boldsymbol{x} = [\boldsymbol{x}^1, \ldots, \boldsymbol{x}^H]$ as follows

$$\boldsymbol{y} = f(\boldsymbol{x}) := \sum_{s=1}^{H} \mathbf{W}_O^s \boldsymbol{y}^s = \sum_{s=1}^{H} \mathbf{W}_O^s f^s(\boldsymbol{x}^s) = \sum_{s=1}^{H} \mathbf{W}_O^s \left( \mathbf{W}^s \frac{\Phi^s(\boldsymbol{x}^s)}{h^s(\boldsymbol{x}^s)} + \boldsymbol{b}^s \right), \tag{17}$$

where each function $f^s(\boldsymbol{x}^s) = \mathbf{W}^s \frac{\Phi^s(\boldsymbol{x}^s)}{h^s(\boldsymbol{x}^s)} + \boldsymbol{b}^s$ is fitted to the training dataset $\{(\boldsymbol{k}_1^s, \boldsymbol{y}_1^s), \ldots, (\boldsymbol{k}_N^s, \boldsymbol{y}_N^s)\}$. Following the same derivation and choosing $\{\Phi^s, h^s, \boldsymbol{b}^s\}_{s=1}^{H}$ as in Section 2.1, we can rewrite $f(\boldsymbol{x})$ in terms of the support vector expansions of the individual functions $f^s(\boldsymbol{x}^s)$, which are the individual softmax attentions

$$f(\boldsymbol{x}) = \sum_{s=1}^{H} \mathbf{W}_O^s \left( \sum_{j=1}^{N} \frac{\Phi^s(\boldsymbol{x}^s)^{\top} \Phi^s(\boldsymbol{k}_j^s)}{h^s(\boldsymbol{x}^s)} \boldsymbol{v}_j^s + \boldsymbol{b}^s \right) = \sum_{s=1}^{H} \mathbf{W}_O^s \left( \sum_{j=1}^{N} \text{softmax}\left( \boldsymbol{x}^{s\top} \boldsymbol{k}_j^s / \sqrt{D} \right) \boldsymbol{v}_j^s \right).$$
$$\tag{18}$$

Comparing Eqn. 18 and Eqn. 3, we see that Eqn. 18 computes the MHA when choosing $\boldsymbol{x}^s = \boldsymbol{q}_i^s$ where $\boldsymbol{q}_i^s, i = 1, \ldots, N$, are the query vectors at the $s^{\text{th}}$ head.

## 2.3   DERIVING NEW ATTENTION MECHANISMS: BATCH NORMALIZED ATTENTION AND MULTIRESOLUTION HEAD ATTENTION

In this section, we employ our primal-dual framework to develop new attention mechanisms. In particular, we derive: 1) the Batch Normalized Attention from employing the batch normalization (Ioffe & Szegedy, 2015); and 2) the Attention with Scaled Heads from using different amounts of training data. By 1) and 2), we demonstrate that *new attentions can be invented by modifying the primal neural network layer and the support vector regression problem in our framework, respectively.*

### 2.3.1 BATCH NORMALIZED ATTENTION

We incorporate the batch normalization into the primal form of the function $f$ in Eqn. 4. Given a training data $\{(\mathbf{k}_1, \mathbf{y}_1), \ldots, (\mathbf{k}_N, \mathbf{y}_N)\} \subset \mathcal{K} \times \mathcal{Y}$, where $\mathcal{K} = \mathbb{R}^D$ and $\mathcal{Y} = \mathbb{R}^{D_v}$ as in Section 2.1, the resultant $f$ is defined as follows

$$f(\mathbf{x}) := \mathbf{W} \frac{\Phi((\mathbf{x} - \boldsymbol{\mu}) \odot \mathbf{s}^{-1})}{h((\mathbf{x} - \boldsymbol{\mu}) \odot \mathbf{s}^{-1})} + \mathbf{b}, \tag{19}$$

where

$$\boldsymbol{\mu} = \frac{1}{N} \sum_{j=1}^{N} \mathbf{k}_j, \ \mathbf{s}^{-1} = \left[ \frac{1}{\sqrt{\sigma_1^2 + \epsilon}}, \ldots, \frac{1}{\sqrt{\sigma_D^2 + \epsilon}} \right]^{\top}, \ \sigma_d^2 = \frac{1}{N} \sum_{j=1}^{N} (\mathbf{k}_j(d) - \boldsymbol{\mu}(d))^2. \tag{20}$$

Here, $d = 1, \ldots, D$, and the mean subtraction and division by the standard deviation is performed element-wise along the feature dimension of $\mathbf{x}$. Following the same derivation as in Section 2.1, we derive the following support vector expansion of $f$

$$f(\mathbf{x}) = \sum_{j=1}^{N} \frac{\Phi((\mathbf{x} - \boldsymbol{\mu}) \odot \mathbf{s}^{-1})^{\top} \Phi((\mathbf{k}_j - \boldsymbol{\mu}) \odot \mathbf{s}^{-1})}{h((\mathbf{x} - \boldsymbol{\mu}) \odot \mathbf{s}^{-1})} \mathbf{v}_j + \mathbf{b}. \tag{21}$$

Here, $\mathbf{v}_j = \left[ \frac{\boldsymbol{\alpha}_j(1) - \tilde{\boldsymbol{\alpha}}_j(1)}{h((\mathbf{k}_j - \boldsymbol{\mu}) \odot \mathbf{s}^{-1})}, \ldots, \frac{\boldsymbol{\alpha}_j(D_v) - \tilde{\boldsymbol{\alpha}}_j(D_v)}{h((\mathbf{k}_j - \boldsymbol{\mu}) \odot \mathbf{s}^{-1})} \right]^{\top}$, where $\boldsymbol{\alpha}_j$ and $\tilde{\boldsymbol{\alpha}}_j$ are the dual variables, $j = 1, \ldots, N$. Same as in Section 2.1, in Eqn. 21, we choose $\Phi$ as in Eqn. 28, $h(\mathbf{x}) := \sum_j^N \Phi(\mathbf{x})^T \Phi(\mathbf{k}_j)$, and $\mathbf{b} = 0$ to obtain the the Batch Normalized Attention, which is defined as follows.

**Definition 1 (Batch Normalized Attention)** *Given a set of the key and value vectors $\{\mathbf{k}_j, \mathbf{v}_j\}_{j=1}^N$, for each query vector $\mathbf{q}_i$, $i = 1, \ldots, N$, the Batch Normalized Attention (Attention-BN) computes the corresponding output vector $\mathbf{h}_i$ of the query $\mathbf{q}_i$ by the following attention formula:*

$$\mathbf{h}_i = \sum_{j=1}^{N} softmax \left( ((\mathbf{q}_i - \boldsymbol{\mu}) \odot \mathbf{s}^{-1})^{\top} ((\mathbf{k}_j - \boldsymbol{\mu}) \odot \mathbf{s}^{-1}) / \sqrt{D} \right) \mathbf{v}_j, \tag{22}$$

*where*

$$\boldsymbol{\mu} = \frac{1}{N} \sum_{j=1}^{N} \mathbf{k}_j, \ \mathbf{s}^{-1} = \left[ \frac{1}{\sqrt{\sigma_1^2 + \epsilon}}, \ldots, \frac{1}{\sqrt{\sigma_D^2 + \epsilon}} \right]^{\top}, \ \sigma_d^2 = \frac{1}{N} \sum_{j=1}^{N} (\mathbf{k}_j(d) - \boldsymbol{\mu}(d))^2. \tag{23}$$

**The Effect of Normalization.** Expanding the dot product in the Attention-BN (see Appendix E), Eqn. 22 becomes

$$\mathbf{h}_i = \sum_{j=1}^{N} softmax \left( \sum_{d=1}^{D} \frac{\mathbf{q}_i(d) \mathbf{k}_j(d) - \frac{1}{N} \sum_{j'=1}^{N} \mathbf{k}_{j'}(d) \mathbf{k}_j(d)}{\sqrt{D}(\sigma_d^2 + \epsilon)} \right) \mathbf{v}_j. \tag{24}$$

Eqn. 24 implies that in the Attention-BN, the similarity between the query $\mathbf{q}_i$ and the key $\mathbf{k}_j$ is adjusted by the similarity between the key $\mathbf{k}_j$ and all the keys $\mathbf{k}_{j'}$, $j' = 1, \ldots, N$. In particular, if the key $\mathbf{k}_j$ is too similar to other keys, the query $\mathbf{q}_i$ will attend to it less and vice versa.

### 2.3.2 ATTENTION WITH SCALED HEADS

The *Attention with Scaled Heads*, named Attention-SH, is derived based on the derivation of the MHA in Section 2.2.3. The key idea underlying the Attention-SH is to train multiple support vector regression problems using different amounts of training data. In particular, the Attention-SH follows Eqn. 17 in Section 2.2.3 and defines the same regression function $f$ as the MHA. However, the Attention-SH fits the function $f^s$, $s = 1, \ldots, H$, in Eqn. 17 with training sets $\{(\mathbf{k}_1^1, \mathbf{y}_1^1), \ldots, (\mathbf{k}_{N_1}^1, \mathbf{y}_{N_1}^1)\}, \ldots, \{(\mathbf{k}_1^H, \mathbf{y}_1^H), \ldots, (\mathbf{k}_{N_H}^H, \mathbf{y}_{N_H}^H)\} \subset \mathcal{K} \times \mathcal{Y}$ of different sizes $N_1, \ldots, N_H$, where $\mathcal{K} = \mathbb{R}^D$ and $\mathcal{Y} = \mathbb{R}^{D_v}$. The resultant support vector expansion yields the formula of the Attention-SH as in the following definition.

**Definition 2 (Attention with Scaled Heads)** *Given $H$ sets of the key and value vectors $\{\mathbf{k}_j^1, \mathbf{v}_j^1\}_{j=1}^{N_1}, \ldots, \{\mathbf{k}_j^H, \mathbf{v}_j^H\}_{j=1}^{N_H}$, for each set of $H$ query vectors $\mathbf{q}_i^1, \ldots, \mathbf{q}_i^H$, $i = 1, \ldots, N$, the Attention with Scaled Heads (Attention-SH) computes the corresponding output vector $\mathbf{h}_i$ of the queries $\mathbf{q}_i^1, \ldots, \mathbf{q}_i^H$ by the following attention formula:*

$$\mathbf{h}_i = \sum_{s=1}^{H} \mathbf{W}_O^s \left( \sum_{j=1}^{N_s} softmax \left( \mathbf{q}_i^{s\top} \mathbf{k}_j^s / \sqrt{D} \right) \mathbf{v}_j^s \right). \tag{25}$$

Table 1: Test Accuracy (%) of the Attention-BN/SH/BN+SH vs. the baseline softmax attention on a subset of the UEA Time Series Classification Archive benchmark (Bagnall et al., 2018). Our proposed attentions significantly outperform the baseline. We also include the reported results from (Zerveas et al., 2021) and (Wu et al., 2022) (in parentheses) in addition to our reproduced results.

| Dataset/Model | *Baseline Softmax* | Attention-BN | Attention-SH | Attention-BN+SH |
|---|---|---|---|---|
| ETHANOLCONCENTRATION | $32.08 \pm 1.24$ (33.70) | $33.33 \pm 0.44$ | $33.59 \pm 0.58$ | $\mathbf{34.35 \pm 0.53}$ |
| FACEDETECTION | $68.70 \pm 0.61$ (68.10) | $68.62 \pm 0.26$ | $\mathbf{68.83 \pm 0.16}$ | $68.67 \pm 0.13$ |
| HANDWRITING | $32.08 \pm 0.88$ (30.50) | $33.17 \pm 0.20$ | $33.29 \pm 0.42$ | $\mathbf{33.45 \pm 0.61}$ |
| HEARTBEAT | $75.77 \pm 1.01$ (77.60) | $76.10 \pm 0.98$ | $76.25 \pm 1.02$ | $\mathbf{76.26 \pm 1.07}$ |
| JAPANESEVOWELS | $99.46 \pm 0.27$ (99.40) | $\mathbf{99.55 \pm 0.31}$ | $99.46 \pm 0.27$ | $\mathbf{99.55 \pm 0.31}$ |
| PEMS-SF | $82.66 \pm 0.51$ (82.10) | $\mathbf{84.77 \pm 0.33}$ | $83.04 \pm 0.86$ | $83.81 \pm 0.62$ |
| SELFREGULATIONSCP1 | $91.46 \pm 0.35$ (92.50) | $91.58 \pm 0.39$ | $91.70 \pm 0.39$ | $\mathbf{92.04 \pm 0.36}$ |
| SELFREGULATIONSCP2 | $54.72 \pm 0.74$ (53.90) | $56.11 \pm 0.71$ | $55.93 \pm 0.85$ | $\mathbf{57.04 \pm 0.82}$ |
| SPOKENARABICDIGITS | $99.33 \pm 0.02$ (99.30) | $99.23 \pm 0.09$ | $99.34 \pm 0.11$ | $\mathbf{99.42 \pm 0.28}$ |
| UWAVEGESTURELIBRARY | $84.45 \pm 0.72$ (85.60) | $86.46 \pm 0.81$ | $86.77 \pm 0.78$ | $\mathbf{87.60 \pm 0.67}$ |
| AVERAGE ACCURACY | $72.07 \pm 0.47$ (72.27) | $72.89 \pm 0.09$ | $72.82 \pm 0.12$ | $\mathbf{73.22 \pm 0.33}$ |

**Remark 5** *For a given input sequence $\mathbf{X} := [\boldsymbol{x}_1, \cdots, \boldsymbol{x}_N]^\top \in \mathbb{R}^{N \times D_x}$ of $N$ feature vectors in self-attention, in order to generate the sets of $\{\boldsymbol{k}_j^s, \boldsymbol{v}_j^s\}_{j=1}^{N_s}$ at the scale $s^{th}$, we can downsample the input $\mathbf{X}$ before projecting into the key matrix $\mathbf{K}$ and the value matrix $\mathbf{V}$. There are multiples approaches to downsampling $\mathbf{X}$, such as using the average-pooling, max-pooling, 1-D convolution, or K-means clustering. In this paper, we employ the average-pooling to downsample $\mathbf{X}$.*

**Linear Attention with Batch Normalization and Scaled Heads.** The Attention-BN/SH can be extended to use with the linear attention. In particular, in the Linear Attention-BN/SH, we replace the softmax kernel in Eqn. 22 and Eqn. 25 by the linear kernel, respectively.

## 3 EXPERIMENTAL RESULTS

In this section, we empirically demonstrate the advantages of our Attention-BN, Attention-SH, and their combination (Attention-BN+SH) over the baseline softmax attention on the UEA time-series classification benchmark (Bagnall et al., 2018), the Long Range Arena benchmark (Tay et al., 2021), and the image classification task on the Imagenet dataset (Deng et al., 2009; Russakovsky et al., 2015). We aim to show that: (i) Attention-BN significantly outperforms the softmax baseline across tasks; (ii) Attention-SH achieves better or comparable accuracy while saving computation and memory compared to the baseline; (iii) Attention-BN+SH, which combines both Attention-BN and Attention-SH, results in the best model performance in term of accuracy and efficiency; (iv) all our proposed models help reduce the redundancy in multi-head attention and benefit learning of the long-term dependency in long input sequences; (v) Attention-BN and Attention-SH can be applied on other attention mechanisms beyond the softmax attention. When combined with the linear attention (Katharopoulos et al., 2020), the resultant Linear Attention-BN and Linear Attention-SH yield similar advantages mentioned in (i), (ii), (iii) and (iv) over the baseline linear attention.

In our experiments, we compare the proposed models with the baseline softmax and linear attentions of the same configuration. For the Attention-BN and Attention-BN+SH, we observe that recentering queries and keys alone is sufficient for improving the model performance. In addition, weighting $\boldsymbol{\mu}$ with a constant $\beta$, as in Eqn. 26 in the Appendix, enables the Attention-BN/BN+SH to adjust the effect of normalization to the attention score and help increase the accuracy. Our results are averaged over 5 runs. Details on datasets, models, and training are provided in Appendix A.

**UEA Time Series Classification.** Table 1 compares the accuracy of the Attention-BN and Attention-SH with the baseline softmax attention on 10 tasks in the UEA Time Series Classification benchmark (Bagnall et al., 2018). Both Attention-BN and Attention-SH significantly outperform the softmax baseline on most tasks and on average among all tasks. When combining two models, the resulting Attention-BN+SH yields the best accuracy with more than $1\%$ overall improvement over the softmax baseline. Notably, the Attention-SH and Attention-BN+SH are much more efficient than the baseline since they need much fewer keys and values in computing the attention output. The efficiency advantage of the Attention-SH/BN+SH is discussed and analyzed in detail in Section 4.

**Long Range Arena (LRA) benchmark.** In this experiment, we verify the advantage of our methods over the softmax baseline on tasks that involve very long sequences (e.g., the sequence length can be up to 4K) in the LRA benchmark (Tay et al., 2021). Those tasks require the model to capture long-range dependency in the input sequence. The summarized results in Table 2 indicate significant improvements of Attention-BN/SH/BN+SH over the baseline softmax attention. Same as in the UEA

Table 2: Test Accuracy (%) of the Attention-BN/SH/BN+SH vs. the baseline softmax attention on the LRA benchmark (Tay et al., 2021). Our models significantly outperform the softmax baseline.

| Dataset/Model | Baseline Softmax | Attention-BN | Attention-SH | Attention-BN+SH |
|---|---|---|---|---|
| LISTOPS | $36.76 \pm 0.42$ | $37.32 \pm 0.06$ | $37.08 \pm 0.48$ | $\mathbf{37.33 \pm 0.53}$ |
| TEXT | $64.90 \pm 0.07$ | $65.07 \pm 0.08$ | $\mathbf{65.19 \pm 0.19}$ | $65.03 \pm 0.35$ |
| RETRIEVAL | $79.68 \pm 0.52$ | $81.05 \pm 0.08$ | $80.74 \pm 0.32$ | $\mathbf{81.20 \pm 0.11}$ |
| IMAGE | $39.23 \pm 1.35$ | $\mathbf{39.75 \pm 1.21}$ | $38.87 \pm 1.24$ | $39.18 \pm 1.27$ |
| PATHFINDER | $72.72 \pm 0.75$ | $73.24 \pm 0.67$ | $\mathbf{74.00 \pm 0.29}$ | $73.98 \pm 0.55$ |
| AVERAGE ACCURACY | $58.66 \pm 0.26$ | $59.28 \pm 0.25$ | $59.18 \pm 0.22$ | $\mathbf{59.35 \pm 0.29}$ |

Table 3: Top-1 and top-5 accuracy (%) of the Attention-BN/SH/SH+BN Deit vs. the baseline softmax attention Deit on the ImageNet benchmark. The Attention-BN Deit outperforms the baseline in terms of accuracy. The Attention-SH/BN+SH Deit achieve comparable accuracy with the baseline while being more efficient.

| Metric/Model | Baseline Softmax Deit | Attention-BN Deit | Attention-SH Deit | Attention-BN+SH Deit |
|---|---|---|---|---|
| Top-1 Acc (%) | $72.23 \pm 0.23$ | $\mathbf{72.79 \pm 0.21}$ | $72.08 \pm 0.23$ | $72.25 \pm 0.22$ |
| Top-5 Acc (%) | $91.13 \pm 0.15$ | $\mathbf{91.43 \pm 0.12}$ | $91.05 \pm 0.14$ | $91.14 \pm 0.12$ |

Figure 1: (Left) FLOPS ratios and (Right) memory usage ratios between the Attention-BN+SH and the softmax baseline trained on retrieval task for different model dimensions and sequence lengths. The reduction in computation and memory when using our models improves with sequence length. When scaling up the model, our methods remain significantly more beneficial than the baseline.

Time Series experiment, on this LRA benchmark, Attention-BN and Attention-SH both outperform the softmax attention on most five tasks. Moreover, Attention-BN+SH, which combines these two attention mechanisms, results in the most accurate models on average across tasks. Specifically, for the retrieval task, the most challenging task with the largest sequence length in the LRA benchmark, Attention-BN+SH achieve a remarkable improvement of more than $1.5\%$ over the baseline.

**Image Classification on Imagenet.** We corroborate the advantage of our proposed attention over the baseline softmax attention when scaled up for the large-scale ImageNet image classification task. We summarize the results in Table 3. The Deit model (Touvron et al., 2021) equiped with the Attention-BN yields better performance than the softmax baseline. Meanwhile, Attention-SH/BN+SH Deit perform on par with the baseline while being more efficient. These results, together with other results above justify the benefits of our proposed methods across various tasks and data modalities, proving the effectiveness of our primal-dual approach to develop new attentions.

## 4 EMPIRICAL ANALYSIS

**Efficiency Analysis.** The Attention-BN+SH not only improves the accuracy of the model remarkably but also help reduce the computational and memory cost significantly. Fig.1 presents the efficiency benefits of our Attention-BN+SH trained on the retrieval task when the model dimension $D$ and sequence lengths $N$ grow. The efficiency advantage of our model increase as $N$ increase. In addition, the scaled-up models (with large $D$) remains significantly more efficient than the baseline. When the model dimension is 64 and sequence length is 4096, which is the standard configuration of the task, the model's FLOPS, in both training and inference, reduce almost $25\%$, whereas the reductions for memory usage in training and testing are $31.9\%$ and $47.3\%$, respectively. Notably, this efficient model also outperforms the baseline with more than $1.5\%$ improvement in accuracy. These results prove the benefits of applying the Attention-BN+SH for long-sequence tasks and large-scale models.

**New Attentions Helps Reduce Head Redundancy.** We compute the average $\mathcal{L}_2$ distances between heads to analyze the attention diversity. Given our trained models for the retrieval task, the layer-average mean and standard deviation of distances between heads are reported in Table 4. All our introduced attentions attain greater $\mathcal{L}_2$ distances compared to the baseline, reducing the risk of learning redundant heads. In particular, Attention-SH has the highest head difference, indicating the model's attention patterns are most spread out between heads.

Table 4: Layer-average mean and standard deviation of $\mathcal{L}_2$ distances between heads of Attention-BN/SH/BN+SH versus dot-product attention transformer trained on the retrieval task. Our attentions attain greater $\mathcal{L}_2$ distances between heads than the baseline, suggesting that they capture more diverse attention patterns.

| MetricModel | Baseline Softmax | Attention-BN | Attention-SH | Attention-BN+SH |
|---|---|---|---|---|
| Mean | 2.01 | 2.45 | **3.81** | 3.19 |
| Std | 0.39 | 0.66 | 0.75 | **1.01** |

**Combining Attention-BN and Attention-SH with Other Attentions.** Our methods can be extended to combine with other attention mechanisms. We study the Linear Attention-BN/SH/BN+SH, that combine the Attention-BN/SH/BN+SH with the linear attention (Katharopoulos et al., 2020) as explained at the end of Section 2.3. We summarize our results in Table 5 in Appendix B.1.

## 5   RELATED WORK

**Interpretation of Attention Mechanism.** Recent works have focused on understanding the attention mechanism in transformers from different perspectives. (Tsai et al., 2019) considers attention as a weighted moving average over the inputs via a smoothing kernel. (Nguyen et al., 2022) draws a connection between self-attention and nonparametric kernel regression. With this understanding, the work explores better regression estimators, e.g. the generalized Fourier nonparametric regression estimator, to improve transformers. In addition, (Cao, 2021) then shows that the linear transformer (Katharopoulos et al., 2020) corresponds to a Petrov-Galerkin projection (Reddy, 2004) and proves that the softmax normalization in the softmax attention is sufficient but not necessary. Other works that employ ordinary/partial differential equations to provide an interpretation for attention include (Lu et al., 2019; Sander et al., 2022). From a probabilistic perspective, (Tang & Matteson, 2021; Gabbur et al., 2021; Zhang & Feng, 2021) propose Gaussian mixture model frameworks to study the self-attention in transformers. Using graph-structured learning and message passing in graphical models is another attempt at understanding the attention mechanism Wang et al. (2018); Shaw et al. (2018); Kreuzer et al. (2021). Optimization perspectives of attention mechanisms are recently explored. (Sander et al., 2022) connects transformers with an optimization process across iterations by specifically constructing the core energy function. (Sahiner et al., 2022) derive finite-dimensional convex equivalence of attentions that can be solved for global optimality. Different from these approaches, our primal-dual framework focuses on deriving attention as the dual expansion of a primal neural network layer via solving a support vector regression problem. This framework allows us to not only explain many different types of attention mechanisms but also create new ones.

**Efficient Transformers.** Recently, efficient transformers have been studied (Roy et al., 2021). Among them are sparse transformers which incorporate sparse structures into the attention matrix (Parmar et al., 2018; Liu et al., 2018; Qiu et al., 2019; Child et al., 2019; Beltagy et al., 2020). Another class of efficient transformers are models that aim to have better coverage by integrating different access patterns (Child et al., 2019; Ho et al., 2019), which can also be learned from the data (Kitaev et al., 2020; Roy et al., 2021; Tay et al., 2020). An emerging body of work is proposed to distill and prune the model, including (Sanh et al., 2019; Sun et al., 2019; Voita et al., 2019; Sajjad et al., 2020). In other works, a side memory module is utilized in order to access multiple tokens simultaneously (Lee et al., 2019; Sukhbaatar et al., 2019; Asai & Choi, 2020; Beltagy et al., 2020). Low-rank and kernelization methods have been proposed to improve the efficiency of self-attention calculation (Tsai et al., 2019; Wang et al., 2020; Katharopoulos et al., 2020; Choromanski et al., 2021; Shen et al., 2021; Peng et al., 2021). Our Attention-SH/BN+SH is orthogonal to these methods.

## 6   CONCLUDING REMARKS

In this paper, we derive self-attention as a support vector expansion that solves a support vector regression (SVR) problem and provide a principled primal-dual framework to analyze and synthesize attention mechanisms. We then use our framework to invent two new attention mechanisms, the Batch Normalized Attention (Attention-BN) and the Attention with Scaled Heads (Attention-SH), that improve the accuracy and the efficiency of the baseline softmax attention. In our work, we approximate and learn the dual variables $\boldsymbol{\alpha}_j$ and $\tilde{\boldsymbol{\alpha}}_j$ using the value vector $\boldsymbol{v}_j$, $j = 1, \ldots, N$ in self-attention. It is natural to include more inductive biases and structures of those dual variables from solving the dual optimization problem of SVR into the value vectors $\boldsymbol{v}_j$. Furthermore, extending our framework to explain the attention modules that compute the attention using neural network layers or convolutional neural network layers applying on the input feature, such as the Convolutional Block Attention Module (Woo et al., 2018), is an interesting research direction. We leave these exciting research ideas as future work.

ACKNOWLEDGEMENTS

This material is based on research sponsored by the NSF under Grant# 2030859 to the Computing Research Association for the CIFellows Project (CIF2020-UCLA-38). SJO acknowledges support from the ONR N00014-20-1-2093 and N00014-20-1-2787 and the NSF DMS 2208272 and 1952339. RGB acknowledges support from the NSF grants CCF-1911094, IIS-1838177, and IIS-1730574; ONR grants N00014-18-12571, N00014-20-1-2534, and MURI N00014-20-1-2787; AFOSR grant FA9550-22-1-0060; and a Vannevar Bush Faculty Fellowship, ONR grant N00014-18-1-2047. ALB acknowledges support from the NSF grants DMS-2152717 and DMS-1952339. NH acknowledges support from the NSF IFML 2019844 and the NSF AI Institute for Foundations of Machine Learning.

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

# Supplement to "A Primal-Dual Framework for Transformers and Neural Networks"

## A  ADDITIONAL DETAILS ON THE EXPERIMENTS

This section provides datasets, models, and training details for experiments in Section 3. As mentioned in Section 3, for Attention-BN models, recentering queries and keys alone is sufficient for accuracy improvement, and we weight the mean $\boldsymbol{\mu}$ in Eqn 22 with a constant $\beta$. Hence Eqn 22 is simplified to:

$$\boldsymbol{h}_i = \sum_{j=1}^{N} \text{softmax}\left((\boldsymbol{q}_i - \beta\boldsymbol{\mu})^\top(\boldsymbol{k}_j - \beta\boldsymbol{\mu})/\sqrt{D}\right)\boldsymbol{v}_j. \tag{26}$$

In our experiments, we consider the constant $\beta$ in Attention-BN/BN+SH and the different downsampling scales in Attention-SH/SH+BN as hyper-parameters to finetune. All of our experiments are conducted on a server with 4 NVIDIA A100 GPUs.

### A.1  UEA TIME SERIES CLASSIFICATION

**Datasets and metrics** The benchmark (Bagnall et al., 2018) consists of 30 datasets. Following (Wu et al., 2022), we choose 10 datasets, which vary in input sequence lengths, the number of classes, and dimensionality, to evaluate our models on temporal sequences. We report the test accuracy as evaluation for the benchmark.

**Models and baselines** The experiment setups and configurations for the softmax/linear baseline and our models are the same as in (Wu et al., 2022) [1] (for the PEMS-SF, SelfRegulationSCP2, UWaveGestureLibrary datasets) and (Zerveas et al., 2021) [2] (for the other tasks). In all models, the number of heads is 8, whereas the model dimension and number of transformer layers are varied. For Attention-SH/SH+BN, we downsample keys and values by the factor of 2, after every two successive heads.

### A.2  LONG RANGE ARENA BENCHMARK

**Datasets and metrics** We adopt the tasks: Listops (Nangia & Bowman, 2018), byte-level IMDb reviews text classification (Maas et al., 2011), byte-level document retrieval (Radev et al., 2013), CIFAR-10 image classification (Krizhevsky et al., 2009) and the Pathfinder challenge (Linsley et al., 2018) in the LRA benchmark for our experiments. They consist of long sequences of length $2K$, $4K$, $4K$, $1K$, and $1K$ respectively. The evaluation protocol and metric are the same as in (Tay et al., 2021).

**Models and baselines** All our models and softmax/linear baselines follow the same architecture and configuration as in (Zhu et al., 2021) [3]. Each model consists of two layers and 64 embedding dimensions. While one head at each layer remains intact, the keys and values of the other heads are halved in our Attention-SH/SH+BN experiments.

### A.3  IMAGE CLASSIFICATION ON IMAGENET

**Datasets and metrics** The ImageNet dataset (Deng et al., 2009; Russakovsky et al., 2015) consists of $1.28M$ training images and $50K$ validation images. The task is to classify 1000 categories. Top-1 and top-5 accuracies are reported.

**Models and baselines** Our baseline is DeiT-tiny model (Touvron et al., 2021) with 12 transformer layers, 4 attention heads per layer, and the model dimension of 192. For model setting and setting and configuration, we follow (Touvron et al., 2021) [4]. The downsampling scales in Attention-SH/BN+SH models are $[1, 1, 2, 4]$ for 4 heads at each layer, respectively.

---

[1] Implementation available at https://github.com/thuml/Flowformer.

[2] Implementation available at https://github.com/gzerveas/mvts_transformer.

[3] Implementation available at https://github.com/NVIDIA/transformer-ls.

[4] Implementation available at https://github.com/facebookresearch/deit.

Table 5: Test Accuracy (%) of the Linear Attention-BN/SH/BN+SH vs. the baseline Linear Attention (Katharopoulos et al., 2020) on the UEA Time Series Classification Archive benchmark (Bagnall et al., 2018). Our proposed attentions outperform the baseline.

| Dataset/Model | *Baseline Linear* | Linear Attention-BN | Linear Attention-SH | Linear Attention-BN+SH |
|---|---|---|---|---|
| EthanolConcentration | $33.84 \pm 0.66$ | $\mathbf{34.98 \pm 0.74}$ | $34.76 \pm 0.69$ | $34.35 \pm 0.70$ |
| FaceDetection | $69.17 \pm 0.32$ | $69.22 \pm 0.17$ | $\mathbf{69.38 \pm 0.17}$ | $69.12 \pm 0.19$ |
| HandWriting | $32.87 \pm 0.27$ | $32.86 \pm 0.49$ | $32.82 \pm 0.12$ | $\mathbf{32.98 \pm 0.36}$ |
| HeartBeat | $75.61 \pm 0.73$ | $75.78 \pm 0.71$ | $74.96 \pm 0.62$ | $\mathbf{75.94 \pm 0.68}$ |
| JapaneseVowels | $99.37 \pm 0.16$ | $\mathbf{99.60 \pm 0.19}$ | $99.28 \pm 0.41$ | $99.33 \pm 0.19$ |
| PEMS-SF | $83.43 \pm 0.88$ | $85.74 \pm 0.67$ | $\mathbf{86.51 \pm 0.88}$ | $84.97 \pm 0.76$ |
| SelfRegulationSCP1 | $90.90 \pm 0.40$ | $91.81 \pm 0.69$ | $90.76 \pm 0.59$ | $\mathbf{91.92 \pm 0.60}$ |
| SelfRegulationSCP2 | $55.18 \pm 0.89$ | $\mathbf{56.11 \pm 0.94}$ | $54.44 \pm 0.88$ | $55.74 \pm 0.92$ |
| SpokenArabicDigits | $\mathbf{99.07 \pm 0.10}$ | $99.01 \pm 0.07$ | $99.03 \pm 0.18$ | $98.91 \pm 0.17$ |
| UWaveGestureLibrary | $85.63 \pm 0.81$ | $\mathbf{86.04 \pm 0.86}$ | $84.89 \pm 1.00$ | $85.78 \pm 0.75$ |
| Average Accuracy | $72.51 \pm 0.34$ | $\mathbf{73.12 \pm 0.20}$ | $72.68 \pm 0.40$ | $72.90 \pm 0.23$ |

Table 6: Top-1 and top-5 accuracy (%) of the Attention-Conv2D Deit vs. the baseline Deit with the softmax attention on the ImageNet image classification task. The Attention-Conv2D Deit significantly outperforms the baseline in both top-1 and top-5 accuracy.

| Metric/Model | *Baseline Softmax Deit* | Attention-Conv2D Deit |
|---|---|---|
| Top-1 Acc (%) | $72.23 \pm 0.23$ | $\mathbf{73.18 \pm 0.24}$ |
| Top-5 Acc (%) | $91.13 \pm 0.15$ | $\mathbf{91.52 \pm 0.13}$ |

Table 7: Test Accuracy (%) of the Attention-Conv1D vs. the baseline softmax attention on 5 tasks of the LRA benchmark (Tay et al., 2021). Our models outperform the softmax baseline.

| Dataset/Model | *Baseline Softmax* | Attention-Conv1D |
|---|---|---|
| ListOps | $36.76 \pm 0.42$ | $\mathbf{37.20 \pm 0.05}$ |
| Text | $64.90 \pm 0.07$ | $\mathbf{64.92 \pm 0.44}$ |
| Retrieval | $79.68 \pm 0.52$ | $\mathbf{80.75 \pm 0.15}$ |
| Image | $\mathbf{39.23 \pm 1.35}$ | $39.18 \pm 0.59$ |
| Pathfinder | $72.72 \pm 0.75$ | $\mathbf{73.01 \pm 0.24}$ |
| Average Accuracy | $58.66 \pm 0.26$ | $\mathbf{59.01 \pm 0.20}$ |

# B  Additional Experimental Results

## B.1  UEA Time Series Classification using the Linear Attention-BN/SH/BN+SH

Table 5 summarizes the comparison between the Linear Attention-BN/SH/BN+SH and the baseline Linear Attention on the UEA Time Series Classification task. The Linear Attention-BN/SH/BN+SH achieve better accuracy than the Linear Attention baseline while being more efficient.

## B.2  Convolution Attention

Table 6 demonstrates the advantage of Attention-Conv2D (Def. 3, Section G) over softmax Deit on the ImageNet image classification task. Furthemore, as shown in Table 7, the Attention-Conv1D (Def. 4, Section G) outperforms the baseline softmax attention on 5 tasks of the LRA benchmark (Tay et al., 2021).

## B.3  Additional experiments on the UEA Timeseries Classification benchmark and the UCR Time Series Regression Archive

In this section, we further demonstrate the advantage of our Attention-BN/SH/BN+SH on additional 15 tasks in the UEA Time Series Classification benchmark and on 6 tasks in the UCR Time Series Regression benchmark. The results in Table 8 and 9 show that our Attention-BN and Attention-SH+BN outperform the baseline softmax transformers significantly on both of these benchmarks, while the attention-SH has comparable performance with the baseline but being more effiicient.

## B.4  UEA Time Series Classification using the Sparse Attention-BN/SH/BN+SH

Table 10 summarizes the comparison between the Sparse Attention-BN/SH/BN+SH and the Sparse Attention baseline on a subset of the UEA Time Series Classification benchmark. Our models when combined with Sparse Attention achieve significantly better accuracy than the Sparse Attention baseline while the Sparse Attention-SH/BN+SH are more efficient (See Fig. 3 and Fig. 4 in Appendix C).

Table 8: Root mean square error (RMSE) of the Attention-BN/SH/BN+SH vs. the baseline softmax attention on 6 UCR Time Series Regression tasks (Tan et al., 2020). Smaller RMSE indicates better performance.

| Dataset/Model | Baseline Softmax | Attention-BN | Attention-SH | Attention-BN+SH |
|---|---|---|---|---|
| APPLIANCESENERGY | $3.44 \pm 0.06$ | $3.38 \pm 0.34$ | $3.39 \pm 0.02$ | $\mathbf{3.37 \pm 0.23}$ |
| BENZENECONCENTRATION | $0.91 \pm 0.03$ | $\mathbf{0.89 \pm 0.17}$ | $1.00 \pm 0.09$ | $0.90 \pm 0.08$ |
| BEIJINGPM10 | $92.31 \pm 1.06$ | $\mathbf{92.00 \pm 0.89}$ | $92.82 \pm 0.92$ | $92.40 \pm 0.85$ |
| BEIJINGPM25 | $59.73 \pm 1.21$ | $59.55 \pm 0.92$ | $59.66 \pm 0.88$ | $\mathbf{59.24 \pm 1.22}$ |
| LIVEFUELMOISTURE | $43.08 \pm 0.17$ | $\mathbf{43.01 \pm 0.50}$ | $43.65 \pm 0.09$ | $43.79 \pm 0.49$ |
| IEEEPPG | $32.12 \pm 1.25$ | $\mathbf{30.69 \pm 0.64}$ | $31.38 \pm 1.02$ | $30.73 \pm 1.20$ |
| AVERAGE RMSE | $38.60 \pm 0.67$ | $\mathbf{38.25 \pm 0.30}$ | $38.65 \pm 0.27$ | $38.40 \pm 0.51$ |

Table 9: Accuracy (%) of the Attention-BN/SH/BN+SH vs. the baseline softmax attention on other 15 UEA Time Series classification tasks (Bagnall et al., 2018).

| Dataset/Model | Baseline Softmax | Attention-BN | Attention-SH | Attention-BN+SH |
|---|---|---|---|---|
| ARTICULARYWORDRECOGNITION | $97.44 \pm 0.42$ | $98.22 \pm 0.87$ | $97.22 \pm 0.95$ | $\mathbf{98.44 \pm 0.41}$ |
| BASICMOTIONS | $98.75 \pm 1.25$ | $99.38 \pm 1.08$ | $99.37 \pm 1.06$ | $\mathbf{99.78 \pm 0.51}$ |
| EPILEPSY | $\mathbf{93.71 \pm 1.23}$ | $92.27 \pm 0.74$ | $89.13 \pm 1.07$ | $92.02 \pm 1.02$ |
| ERING | $95.18 \pm 0.52$ | $95.18 \pm 0.37$ | $94.72 \pm 0.66$ | $\mathbf{95.46 \pm 0.40}$ |
| FINGERMOVEMENTS | $59.67 \pm 0.47$ | $63.00 \pm 0.41$ | $61.33 \pm 0.70$ | $\mathbf{63.66 \pm 0.64}$ |
| LIBRAS | $85.00 \pm 0.45$ | $\mathbf{85.37 \pm 0.69}$ | $83.88 \pm 0.45$ | $85.00 \pm 0.78$ |
| NATOPS | $95.00 \pm 0.45$ | $95.37 \pm 0.26$ | $\mathbf{96.29 \pm 0.69}$ | $95.74 \pm 0.94$ |
| RACKETSPORTS | $87.28 \pm 0.82$ | $87.93 \pm 0.31$ | $88.16 \pm 0.54$ | $\mathbf{89.03 \pm 0.64}$ |
| ATRIALFIBRILLATION | $33.33 \pm 2.71$ | $41.67 \pm 2.88$ | $35.00 \pm 2.89$ | $\mathbf{41.68 \pm 2.80}$ |
| CRICKET | $94.90 \pm 0.65$ | $95.37 \pm 0.65$ | $93.98 \pm 0.65$ | $\mathbf{96.29 \pm 0.65}$ |
| STANDWALKJUMP | $50.00 \pm 2.33$ | $\mathbf{55.55 \pm 2.14}$ | $50.01 \pm 2.34$ | $55.00 \pm 2.08$ |
| HANDMOVEMENTDIRECTION | $63.96 \pm 2.30$ | $64.41 \pm 2.76$ | $61.71 \pm 2.64$ | $\mathbf{66.66 \pm 2.54}$ |
| LSST | $58.54 \pm 0.54$ | $57.05 \pm 0.26$ | $\mathbf{60.34 \pm 0.73}$ | $59.91 \pm 0.34$ |
| DUCKDUCKGEESE | $64.50 \pm 1.96$ | $65.00 \pm 1.73$ | $64.50 \pm 1.95$ | $\mathbf{65.50 \pm 1.66}$ |
| MOTORIMAGERY | $58.66 \pm 1.25$ | $60.67 \pm 1.69$ | $59.00 \pm 1.41$ | $\mathbf{62.00 \pm 0.81}$ |
| AVERAGE ACCURACY | $75.73 \pm 0.51$ | $77.10 \pm 0.22$ | $75.61 \pm 0.18$ | $\mathbf{77.74 \pm 0.24}$ |

Table 10: Test Accuracy (%) of the Sparse Attention-BN/SH/BN+SH vs. the baseline Sparse Attention (Child et al., 2019) on a subset of the UEA Time Series Classification Archive benchmark (Bagnall et al., 2018). Our proposed attentions outperform the baseline.

| Dataset/Model | Baseline Sparse | Sparse Attention-BN | Sparse Attention-SH | Sparse Attention-BN+SH |
|---|---|---|---|---|
| ETHANOLCONCENTRATION | $33.33 \pm 1.23$ | $33.33 \pm 0.78$ | $32.50 \pm 0.57$ | $\mathbf{33.46 \pm 0.71}$ |
| FACEDETECTION | $68.58 \pm 0.95$ | $68.65 \pm 0.44$ | $\mathbf{68.67 \pm 0.78}$ | $68.44 \pm 0.51$ |
| HANDWRITING | $31.08 \pm 0.38$ | $31.79 \pm 0.44$ | $32.75 \pm 0.39$ | $\mathbf{33.37 \pm 0.61}$ |
| HEARTBEAT | $74.95 \pm 0.81$ | $75.98 \pm 0.72$ | $74.96 \pm 0.80$ | $\mathbf{76.09 \pm 0.75}$ |
| JAPANESEVOWELS | $99.45 \pm 0.10$ | $\mathbf{99.54 \pm 0.12}$ | $99.18 \pm 0.14$ | $99.36 \pm 0.34$ |
| PEMS-SF | $82.08 \pm 0.63$ | $83.81 \pm 0.47$ | $82.66 \pm 0.63$ | $\mathbf{84.01 \pm 0.89}$ |
| SELFREGULATIONSCP1 | $91.24 \pm 0.85$ | $91.69 \pm 0.42$ | $91.47 \pm 0.84$ | $\mathbf{91.70 \pm 0.16}$ |
| SELFREGULATIONSCP2 | $55.18 \pm 0.69$ | $\mathbf{58.52 \pm 0.71}$ | $55.92 \pm 0.94$ | $56.67 \pm 0.68$ |
| SPOKENARABICDIGITS | $99.04 \pm 0.06$ | $99.10 \pm 0.15$ | $99.06 \pm 0.13$ | $\mathbf{99.15 \pm 0.09}$ |
| UWAVEGESTURELIBRARY | $84.90 \pm 0.39$ | $85.73 \pm 0.38$ | $85.31 \pm 0.88$ | $\mathbf{86.56 \pm 0.25}$ |
| AVERAGE ACCURACY | $71.98 \pm 0.38$ | $72.81 \pm 0.15$ | $72.25 \pm 0.24$ | $\mathbf{72.88 \pm 0.35}$ |

Table 11: Test Accuracy (%) of the Attention-BN/BN+SH with $\beta$ is learnable or set as a hyperparameter on the retrieval task (Tay et al., 2021).

| Model | Retrieval |
|---|---|
| Attention-BN (learn $\beta$) | $80.77 \pm 0.23$ |
| Attention-BN+SH (learn $\beta$) | $\mathbf{81.31 \pm 0.25}$ |
| Attention-BN ($\beta$ as a hyperparameter) | $81.05 \pm 0.08$ |
| Attention-BN+SH ($\beta$ as a hyperparameter) | $81.20 \pm 0.11$ |

### B.5 ATTENTION-BN/BN+SH WITH LEARNABLE $\beta$

We experiment with our Attention-BN/BN+SH with learnable $\beta$ on the retrieval task. Table 11 shows that learning $\beta$ does not improve much over setting $\beta$ to be a hyperparameter.

## C ADDITIONAL RESULTS ON EFFICIENCY ANALYSIS

This section provides more efficiency analysis on our models.

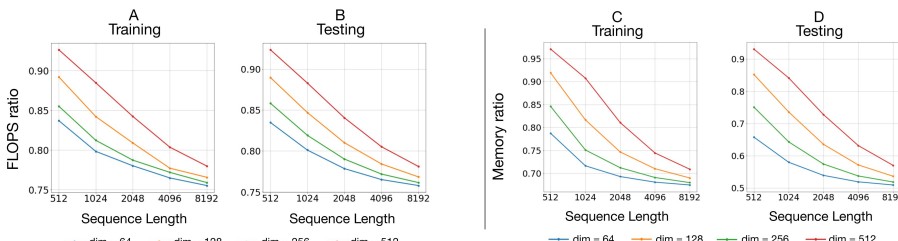

Figure 2: (Left) FLOPS ratios and (Right) memory usage ratios between the Attention-SH and the softmax attention baseline trained on the LRA retrieval task for different model dimensions and sequence lengths.

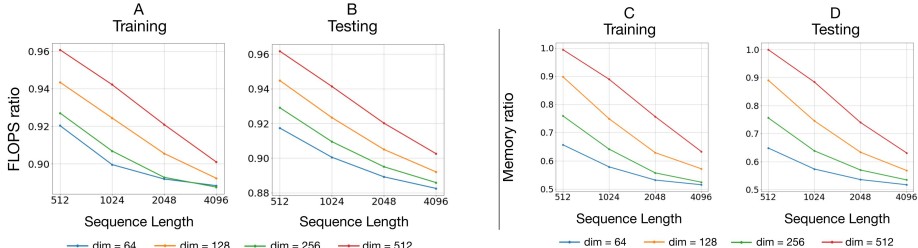

Figure 3: (Left) FLOPS ratios and (Right) memory usage ratios between the Sparse Attention-BN+SH and the Sparse Attention baseline trained on the LRA retrieval task for different model dimensions and sequence lengths. When using our models, the reduction in computation and memory improves with sequence length. When scaling up the model with greater model dimension, our methods remain significantly more efficient than the baseline.

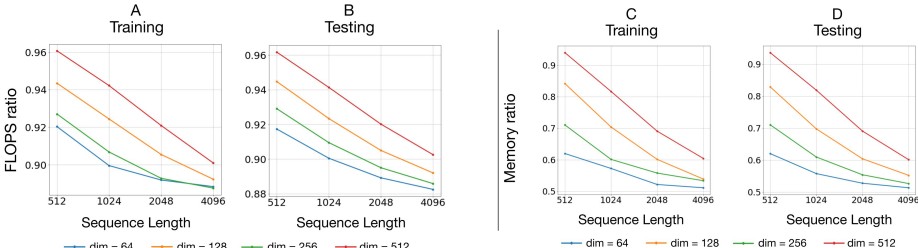

Figure 4: (Left) FLOPS ratios and (Right) memory usage ratios between the Sparse Attention-SH and the Sparse Attention baseline trained on the LRA retrieval task for different model dimensions and sequence lengths. When using our models, the reduction in computation and memory improves with sequence length. When scaling up the model with greater model dimension, our methods remain significantly more efficient than the baseline.

**Attention-SH.** Fig.2 shows the efficiency benefits of our Attention-SH when trained on the retrieval task. Same as in the case of Attention-SH+BN, the efficiency benefits of our Attention-SH over the baseline Softmax attention grows when $N$ and $D$ increase.

**Sparse Attention-SH/BN+SH.** Fig.3 and Fig.4 show that the efficiency advantages of our Sparse Attention-BN+SH and Sparse Attention-SH, respectively, increase as the model dimension $D$ and sequence length $N$ grow. All models are trained on the LRA retrieval task. In addition to the efficiency advantage, the Sparse Attention-BN+SH also significantly outperforms the Sparse Attention baseline in terms of accuracy in this task (79.86% vs. 78.20%) while the Sparse Attention-SH achieves a comparable result to the baseline. More accuracy advantages of the Sparse Attention-BN/SH/BN+SH over the Sparse Attention baseline are given in Table 10.

## D    DERIVING SOFTMAX ATTENTION.

Choosing the appropriate $h(\boldsymbol{x})$ and $\Phi(\boldsymbol{x})$ allows us to derive the popular softmax attention given in Eqn. 1 and 2. In particular, if we choose $h(\boldsymbol{x}) := \sum_j^N \Phi(\boldsymbol{x})^T \Phi(\boldsymbol{k}_j)$, Eqn. 10 becomes

$$f(\boldsymbol{x}) = \sum_{j=1}^{N} \frac{\Phi(\boldsymbol{x})^\top \Phi(\boldsymbol{k}_j)}{\sum_{j'}^{N} \Phi(\boldsymbol{x})^T \Phi(\boldsymbol{k}_{j'})} \boldsymbol{v}_j + \boldsymbol{b} = \frac{\sum_{j=1}^{N} \Phi(\boldsymbol{x})^\top \Phi(\boldsymbol{k}_j) \boldsymbol{v}_j}{\sum_{j'}^{N} \Phi(\boldsymbol{x})^T \Phi(\boldsymbol{k}_{j'})} + \boldsymbol{b}. \tag{27}$$

We then select $\Phi(\boldsymbol{x}) = \left(a_{l_0}^{(0)}, a_1^{(1)}, \ldots, a_{l_1}^{(1)}, \ldots, a_1^{(t)}, \ldots, a_{l_t}^{(t)}, \ldots\right)$ where $l_t = \binom{D+t-1}{t}$ and

$$a_l^{(t)} = \frac{(x_1/\sqrt[4]{D})^{n_1} \ldots (x_D/\sqrt[4]{D})^{n_D}}{\sqrt{n_1! \ldots n_D!}} \mid n_1 + \cdots + n_D = t,\ 1 \le l \le l_t. \tag{28}$$

Since

$$\exp\left(\boldsymbol{x}^T \boldsymbol{y}\right) = \sum_{t=0}^{\infty} \frac{(\boldsymbol{x}^T \boldsymbol{y})^t}{t!} = \sum_{t=0}^{\infty} \sum_{n_1+\cdots+n_D=t} \left(\frac{x_1^{n_1} \ldots x_D^{n_D}}{\sqrt{n_1! \ldots n_D!}}\right) \left(\frac{y_1^{n_1} \ldots y_D^{n_D}}{\sqrt{n_1! \ldots n_D!}}\right), \tag{29}$$

then Eqn. 27 becomes

$$f(\boldsymbol{x}) = \sum_{j=1}^{N} \frac{\sum_{t=0}^{\infty} \sum_{n_1+\cdots+n_D=t} \left(\frac{\left(\frac{x_1}{\sqrt[4]{D}}\right)^{n_1} \cdots \left(\frac{x_D}{\sqrt[4]{D}}\right)^{n_D}}{\sqrt{n_1!\ldots n_D!}}\right) \left(\frac{\left(\frac{k_{j1}}{\sqrt[4]{D}}\right)^{n_1} \cdots \left(\frac{k_{jD}}{\sqrt[4]{D}}\right)^{n_D}}{\sqrt{n_1!\ldots n_D!}}\right)}{\sum_{j'=1}^{N} \sum_{t=0}^{\infty} \sum_{n_1+\cdots+n_D=t} \left(\frac{\left(\frac{x_1}{\sqrt[4]{D}}\right)^{n_1} \cdots \left(\frac{x_D}{\sqrt[4]{D}}\right)^{n_D}}{\sqrt{n_1!\ldots n_D!}}\right) \left(\frac{\left(\frac{k_{j'1}}{\sqrt[4]{D}}\right)^{n_1} \cdots \left(\frac{k_{j'D}}{\sqrt[4]{D}}\right)^{n_D}}{\sqrt{n_1!\ldots n_D!}}\right)} \boldsymbol{v}_j + \boldsymbol{b}$$

$$= \sum_{j=1}^{N} \frac{\exp\left(\left(\frac{\boldsymbol{x}}{\sqrt[4]{D}}\right)^{\top} \frac{\boldsymbol{k}_j}{\sqrt[4]{D}}\right)}{\sum_{j'=1}^{N} \exp\left(\left(\frac{\boldsymbol{x}}{\sqrt[4]{D}}\right)^{\top} \frac{\boldsymbol{k}_{j'}}{\sqrt[4]{D}}\right)} \boldsymbol{v}_j + \boldsymbol{b} = \sum_{j=1}^{N} \frac{\exp\left(\boldsymbol{x}^{\top}\boldsymbol{k}_j/\sqrt{D}\right)}{\sum_{j'=1}^{N} \exp\left(\boldsymbol{x}^{\top}\boldsymbol{k}_{j'}/\sqrt{D}\right)} \boldsymbol{v}_j + \boldsymbol{b}. \tag{30}$$

Let $\boldsymbol{x} = \boldsymbol{q}_i$, $\boldsymbol{b} = 0$ and relax the boundness constraint of $\boldsymbol{v}_j$ in Remark 1. Eqn. 30 becomes Eqn. 2 of the softmax attention (Vaswani et al., 2017).

# E  BATCH NORMALIZED ATTENTION: DERIVATION OF EQN. 24

$$\boldsymbol{h}_i = \sum_{j=1}^{N} \text{softmax}\left(\sum_{d=1}^{D} \frac{(\boldsymbol{q}_i(d) - \boldsymbol{\mu}(d))(\boldsymbol{k}_j(d) - \boldsymbol{\mu}(d))}{\sqrt{D}(\sigma_d^2 + \epsilon)}\right) \boldsymbol{v}_j$$

$$= \sum_{j=1}^{N} \text{softmax}\left(\sum_{d=1}^{D} \frac{\boldsymbol{q}_i(d)\boldsymbol{k}_j(d) - \boldsymbol{q}_i(d)\boldsymbol{\mu}(d) - \boldsymbol{\mu}(d)\boldsymbol{k}_j(d) + \boldsymbol{\mu}(d)\boldsymbol{\mu}(d)}{\sqrt{D}(\sigma_d^2 + \epsilon)}\right) \boldsymbol{v}_j$$

$$= \sum_{j=1}^{N} \frac{\exp\left(\sum_{d=1}^{D} \frac{\boldsymbol{q}_i(d)\boldsymbol{k}_j(d) - \boldsymbol{q}_i(d)\boldsymbol{\mu}(d) - \boldsymbol{\mu}(d)\boldsymbol{k}_j(d) + \boldsymbol{\mu}(d)\boldsymbol{\mu}(d)}{\sqrt{D}(\sigma_d^2 + \epsilon)}\right)}{\sum_{j'=1}^{N} \exp\left(\sum_{d=1}^{D} \frac{\boldsymbol{q}_i(d)\boldsymbol{k}_{j'}(d) - \boldsymbol{q}_i(d)\boldsymbol{\mu}(d) - \boldsymbol{\mu}(d)\boldsymbol{k}_{j'}(d) + \boldsymbol{\mu}(d)\boldsymbol{\mu}(d)}{\sqrt{D}(\sigma_d^2 + \epsilon)}\right)} \boldsymbol{v}_j$$

$$= \sum_{j=1}^{N} \frac{\exp\left(\sum_{d=1}^{D} \frac{\boldsymbol{q}_i(d)\boldsymbol{k}_j(d) - \boldsymbol{\mu}(d)\boldsymbol{k}_j(d)}{\sqrt{D}(\sigma_d^2 + \epsilon)}\right) \exp\left(\sum_{d=1}^{D} \frac{\boldsymbol{\mu}(d)\boldsymbol{\mu}(d) - \boldsymbol{q}_i(d)\boldsymbol{\mu}(d)}{\sqrt{D}(\sigma_d^2 + \epsilon)}\right)}{\sum_{j'=1}^{N} \exp\left(\sum_{d=1}^{D} \frac{\boldsymbol{q}_i(d)\boldsymbol{k}_{j'}(d) - \boldsymbol{\mu}(d)\boldsymbol{k}_{j'}(d)}{\sqrt{D}(\sigma_d^2 + \epsilon)}\right) \exp\left(\sum_{d=1}^{D} \frac{\boldsymbol{\mu}(d)\boldsymbol{\mu}(d) - \boldsymbol{q}_i(d)\boldsymbol{\mu}(d)}{\sqrt{D}(\sigma_d^2 + \epsilon)}\right)} \boldsymbol{v}_j$$

$$= \sum_{j=1}^{N} \frac{\exp\left(\sum_{d=1}^{D} \frac{\boldsymbol{q}_i(d)\boldsymbol{k}_j(d) - \boldsymbol{\mu}(d)\boldsymbol{k}_j(d)}{\sqrt{D}(\sigma_d^2 + \epsilon)}\right)}{\sum_{j'=1}^{N} \exp\left(\sum_{d=1}^{D} \frac{\boldsymbol{q}_i(d)\boldsymbol{k}_{j'}(d) - \boldsymbol{\mu}(d)\boldsymbol{k}_{j'}(d)}{\sqrt{D}(\sigma_d^2 + \epsilon)}\right)} \boldsymbol{v}_j$$

$$= \sum_{j=1}^{N} \text{softmax}\left(\sum_{d=1}^{D} \frac{\boldsymbol{q}_i(d)\boldsymbol{k}_j(d) - \boldsymbol{\mu}(d)\boldsymbol{k}_j(d)}{\sqrt{D}(\sigma_d^2 + \epsilon)}\right) \boldsymbol{v}_j$$

$$= \sum_{j=1}^{N} \text{softmax}\left(\sum_{d=1}^{D} \frac{\boldsymbol{q}_i(d)\boldsymbol{k}_j(d) - \frac{1}{N}\sum_{j'=1}^{N} \boldsymbol{k}_{j'}(d)\boldsymbol{k}_j(d)}{\sqrt{D}(\sigma_d^2 + \epsilon)}\right) \boldsymbol{v}_j. \tag{31}$$

# F  ATTENTION WITH THE RESIDUAL CONNECTION AND MATRIX PROJECTIONS

In this supplement, we first discuss attention with the residual connection and matrix projections in Appendix F.

Suppose we are given a training data $\{(\boldsymbol{x}_1, \boldsymbol{y}_1), \ldots, (\boldsymbol{x}_N, \boldsymbol{y}_N)\} \subset \mathcal{X} \times \mathcal{Y}$, where $\mathcal{X} = \mathbb{R}^{D_x}$ and $\mathcal{Y} = \mathbb{R}^{D_v}$. Here, $\boldsymbol{x}_1, \ldots, \boldsymbol{x}_N$ are the training inputsd, and $\boldsymbol{y}_1, \ldots, \boldsymbol{y}_N$ are the training targets. In

order to derive the attention with the residual connection and query, key, and value matrix projections, we define the function $f$ as follows

$$y = f(x) := W \frac{\Phi(W^{\text{proj}}x)}{h(x)} + x + b, \tag{32}$$

where $x \in \mathcal{X} = \mathbb{R}^{D_x}$, $W^{\text{proj}} = [w_1^{\text{proj}}, \ldots, w_D^{\text{proj}}]^\top \in \mathbb{R}^{D \times D_x}$, $\Phi(\cdot) = [\phi_1(\cdot), \ldots, \phi_{D_\phi}(\cdot)] : \mathbb{R}^D \to \mathbb{R}^{D_\phi}$, $W = [w_1, \ldots, w_{D_v}]^\top \in \mathbb{R}^{D_v \times D_\phi}$, $b \in \mathbb{R}^{D_v}$, and $h(x)$ is a vector-scalar function. We fit the function $f$ to the training data $\{(x_1, y_1), \ldots, (x_N, y_N)\}$ with an $\mathbb{L}_2$ regularization on $W$ and $W^{proj}$ by solving the following convex optimization problem:

$$\underset{\substack{W, W^{\text{proj}} \\ \xi_j, \tilde{\xi}_j, j=1,\ldots,N}}{\text{minimize}} \quad \frac{1}{2} \sum_{d=1}^{D_v} \|w_d\|^2 + \frac{1}{2} \sum_{d=1}^{D_v} \|w_d^{\text{proj}}\|^2 + C \sum_{j=1}^{N} \sum_{d=1}^{D_v} \left( \xi_j(d) + \tilde{\xi}_j(d) \right)$$

subject to
$$\begin{cases} y_j(d) - w_d^\top \Phi(W^{\text{proj}}x_j)/h(x_j) - x_j - b(d) \leq \epsilon + \xi_j(d) \\ w_d^\top \Phi(W^{\text{proj}}x_j)/h(x_j) + x_j + b(d) - y_j(d) \leq \epsilon + \tilde{\xi}_j(d) \\ \xi_j(d), \tilde{\xi}_j(d) \geq 0 \end{cases}, \; j = 1,\ldots,N, \; d = 1,\ldots,D_v.$$

$$\tag{33}$$

The Lagrangian of the optimization problem 33 is given by

$$\mathcal{L}_1 := \frac{1}{2} \sum_{d=1}^{D_v} \|w_d\|^2 + \frac{1}{2} \sum_{d=1}^{D_v} \|w_d^{\text{proj}}\|^2 + C \sum_{j=1}^{N} \sum_{d=1}^{D_v} \left( \xi_j(d) + \tilde{\xi}_j(d) \right) - \sum_{j=1}^{N} \sum_{d=1}^{D_v} \left( \eta_j(d)\xi_j(d) + \tilde{\eta}_j(d)\tilde{\xi}_j(d) \right)$$

$$- \sum_{j=1}^{N} \sum_{d=1}^{D_v} \alpha_j(d) \left( \epsilon + \xi_j(d) - y_j(d) + w_d^\top \frac{\Phi(W^{\text{proj}}x_j)}{h(x_j)} + x_j + b(d) \right)$$

$$- \sum_{j=1}^{N} \sum_{d=1}^{D_v} \tilde{\alpha}_j(d) \left( \epsilon + \tilde{\xi}_j(d) + y_j(d) - w_d^\top \frac{\Phi(W^{\text{proj}}x_j)}{h(x_j)} - x_j - b(d) \right),$$

$$\tag{34}$$

Similar to the derivation in Section 2.1, the partial derivatives of $\mathcal{L}_1$ with respect to the primal variable $w_d$, $d = 1, \ldots, D_v$, have to vanish for optimality, which leads to

$$\partial_{w_d} \mathcal{L}_1 = w_d - \sum_{j=1}^{N} (\alpha_j(d) - \tilde{\alpha}_j(d)) \frac{\Phi(W^{\text{proj}}x_j)}{h(x_j)} = 0 \Rightarrow w_d = \sum_{j=1}^{N} (\alpha_j(d) - \tilde{\alpha}_j(d)) \frac{\Phi(W^{\text{proj}}x_j)}{h(x_j)}.$$

$$\tag{35}$$

Note that here we only find the form of the optimal solution for $W = [w_1, \ldots, w_{D_v}]^\top$. The optimal value of $W^{\text{proj}}$ can then be found by optimization algorithm such as the (stochastic) gradient descent when training the transformer.

Let $v_j = [\frac{\alpha_j(1) - \tilde{\alpha}_j(1)}{h(x_j)}, \ldots, \frac{\alpha_j(D_v) - \tilde{\alpha}_j(D_v)}{h(x_j)}]^\top, j = 1, \ldots, N$, we obtain the following support vector expansion of the function $f$:

$$f(x) = \left[ \sum_{j=1}^{N} (\alpha_j(1) - \tilde{\alpha}_j(1)) \frac{\Phi(W^{\text{proj}}x_j)}{h(x_j)}, \ldots, \sum_{j=1}^{N} (\alpha_j(D_v) - \tilde{\alpha}_j(D_v)) \frac{\Phi(W^{\text{proj}}x_j)}{h(x_j)} \right]^\top \frac{\Phi(W^{\text{proj}}x)}{h(x)} + x + b,$$

$$= \left[ \sum_{j=1}^{N} \frac{\alpha_j(1) - \tilde{\alpha}_j(1)}{h(x_j)} \frac{\Phi(W^{\text{proj}}x)^\top \Phi(W^{\text{proj}}x_j)}{h(x)}, \ldots, \sum_{j=1}^{N} \frac{\alpha_j(D_v) - \tilde{\alpha}_j(D_v)}{h(x_j)} \frac{\Phi(W^{\text{proj}}x)^\top \Phi(W^{\text{proj}}x_j)}{h(x)} \right]^\top + x + b,$$

$$= \sum_{j=1}^{N} \frac{\Phi(W^{\text{proj}}x)^\top \Phi(W^{\text{proj}}x_j)}{h(x)} v_j + \underbrace{x + b}_{\text{Residual connection}}. \tag{36}$$

Here, the support vector expansion of $f$ already includes a residual connection. The softmax attention can then be derived by selecting $h(x) := \sum_{j}^{N} \Phi(W^{\text{proj}}x)^T \Phi(W^{\text{proj}}x_j)$ and choosing $\Phi$ as in Eqn. 28 in Section 2.1. Note that in Eqn. 36, $\{x_j\}_{j=1}^{N}$ and $x$ are the training samples and test sample, respectively. In order to derive the key, query, and value matrix projections in attention, we can then relax Eqn. 36 by letting $W^{\text{proj}}x_j = W_K x_j$, $W^{\text{proj}}x = W_Q x$, $v_j = W_V x_j$ and choosing the test sample $x$ among the training samples $\{x_j\}_{j=1}^{N}$.

**Remark 6** *Here, for self-attention, we choose the test sample $\boldsymbol{x}$ among the training samples $\{\boldsymbol{x}_j\}_{j=1}^N$ to compute the attention score of a token to other tokens in the same sequence. For cross-attention where a token in a sequence attends to tokens in another sequence, this constraint can be removed.*

## G  2D-CONVOLUTION ATTENTION

In this section, we discuss attention with 2D-convolution. Suppose we are given a training data $\{(\boldsymbol{x}_1^{train}, \boldsymbol{y}_1^{train}), \ldots, (\boldsymbol{x}_{N_H \times N_W}^{train}, \boldsymbol{y}_{N_H \times N_W}^{train})\} \subset \mathcal{X} \times \mathcal{Y}$, where $\mathcal{X} = \mathbb{R}^{D_x}$ and $\mathcal{Y} = \mathbb{R}^{D_v}$. Here, $\boldsymbol{x}_1^{train}, \ldots, \boldsymbol{x}_{N_H \times N_W}^{train}$ are the training inputs, and $\boldsymbol{y}_1^{train}, \ldots, \boldsymbol{y}_{N_H \times N_W}^{train}$ are the training targets. Let $\mathbf{X}^{train} \in \mathbb{R}^{N_H \times N_W \times D_x}$ be the 3D-tensor of training inputs, where $\mathbf{X}^{train}(h, w, d) = \boldsymbol{x}_{N_W \times (h-1)+w}^{train}(d)$. Given a new set of inputs $\{\boldsymbol{x}_1, \ldots, \boldsymbol{x}_{N_H \times N_W}\} \subset \mathcal{X}$ and the corresponding 3D-tensor $\mathbf{X} \in \mathbb{R}^{N_H \times N_W \times D_x}$ of these inputs, where $\mathbf{X}(h, w, d) = \boldsymbol{x}_{N_W \times (h-1)+w}(d)$. We consider the function $f$ applying on the 3D-tensor $\mathbf{X}$ and taking the following form

$$f(\boldsymbol{x}_i) = \mathbf{W} \frac{\Phi(\text{Flatten}(\text{Conv2D}(\mathbf{X}, s))(i))}{h(\boldsymbol{x}_i)}, \ i = 1, \ldots, N_H \times N_W \tag{37}$$

where Conv2D is the depth-wise 2D-convolution (Howard et al., 2017), with the kernel size $s \times s$ and identical kernel channels, applied on the input tensor $\mathbf{X}$. Here, the last dimension of $\mathbf{X}$, i.e., $D_x$, is the depth. Also, $\Phi(\boldsymbol{x}) = [\phi_1(\boldsymbol{x}), \ldots, \phi_{D_\phi}(\boldsymbol{x})] \in \mathbb{R}^{D_\phi}$, $\mathbf{W} = [\boldsymbol{w}_1, \ldots, \boldsymbol{w}_{D_v}]^\top \in \mathbb{R}^{D_v \times D_\phi}$, $\boldsymbol{b} \in \mathbb{R}^{D_v}$, and $h$ is a vector-scalar function. We fit the function $f$ to the training data $\{(\boldsymbol{x}_1^{train}, \boldsymbol{y}_1^{train}), \ldots, (\boldsymbol{x}_{N_H \times N_W}^{train}, \boldsymbol{y}_{N_H \times N_W}^{train})\}$ with an $\mathbb{L}_2$ regularization on $\mathbf{W}$, i.e., a ridge regression, by solving the following convex optimization problem:

$$\begin{aligned}
\underset{\substack{\mathbf{W} \\ \boldsymbol{\xi}_j, \tilde{\boldsymbol{\xi}}_j, j=1,\ldots,N_H \times N_W}}{\text{minimize}} \quad & \frac{1}{2}\|\mathbf{W}\|_{\mathrm{F}}^2 + C \sum_{j=1}^{N_H \times N_W} \sum_{d=1}^{D_v} \left(\boldsymbol{\xi}_j(d) + \tilde{\boldsymbol{\xi}}_j(d)\right) = \frac{1}{2}\sum_{d=1}^{D_v}\|\boldsymbol{w}_d\|^2 + C \sum_{j=1}^{N_H \times N_W} \sum_{d=1}^{D_v} \left(\boldsymbol{\xi}_j(d) + \tilde{\boldsymbol{\xi}}_j(d)\right)
\end{aligned}$$

$$\text{subject to} \quad \begin{cases}
\boldsymbol{y}_j^{train}(d) - \boldsymbol{w}_d^\top \dfrac{\Phi(\text{Flatten}(\text{Conv2D}(\mathbf{X}^{train}, s))(j))}{h(\boldsymbol{x}_j^{train})} - \boldsymbol{b}(d) \le \epsilon + \boldsymbol{\xi}_j(d) \\[2mm]
\boldsymbol{w}_d^\top \dfrac{\Phi(\text{Flatten}(\text{Conv2D}(\mathbf{X}^{train}, s))(j))}{h(\boldsymbol{x}_j^{train})} + \boldsymbol{b}(d) - \boldsymbol{y}_j^{train}(d) \le \epsilon + \tilde{\boldsymbol{\xi}}_j(d) \\[2mm]
\boldsymbol{\xi}_j(d), \tilde{\boldsymbol{\xi}}_j(d) \ge 0, \ j = 1, \ldots, N_H \times N_W, \ d = 1, \ldots, D_v.
\end{cases} \tag{38}$$

The Lagrangian of the optimization problem 38 is given by

$$\begin{aligned}
\mathcal{L} := &\ \frac{1}{2}\sum_{d=1}^{D_v}\|\boldsymbol{w}_d\|^2 + C \sum_{j=1}^{N_H \times N_W} \sum_{d=1}^{D_v} \left(\boldsymbol{\xi}_j(d) + \tilde{\boldsymbol{\xi}}_j(d)\right) - \sum_{j=1}^{N_H \times N_W} \sum_{d=1}^{D_v} \left(\boldsymbol{\eta}_j(d)\boldsymbol{\xi}_j(d) + \tilde{\boldsymbol{\eta}}_j(d)\tilde{\boldsymbol{\xi}}_j(d)\right) \\
&- \sum_{j=1}^{N_H \times N_W} \sum_{d=1}^{D_v} \boldsymbol{\alpha}_j(d)\left(\epsilon + \boldsymbol{\xi}_j(d) - \boldsymbol{y}_j^{train}(d) + \boldsymbol{w}_d^\top \frac{\Phi(\text{Flatten}(\text{Conv2D}(\mathbf{X}^{train}, s))(j))}{h(\boldsymbol{x}_j^{train})} + \boldsymbol{b}(d)\right) \\
&- \sum_{j=1}^{N_H \times N_W} \sum_{d=1}^{D_v} \tilde{\boldsymbol{\alpha}}_j(d)\left(\epsilon + \tilde{\boldsymbol{\xi}}_j(d) + \boldsymbol{y}_j^{train}(d) - \boldsymbol{w}_d^\top \frac{\Phi(\text{Flatten}(\text{Conv2D}(\mathbf{X}^{train}, s))(j))}{h(\boldsymbol{x}_j^{train})} - \boldsymbol{b}(d)\right),
\end{aligned} \tag{39}$$

Similar to the derivation in Section 2.1 in the main text, the partial derivatives of $\mathcal{L}$ with respect to the primal variable $\boldsymbol{w}_d, d = 1, \ldots, D_v$, have to vanish for optimality, which leads to

$$\partial_{\boldsymbol{w}_d}\mathcal{L} = \boldsymbol{w}_d - \sum_{j=1}^{N_H \times N_W} (\boldsymbol{\alpha}_j(d) - \tilde{\boldsymbol{\alpha}}_j(d))\frac{\Phi(\text{Flatten}(\text{Conv2D}(\mathbf{X}^{train}, s))(j))}{h(\boldsymbol{x}_j^{train})} = 0 \tag{40}$$

$$\Rightarrow \boldsymbol{w}_d = \sum_{j=1}^{N_H \times N_W} (\boldsymbol{\alpha}_j(d) - \tilde{\boldsymbol{\alpha}}_j(d))\frac{\Phi(\text{Flatten}(\text{Conv2D}(\mathbf{X}^{train}, s))(j))}{h(\boldsymbol{x}_j^{train})}. \tag{41}$$

Let $\boldsymbol{v}_j = [\frac{\boldsymbol{\alpha}_j(1) - \tilde{\boldsymbol{\alpha}}_j(1)}{h(\boldsymbol{x}_j^{train})}, \ldots, \frac{\boldsymbol{\alpha}_j(D_v) - \tilde{\boldsymbol{\alpha}}_j(D_v)}{h(\boldsymbol{x}_j^{train})}]^\top$, $j = 1, \ldots, N_H \times N_W$, and substitute Eqn. 41 into Eqn. 38, we obtain the following support vector expansion of the linear basis function $f$:

$$f(\boldsymbol{x}_i) = \left[ \sum_{j=1}^{N_H \times N_W} \frac{\boldsymbol{\alpha}_j(1) - \tilde{\boldsymbol{\alpha}}_j(1)}{h(\boldsymbol{x}_j^{train})} \frac{\mathbf{A}_{ij}}{h(\boldsymbol{x}_i)}, \ldots, \sum_{j=1}^{N_H \times N_W} \frac{\boldsymbol{\alpha}_j(D_v) - \tilde{\boldsymbol{\alpha}}_j(D_v)}{h(\boldsymbol{x}_j^{train})} \frac{\mathbf{A}_{ij}}{h(\boldsymbol{x}_i)} \right]^\top + \boldsymbol{b},$$

$$= \sum_{j=1}^{N_H \times N_W} \frac{\mathbf{A}_{ij}}{h(\boldsymbol{x}_i)} \boldsymbol{v}_j + \boldsymbol{b}, \tag{42}$$

where $\mathbf{A}_{ij} := \Phi(\text{Flatten}(\text{Conv2D}(\mathbf{X}, s))(i))^\top \Phi(\text{Flatten}(\text{Conv2D}(\mathbf{X}^{train}, s))(j))$.

Same as in Section 2.1, we set $\boldsymbol{b}_s = 0$. To derive the softmax normalization in attention, we choose $h(\boldsymbol{x}_i) := \sum_{j=1}^{N} \mathbf{A}_{ij}$ and select $\Phi$ as in Eqn. 28. Let the training inputs $\{\boldsymbol{x}_1^{train}, \ldots, \boldsymbol{x}_{N_H \times N_W}^{train}\} \subset \mathcal{X}$ be the attention keys $\{\boldsymbol{k}_1, \ldots, \boldsymbol{k}_{N_H \times N_W}\} \subset \mathcal{K}$, where $\mathcal{K} = \mathbb{R}^D$, in self-attention. Also, let the new inputs $\{\boldsymbol{x}_1, \ldots, \boldsymbol{x}_{N_H \times N_W}\} \subset \mathcal{X}$ be the attention queries $\{\boldsymbol{q}_1, \ldots, \boldsymbol{q}_{N_H \times N_W}\} \subset \mathcal{K}$ in self-attention. We define the 2D-Convolution Attention (Attention-Conv2D) as follows:

**Definition 3 (2D-Convolution Attention)** *Given a set of the key and value vectors $\{\boldsymbol{k}_j, \boldsymbol{v}_j\}_{j=1}^{N_H \times N_W}$, and a set of the query vectors $\{\boldsymbol{q}_i\}_{i=1}^{N_H \times N_W}$. Denote the key tensor and the query tensor by $\mathbf{K} \in \mathbb{R}^{N_H \times N_W \times D}$ and $\mathbf{Q} \in \mathbb{R}^{N_H \times N_W \times D}$, respectively, where $\mathbf{K}(h, w, d) = \boldsymbol{k}_{N_W \times (h-1) + w}(d)$ and $\mathbf{Q}(h, w, d) = \boldsymbol{q}_{N_W \times (h-1) + w}(d)$. The 2D-Convolution Attention (Attention-Conv2D) computes the corresponding output vector $\boldsymbol{h}_i$ of the query $\boldsymbol{q}_i$ by the following attention formula:*

$$\boldsymbol{h}_i = \sum_{j=1}^{N} softmax \left( Flatten(Conv2D(\mathbf{Q}, s))(i)^\top Flatten(Conv2D(\mathbf{K}, s))(j) / \sqrt{D} \right) \boldsymbol{v}_j, \tag{43}$$

*where the $Conv2D(\cdot, s)$ is the depth-wise 2D-convolution (Howard et al., 2017) with the kernel size $s \times s$ and identical kernel channels.*

**Remark 7 (Convolutional Projection for Attention in the Convolutional vision Transformer)**
*The convolutional projections used in the Convolutional vision Transformer (CvT) (Wu et al., 2021) can be derived from Eqn. 42 by letting the training input tensor $\mathbf{X}^{train}$ to be the 2D input matrix of size $N \times D_x$ of the self-attention layer (see Section 1.1 in the main text) reshaped into a 3D tensor of size $N_H \times N_W \times D_x$ where $N = N_H \times N_W$. Here, to avoid confusion, we denote the input of the self-attention layer by $\mathbf{X}^{input}$ and its reshaped version by $Reshape2D(\mathbf{X}^{input})$. We then replace the depth-wise 2D-convolution by the depth-wise separable 2D-convolution in (Wu et al., 2021) and remove the constraint that the kernels have identical channels. In order to derive the convolutional projections for the keys, queries, and values in CvT, for $i, j = 1, \ldots, N$, we let*

$\boldsymbol{k}_j = Flatten(Conv2D(\mathbf{X}^{train}, s))(j)) = \Phi(Flatten(Conv2D(Reshape2D(\mathbf{X}^{input}), s, \mathbf{W}_K))(j),$

$\boldsymbol{q}_i = Flatten(Conv2D(\mathbf{X}, s))(i)) = \Phi(Flatten(Conv2D(Reshape2D(\mathbf{X}^{input}), s, \mathbf{W}_Q))(i),$

$\boldsymbol{v}_j = Flatten(Conv2D(\mathbf{X}^{train}, s))(j)) = \Phi(Flatten(Conv2D(Reshape2D(\mathbf{X}^{input}), s, \mathbf{W}_V))(j).$
$$\tag{44}$$

*Here, we specify the kernel/filter $\mathbf{W}_K$, $\mathbf{W}_Q$, and $\mathbf{W}_V$ to emphasize that the convolutional projections in CvT uses different kernels to compute keys, queries, and values in self-attention. Eqn. 44 matches the convolutional projects in CvT. By choosing $h$ and $\Phi$ similar to above, we can derive the convolutional attention in CvT.*

## H   1D-CONVOLUTION ATTENTION

Following the derivation for the Attention-Conv2D in Appendix G above, we can derive the 1D-Convolution Attention (Attention-Conv1D) in a similar way by letting $\mathbf{X}^{train} \in \mathbb{R}^{N \times D_x}$ and $\mathbf{X} \in \mathbb{R}^{N \times D_x}$ be 2D-matrices of training inputs and new inputs, respectively, and by replacing Conv2D by Conv1D, which is the depth-wise 1D-convolution, with the kernel size $s \times 1$ and identical kernel channels, applied on the input tensor $\mathbf{X}$. Here, the last dimension of $\mathbf{X}$, i.e., $D_x$, is the depth. We define the 1D-Convolution Attention (Attention-Conv1D) as follows:

**Definition 4 (1D-Convolution Attention)** *Given a set of the key and value vectors $\{\boldsymbol{k}_j, \boldsymbol{v}_j\}_{j=1}^{N}$, and a set of the query vectors $\{\boldsymbol{q}_i\}_{i=1}^{N}$. Denote the key matrix and the query matrix by $\mathbf{K} :=$*

$[\boldsymbol{k}_1, \ldots, \boldsymbol{k}_N]^\top \in \mathbb{R}^{N \times D}$ and $\mathbf{Q} := [\boldsymbol{q}_1, \ldots, \boldsymbol{q}_N]^\top \in \mathbb{R}^{N \times D}$, respectively. The 1D-Convolution Attention (Attention-Conv1D) computes the corresponding output vector $\boldsymbol{h}_i$ of the query $\boldsymbol{q}_i$ by the following attention formula:

$$\boldsymbol{h}_i = \sum_{j=1}^{N} softmax\left( Conv1D(\mathbf{Q}, s)(i)^\top Conv1D(\mathbf{K}, s)(j)/\sqrt{D} \right) \boldsymbol{v}_j, \tag{45}$$

where the $Conv1D(\cdot, s)$ is the depth-wise 1D-convolution with the kernel size $s \times 1$ and identical kernel channels.

## I  ATTENTION WITH BATCH NORMALIZATION AND SCALED HEADS

The Attention-BN+SH combines both the Attention-BN and Attention-SH. The Attention-BN+SH fits the function $f^s$, $s = 1, \ldots, H$, in Eqn. 17 with training sets $\{(\boldsymbol{k}_1^1, \boldsymbol{y}_1^1), \ldots, (\boldsymbol{k}_{N_1}^1, \boldsymbol{y}_{N_1}^1)\}, \ldots, \{(\boldsymbol{k}_1^H, \boldsymbol{y}_1^H), \ldots, (\boldsymbol{k}_{N_H}^H, \boldsymbol{y}_{N_H}^H)\} \subset \mathcal{K} \times \mathcal{Y}$ of different sizes $N_1, \ldots, N_H$, where $\mathcal{K} = \mathbb{R}^D$ and $\mathcal{Y} = \mathbb{R}^{D_v}$. The function $f^s$ is defined as:

$$f^s(\boldsymbol{x}) := \mathbf{W}^s \frac{\Phi((\boldsymbol{x} - \boldsymbol{\mu}^s) \odot \mathbf{s}^{s^{-1}})}{h^s((\boldsymbol{x} - \boldsymbol{\mu}^s) \odot \mathbf{s}^{s^{-1}})} + \boldsymbol{b}^s, \tag{46}$$

where

$$\boldsymbol{\mu}^s = \frac{1}{N_s} \sum_{j=1}^{N_s} \boldsymbol{k}_j^s, \ \mathbf{s}^{s^{-1}} = \left[ \frac{1}{\sqrt{\sigma_1^{s^2} + \epsilon}}, \ldots, \frac{1}{\sqrt{\sigma_D^{s^2} + \epsilon}} \right]^\top, \ \sigma_d^{s^2} = \frac{1}{N_s} \sum_{j=1}^{N_s} (\boldsymbol{k}_j^s(d) - \boldsymbol{\mu}^s(d))^2. \tag{47}$$

Following the same derivation as in Section 2.1, we derive the following support vector expansion of $f^s$

$$f^s(\boldsymbol{x}) = \sum_{j=1}^{N_s} \frac{\Phi((\boldsymbol{x} - \boldsymbol{\mu}^s) \odot \mathbf{s}^{s^{-1}})^\top \Phi((\boldsymbol{k}_j^s - \boldsymbol{\mu}^s) \odot \mathbf{s}^{s^{-1}})}{h^s((\boldsymbol{x} - \boldsymbol{\mu}^s) \odot \mathbf{s}^{s^{-1}})} \boldsymbol{v}_j^s + \boldsymbol{b}^s. \tag{48}$$

Here, $\boldsymbol{v}_j^s = \left[ \frac{\boldsymbol{\alpha}_j^s(1) - \tilde{\boldsymbol{\alpha}}_j^s(1)}{h^s((\boldsymbol{k}_j^s - \boldsymbol{\mu}^s) \odot \mathbf{s}^{s^{-1}})}, \ldots, \frac{\boldsymbol{\alpha}_j^s(D_v) - \tilde{\boldsymbol{\alpha}}_j^s(D_v)}{h^s((\boldsymbol{k}_j^s - \boldsymbol{\mu}^s) \odot \mathbf{s}^{s^{-1}})} \right]^\top$, where $\boldsymbol{\alpha}_j^s$ and $\tilde{\boldsymbol{\alpha}}_j^s$ are the dual variables, $j = 1, \ldots, N$. Same as in Section 2.1, in Eqn. 48, we choose $\Phi$ as in Eqn. 28, $h^s(\boldsymbol{x}) := \sum_j^{N_s} \Phi(\boldsymbol{x})^T \Phi(\boldsymbol{k}_j^s)$, and $\boldsymbol{b}^s = 0$ to obtain the Batch Normalized Attention with Scaled Heads (Attention-BN+SH), which is defined as follows:

**Definition 5 (Batch Normalized Attention with Scaled Heads)** *Given $H$ sets of the key and value vectors $\{\boldsymbol{k}_j^1, \boldsymbol{v}_j^1\}_{j=1}^{N_1}, \ldots, \{\boldsymbol{k}_j^H, \boldsymbol{v}_j^H\}_{j=1}^{N_H}$, for each set of $H$ query vectors $\boldsymbol{q}_i^1, \ldots, \boldsymbol{q}_i^H$, $i = 1, \ldots, N$, the Batch Normalized Attention with Scaled Heads (Attention-BN+SH) computes the corresponding output vector $\boldsymbol{h}_i$ of the queries $\boldsymbol{q}_i^1, \ldots, \boldsymbol{q}_i^H$ by the following attention formula:*

$$\boldsymbol{h}_i = \sum_{s=1}^{H} \mathbf{W}_O^s \left( \sum_{j=1}^{N_s} softmax\left( ((\boldsymbol{q}_i^s - \boldsymbol{\mu}^s) \odot \mathbf{s}^{s^{-1}})^\top ((\boldsymbol{k}_j^s - \boldsymbol{\mu}^s) \odot \mathbf{s}^{s^{-1}})/\sqrt{D} \right) \boldsymbol{v}_j^s \right), \tag{49}$$

*where*

$$\boldsymbol{\mu}^s = \frac{1}{N_s} \sum_{j=1}^{N_s} \boldsymbol{k}_j^s, \ \mathbf{s}^{s^{-1}} = \left[ \frac{1}{\sqrt{\sigma_1^{s^2} + \epsilon}}, \ldots, \frac{1}{\sqrt{\sigma_D^{s^2} + \epsilon}} \right]^\top, \ \sigma_d^{s^2} = \frac{1}{N_s} \sum_{j=1}^{N_s} (\boldsymbol{k}_j^s(d) - \boldsymbol{\mu}^s(d))^2. \tag{50}$$

Following the same Remark 5 in Section 2.3.2, given input sequence $\mathbf{X} := [\boldsymbol{x}_1, \cdots, \boldsymbol{x}_N]^\top \in \mathbb{R}^{N \times D_x}$ of $N$ feature vectors in self-attention, in order to generate the sets of $\{\boldsymbol{k}_j^s, \boldsymbol{v}_j^s\}_{j=1}^{N_s}$ at the scale $s^{th}$, we can downsample the input $\mathbf{X}$ before projecting into the key matrix $\mathbf{K}$ and the value matrix $\mathbf{V}$. In this paper, we use the average-pooling to downsample $\mathbf{X}$.

As in the same case of Attention-BN, for Attention-BN+SH, recentering queries and keys alone are sufficient for accuracy improvement, and we weight the mean $\boldsymbol{\mu}$ in Eqn 49 with a constant $\beta$. Hence Eqn. 49 is simplified to:

$$\boldsymbol{h}_i = \sum_{s=1}^{H} \mathbf{W}_O^s \left( \sum_{j=1}^{N_s} softmax\left( (\boldsymbol{q}_i^s - \beta\boldsymbol{\mu}^s)^\top (\boldsymbol{k}_j^s - \beta\boldsymbol{\mu}^s)/\sqrt{D} \right) \boldsymbol{v}_j^s \right). \tag{51}$$

Table 12: The values of $\beta$ for Linear Attention-BN/BN+SH and Sparse Attention-BN/BN+SH trained on the selected 10 UEA tasks.

| Dataset/Model | Linear Attention-BN | Linear Attention-BN+SH | Sparse Attention-BN | Sparse Attention-BN+SH |
|---|---|---|---|---|
| ETHANOLCONCENTRATION | 0.15 | 0.95 | 0.8 | 0.2 |
| FACEDETECTION | 0.6 | 0.6 | 0.6 | 0.6 |
| HANDWRITING | 0.25 | 0.3 | 0.3 | 0.3 |
| HEARTBEAT | 0.6 | 0.15 | 0.4 | 0.5 |
| JAPANESEVOWELS | 0.6 | 0.6 | 0.6 | 0.6 |
| PEMS-SF | 0.35 | 0.65 | 0.5 | 0.6 |
| SELFREGULATIONSCP1 | 0.35 | 0.25 | 0.1 | 0.9 |
| SELFREGULATIONSCP2 | 0.75 | 0.15 | 0.5 | 0.3 |
| SPOKENARABICDIGITS | 0.6 | 0.6 | 0.6 | 0.6 |
| UWAVEGESTURELIBRARY | 0.65 | 0.55 | 0.9 | 0.3 |

Table 13: The values of $\beta$ for Attention-BN/BN+SH trained on 25 UEA Time Series classification tasks (Bagnall et al., 2018) and 6 UEA Time Series Regression tasks.

| Dataset/Model | Attention-BN | Attention-BN+SH |
|---|---|---|
| ETHANOLCONCENTRATION | 0.25 | 0.15 |
| FACEDETECTION | 0.6 | 0.6 |
| HANDWRITING | 0.65 | 0.25 |
| HEARTBEAT | 0.55 | 0.85 |
| JAPANESEVOWELS | 0.6 | 0.6 |
| PEMS-SF | 0.25 | 0.35 |
| SELFREGULATIONSCP1 | 0.5 | 0.85 |
| SELFREGULATIONSCP2 | 1.2 | 0.9 |
| SPOKENARABICDIGITS | 0.65 | 0.6 |
| UWAVEGESTURELIBRARY | 0.1 | 0.2 |
| ARTICULARYWORDRECOGNITION | 0.2 | 0.6 |
| BASICMOTIONS | 0.1 | 0.1 |
| EPILEPSY | 0.3 | 0.2 |
| ERING | 0.1 | 0.9 |
| FINGERMOVEMENTS | 1.0 | 0.3 |
| LIBRAS | 0.3 | 0.7 |
| NATOPS | 1.0 | 0.4 |
| RACKETSPORTS | 1.0 | 0.4 |
| ATRIALFIBRILLATION | 0.9 | 0.6 |
| CRICKET | 0.2 | 1.0 |
| STANDWALKJUMP | 1.0 | 0.6 |
| HANDMOVEMENTDIRECTION | 0.5 | 0.5 |
| LSST | 0.3 | 0.2 |
| DUCKDUCKGEESE | 0.6 | 0.3 |
| MOTORIMAGERY | 0.2 | 0.3 |
| APPLIANCESENERGY | 0.4 | 0.2 |
| BENZENECONCENTRATION | 0.2 | 0.1 |
| BEIJINGPM10 | 0.1 | 0.5 |
| BEIJINGPM25 | 0.1 | 0.2 |
| LIVEFUELMOISTURE | 0.1 | 0.3 |
| IEEEPPG | 0.4 | 0.2 |

Table 14: The values of $\beta$ of Attention-BN/BN+SH trained on the 5 tasks of the LRA benchmark (Tay et al., 2021).

| Dataset/Model | Attention-BN | Attention-BN+SH |
|---|---|---|
| LISTOPS | 0.5 | 0.2 |
| TEXT | 0.5 | 0.8 |
| RETRIEVAL | 1.0 | 1.0 |
| IMAGE | 0.2 | 0.2 |
| PATHFINDER | 0.2 | 0.4 |

## J    HYPERPARAMETERS

In this section, we provide the hyper-parameters for our best models.

### J.1    UEA TIME SERIES CLASSIFICATION AND REGRESSION

For these two benchmarks, use the set of downsampling factors $\mathbf{s} = [1, 1, 2, 2, 4, 4, 8, 8]$ for Attention-SH/BN+SH and Linear/Sparse Attention-SH/BN+SH models trained on the UEA benchmark. Table 13 and Table 12 provide the values of $\beta$ used for our best Attention-BN/BN+SH and Linear Attention-BN/BN+SH, Sparse Attention-BN/BN+SH models trained on subsets of the two benchmarks.

### J.2 LONG RANGE ARENA BENCHMARK

For all 5 tasks of the LRA benchmark, we set the downsampling factors **s** of Attention-SH/BN+SH, Linear/Sparse Attention-SH/BN+SH is $[1, 2]$ and kernel size of Attention-Conv1D models is 5. In addition, Table 14 provides the values $\beta$ of Attention-BN/BN+SH models trained on the benchmark.

### J.3 IMAGENET CLASSIFICATION

This task's $\beta$ of Attention-BN/BN+SH is 1. Attention-SH/BN+SH has the downsampling factor of $[1, 1, 2, 4]$, and the kernel size of Attention-Conv2D is $(2, 2)$.

