# OpenReview forum: "A Primal-Dual Framework for Transformers and Neural Networks"
_ICLR.cc/2023/Conference — ICLR 2023 notable top 25%_

### Official Review · Reviewer_ZNRr · 2022-10-25

**Confidence:** 4
**Correctness:** 3
**Technical Novelty And Significance:** 4
**Empirical Novelty And Significance:** 3
**Recommendation:** 6

**Clarity, Quality, Novelty And Reproducibility:**

The paper is well-written. It is easy to understand the contributions of the paper. Based on support vector expansion, the proposed framework could derive many popular self-attention mechanisms. To further demonstrate the effectiveness, the paper develops two new self-attention mechanisms, which perform better than the softmax self-attention mechanism. However, the theoretical guidance of the proposed framework needs more elaborations.

**Details Of Ethics Concerns:**

The paper does not involve ethics concerns.

**Strength And Weaknesses:**

Strength:
The paper presents a framework to derive the popular self-attention mechanisms, which is based on support vector expansion. Then one may develop self-attention mechanisms that do not rely on heuristics and experience. Notably, the proposed framework could derive many popular self-attention mechanisms. Also, it invents two new mechanisms. These demonstrate the effectiveness of the proposed framework.

Weaknesses:
The main weakness is that the proposed framework lacks practical and theoretical guidance.
First, what is the motivation of the used function in (4)? Why is the function \Phi(x) divided by the vector-scalar function h(x)?
Second, are there any rules for choosing \Phi(x) and h(x)? Can any function be used as \Phi(x) or h(x)? The invented two new self-attention mechanisms still from the existing experiences.
Third, what makes one self-attention mechanism performs better? Could this be explained by the support vector expansion?
Forth, the paper shows that some self-attention mechanisms can be combined together. Can the proposed framework derive these combinations?

Other comments:
It is better to include the linear, sparse, multi-head self-attentions in all the comparisons.

**Summary Of The Paper:**

The paper proposes the primal-dual framework to derive popular self-attention mechanisms. The framework bases on a support vector expansion by solving a support vector regression problem. The paper shows that several popular self-attention mechanisms can be derived under the framework. It also invents two new self-attention mechanisms by employing the batch normalization and different amounts of training data, respectively. Experiments demonstrate the advantages of the proposed two self-attention mechanisms by comparing with the softmax self-attention.

**Summary Of The Review:**

The paper presents a framework to derive some popular self-attention mechanisms. It also invents two new self-attention mechanisms, which outperform the softmax self-attention. These two contributions are novel and well-supported. However, the proposed framework lacks practical and theoretical guidance, which needs to be further strengthened.

---

> ### Author Response · Authors · 2022-11-16
> **Response to Reviewer ZNRr (1)**
>
> Thank you for your thoughtful review and valuable feedback. Below we address your concerns.
>
> -----
> **Q1.  The main weakness is that the proposed framework lacks practical and theoretical guidance. First, what is the motivation of the used function in (4)? Why is the function $\Phi(x)$ divided by the vector-scalar function $h(x)$? Second, are there any rules for choosing $\Phi(x)$ and $h(x)$? Can any function be used as $\Phi(x)$ or $h(x)$?  The invented two new self-attention mechanisms still from the existing experiences.**
>
> **Reply:** Thanks for your comments. First, the function in Eqn. (4) is to derive the self-attention mechanism described by Eqn. (1) and (2) in our paper. The form of function f in Eqn. (4) is inherited from the support vector regression problem [1,2,3,4]. The vector-scalar function $h(x)$ provides us the flexibility to derive the Softmax attention or the linear attention as pointed out in paragraph “Deriving Softmax Attention” in Section 2.1 and in Section 2.2.1, “Linear Attention”, in our paper. Note that the function in Eqn. (4) has the form of a neural network layer, as pointed out in Remark 4 in Section 2.1 of our paper.
>
>
> Second, choosing $\Phi(x)$ can be guided by existing work in support vector regression. **SVR problems have been well-studied in literature and many theoretical results for SVR have been developed to provide a good understanding of this problem** [1,2,3,4]. For example, in [5] and [6], robust SVR and SVM formulations have been proposed with theoretical guarantees that can be used to develop a new class of robust attention via our primal-dual framework.  Furthermore, the relevance vector machine (RVM) [7,8] has been developed as a probabilistic sparse kernel model identical in functional form to the SVM, which again can be employed to develop new probabilistic attention via our primal-dual framework. Thus, **choosing $\Phi(x)$ via changing the SVR problem is principled and not ad-hoc**. In our paper, we propose the Attention with Scaled Heads (Attention-SH) as a simple but interesting example of how to develop new attention starting from the SVR problem in which we train multiple support vector regression problems using different amounts of training data to derive a new attention mechanism with multiscale heads. In Attention-SH, each token is allowed to attend to a group of tokens of different sizes at different attention heads.
>
> Choosing $h(x)$ can be motivated by neural network layers. Neural network layers have been intensively studied, especially over the last 10 years. Since the AlexNet [9] was proposed in 2012, many types of neural network layers have been developed to keep improving the state-of-the-art in a wide range of practical tasks, such as the residual layer [10], the squeeze-and-excitation layer [11], the inception layer [12], the batch normalization layer [13], and the neural tangent kernel layer [14]. **Compared to the attention layers, the neural network layers are better understood, and the experience in developing neural network layers has been accumulated in our community over many years. Our primal-dual framework allows the development of new attention to inherit from this understanding and experience in developing neural network layers**. The Batch Normalized Attention (Attention-BN) proposed in our paper is an example of how new attention can be developed from a well-known neural network layer, i.e. the batch normalization layer.
>
> It is true that both Attention-SH and Attention-BN are inspired by the existing experiences in the sense that Attention-SH is inspired by the multiscale approximation [15] and Attention-BN is derived from the batch normalization layer used in neural networks [13]. However, these attention mechanisms and their derivation are new and not overlapped with other existing attention mechanisms.

---

> > ### Author Response · Authors · 2022-11-16
> > **Response to Reviewer ZNRr (2)**
> >
> > **References**
> >
> > [1] Smola, Alex J., and Bernhard Schölkopf. "A tutorial on support vector regression." Statistics and computing 14, no. 3 (2004): 199-222.
> >
> > [2] Schölkopf, Bernhard, Alexander J. Smola, and Francis Bach. Learning with kernels: support vector machines, regularization, optimization, and beyond. MIT press, 2002.
> >
> > [3] Awad, Mariette, and Rahul Khanna. "Support vector regression." In Efficient learning machines, pp. 67-80. Apress, Berkeley, CA, 2015.
> >
> > [4] Drucker, Harris, Christopher J. Burges, Linda Kaufman, Alex Smola, and Vladimir Vapnik. "Support vector regression machines." Advances in neural information processing systems 9 (1996).
> >
> > [5] Lv, Yuan, and Zhong Gan. "Robust ε-support vector regression." Mathematical Problems in Engineering (2014).
> >
> > [6] Xu, Huan, Constantine Caramanis, and Shie Mannor. "Robustness and Regularization of Support Vector Machines." Journal of machine learning research 10, no. 7 (2009).
> >
> > [7] Tipping, Michael. "The relevance vector machine." Advances in neural information processing systems 12 (1999).
> >
> > [8] Tipping, Michael E. "Sparse Bayesian learning and the relevance vector machine." Journal of machine learning research 1, no. Jun (2001): 211-244.
> >
> > [9] Krizhevsky, Alex, Ilya Sutskever, and Geoffrey E. Hinton. "Imagenet classification with deep convolutional neural networks." Communications of the ACM 60, no. 6 (2017): 84-90.
> >
> > [10] He, Kaiming, Xiangyu Zhang, Shaoqing Ren, and Jian Sun. "Deep residual learning for image recognition." In Proceedings of the IEEE conference on computer vision and pattern recognition, pp. 770-778. 2016.
> >
> > [11] Hu, Jie, Li Shen, and Gang Sun. "Squeeze-and-excitation networks." In Proceedings of the IEEE conference on computer vision and pattern recognition, pp. 7132-7141. 2018.
> >
> > [12] Szegedy, Christian, Vincent Vanhoucke, Sergey Ioffe, Jon Shlens, and Zbigniew Wojna. "Rethinking the inception architecture for computer vision." In Proceedings of the IEEE conference on computer vision and pattern recognition, pp. 2818-2826. 2016.
> >
> > [13] Ioffe, Sergey, and Christian Szegedy. "Batch normalization: Accelerating deep network training by reducing internal covariate shift." In International conference on machine learning, pp. 448-456. PMLR, 2015.
> >
> > [14] Jacot, Arthur, Franck Gabriel, and Clément Hongler. "Neural tangent kernel: Convergence and generalization in neural networks." Advances in neural information processing systems 31 (2018).
> >
> > [15] Mallat, Stéphane. A wavelet tour of signal processing. Elsevier, 1999.

---

> > > ### Author Response · Authors · 2022-11-16
> > > **Response to Reviewer ZNRr (3)**
> > >
> > > **Q2. Third, what makes one self-attention mechanism performs better? Could this be explained by the support vector expansion?**
> > >
> > > **Reply:** What makes one self-attention mechanism performs better depends on what we would like to achieve. For example, as mentioned in the reply to Q1, if we would like to obtain an attention mechanism that is robust to noise and perturbations, then robust SVR and SVM [5,6] can be used to develop the corresponding robust attention. Other examples are our Attention-BN and Attention-SH. The advantage of Attention-BN is to improve the accuracy of the model by taking into account the similarity between the key vectors (see Tables 1, 2, 3, 5, 8, 9, and 10 in our paper). The advantage of Attention-SH is to enhance the efficiency of the model by using the multiscale heads (see Figure 2 and 4 in Appendix C of the revised manuscript). These two attentions are complementary to each other. Tables 1, 2, 3, 5, 8, 9, and 10 and Figure 1 and 3 in our paper show that combining Attention-BN and Attention-SH, i.e. the Attention-BN+SH, improves both the model’s accuracy and efficiency.
> > >
> > > **References**
> > >
> > > [5] Lv, Yuan, and Zhong Gan. "Robust ε-support vector regression." Mathematical Problems in Engineering (2014).
> > >
> > > [6] Xu, Huan, Constantine Caramanis, and Shie Mannor. "Robustness and Regularization of Support Vector Machines." Journal of machine learning research 10, no. 7 (2009).
> > >
> > > -----
> > > **Q3. Forth, the paper shows that some self-attention mechanisms can be combined together. Can the proposed framework derive these combinations?**
> > >
> > > **Reply:** Yes, our proposed framework can derive these combinations. We have included the derivation for Attention-BN+SH in Appendix I of our revised manuscript. The derivation for the combination of the Linear Attention and Attention-BN/SH can be easily achieved by extending the derivation of the Attention-BN/SH with the normalization h(x) chosen for the Linear Attention as in Section 2.2.1 of our paper.
> > >
> > > -----
> > > **Q4. It is better to include the linear, sparse, multi-head self-attentions in all the comparisons.**
> > >
> > > **Reply:** Thanks for your suggestion. We have included the Linear Attention in Table 5 in Appendix B.1 of our paper. The baseline results included in Table 1, 2, 3, 4, 6, 7, 8, and 9 are for the baseline Softmax multi-head self-attentions.
> > >
> > > Following your suggestion, in addition to the Linear Attention, we have also combined our Attention-BN and Attention-SH with the Sparse Attention [1] and summarized the results in comparison with the Sparse Attention in Table 10 in Appendix B.4 of our revised manuscript. As shown in Table 10, when combining with the Sparse Attention, our Sparse Attention-BN, Sparse Attention-SH, and Sparse Attention-BN+SH outperform the Sparse Attention in terms of accuracy while our Sparse Attention-BN+SH and Attention-SH are more efficient than the baseline Sparse Attention. The efficiency advantages of the Sparse Attention-BN+SH and Sparse Attention-SH over the baseline Sparse Attention grow as the model dimension D and the sequence length N increase, as presented in Figure 3 and Figure 4 in Appendix C of our revised manuscript.
> > >
> > > **References**
> > >
> > > [1] Child, Rewon, Scott Gray, Alec Radford, and Ilya Sutskever. "Generating long sequences with sparse transformers." arXiv preprint arXiv:1904.10509 (2019).
> > >
> > > -----
> > >
> > > We hope we have cleared your concerns about our work. We have also revised our manuscript according to your comments, and we would appreciate it if we can get your further feedback at your earliest convenience.

---

> > > > ### Author Response · Authors · 2022-11-24
> > > > **Response to Reviewer ZNRr - Any further questions on our current draft**
> > > >
> > > > We would like to thank you again for your thoughtful reviews and valuable feedback.
> > > >
> > > > We would appreciate it if you could let us know if our responses have addressed your concerns and whether you still have any other questions on the current draft and our rebuttal.
> > > >
> > > > We would be happy to do any follow-up discussion or address any additional comments.

---

### Official Review · Reviewer_2btF · 2022-10-25

**Confidence:** 4
**Correctness:** 4
**Technical Novelty And Significance:** 4
**Empirical Novelty And Significance:** 3
**Recommendation:** 8

**Clarity, Quality, Novelty And Reproducibility:**

The clarity of the paper is great, with clear organization and well-written mathematical derivations. The technical derivation from a SVR model to self-attention seems novel, but the motivation and significance of this derivation is unclear at the moment. Specifically, the two newly proposed attention mechanisms derived from the primal-dual framework does not seem novel compared to existing work. Reproducibility is good as experimental code has been shared alongside the main draft. Adding information on the hyperparameter tuning procedure (as mentioned in the previous block) would make it even better.

**Strength And Weaknesses:**

### Strengths

- The paper presents an interesting discovery that connects neural network layers and self-attention via a primal-dual formulation of the SVR problem.
- A wide range of experimentation has been conducted across multiple well-known benchmarks, together covering various input modalities such as time-series and images.
- The advantage of proposed methods is also supported with comprehensive analyses measuring the empirical efficiency as well as redundancy across attention heads.

### Weaknesses

- The motivation is a bit unclear: the introduction mentions that "most attention layers are developed based on intuitions and heuristic approaches", but the SVR formulation and presented experiments do not provide much further insight, as to whether utilizing the dual-formulation of a neural network layer can bring significant benefit towards better attention-based model diversity and/or performance.
- Related to the previous comment, the new attention mechanisms, Attention-BN and Attention-SH, somewhat overlap with existing methods to be considered empirically novel. For instance, Attention-SH with sequence subsampling is similar to existing work on convolution-augmented attention [1,2]. Further utilizing the different design choices in the SVR formulation (e.g. choosing a different normalization $h(x)$) could bring a larger impact and increase the value of the primal-dual interpretation.
- Experimental results are slightly lacking as it does not show results across all 30 and 5 tasks available in UEA Time-Series and Long Range Arena benchmarks, respectively. Is there a specific reason why experiments were run on subsets from the two benchmarks instead of the entire set of tasks? If not, could the comprehensive results from all tasks be shared as well?

[1] Gulati et al., 2020. Conformer: Convolution-augmented Transformer for Speech Recognition.
[2] Wu et al., 2021. CvT: Introducing Convolutions to Vision Transformers.


### Other questions/comments

- What are the hyperparameter setups tested for 1) the mean weighting factor $\beta$ and 2) downsampling scales for Attention-SH/SH+BN? For better reproducibility, what were the actual values used for the results presented?
- Is there a reason behind having the weighting factor $\beta$ as a constant given as hyperparameter? In order to have Attention-BN/BN+SH "flexibly adjust the effect of normalization", letting $\beta$ be a trainable parameter similar to the scale and shift parameters in BatchNorm seems more suitable than fixing it as constant.
- Given that all experimental results are averaged over 5 runs with different seeds, could the standard deviations be added in Tables 1, 2, and 3?
- How is the head redundancy measured with average $\mathcal{L}_2$-distance? What do the "distances between heads" exactly measure?
- In Equation 12, gathering all the $D^{1/4}$ terms in the numerator results in $D^{j/4}$, which doesn't lead to the $\sqrt{D}$ term in Equation 14. Is there an additional step that is missing?
- Duplicate indexing by $j$ in Equation 11.
- Typo in Remark 1: "As a results" &rarr; "As a result"
- Typo after Equation 24: "... less and vice versus." &rarr; "... less and vice versa."
- Typo in Table 1 (Attention-BN+SH on HeartBeat): Should it be 76.26 instead of 72.26?
- Typo in Section 4: "... more efficiency than the baseline." &rarr; "... more efficient than the baseline."
- Typo in Appendix A: "... recentering keys and values alone ..." &rarr; Should it be "... recentering queries and keys alone ..."?

**Summary Of The Paper:**

This paper proposes a primal-dual interpretation between neural network layers and self-attention through support vector expansion from a Support Vector Regression (SVR) problem. Based on this insight, the paper derives well-known attention mechanisms such as linear, sparse, and multi-head attention. The paper also proposes two new attention methods, 1) Batch Normalized Attention (Attention-BN) and 2) Attention with Scaled Head (Attention-SH), leveraging the SVR interpretation. Experiments on UEA Time-Series, Long Range Arena, and ImageNet benchmarks show that the proposed methods show improvements in performance as well as efficiency compared to the baseline softmax attention.

**Summary Of The Review:**

The paper proposes an interesting direction of interpreting self-attention as a dual formulation of the SVR problem on a neural network layer. However, the newly proposed attention mechanisms as well as presented experiments are not convincing enough to fully assert the significance of the primal-dual framework.

---

> ### Author Response · Authors · 2022-11-16
> **Response to Reviewer 2btF (1)**
>
> Thank you for your thoughtful review and valuable feedback. Below we address your concerns.
>
> -----
>
> **Q1. The motivation is a bit unclear: the introduction mentions that "most attention layers are developed based on intuitions and heuristic approaches", but the SVR formulation and presented experiments do not provide much further insight, as to whether utilizing the dual-formulation of a neural network layer can bring significant benefit towards better attention-based model diversity and/or performance.**
>
> **Reply:** Thanks for your comments. We respectfully disagree with the reviewer’s comment that SVR formulation does not provide much further insight into the development of attention. We also believe there is a misunderstanding of the benefit of our primal-dual framework for attention layers and neural network layers. Please allow us to clear this misunderstanding by explaining our principled approach to designing new attention and then discussing the Attention-BN and Attention-SH as two examples of our principled design strategy for attention. Given our primal-dual framework for attention layers and neural network layers via solving a Support Vector Regression problem, **new attentions can be invented by 1) modifying the support vector regression problem and 2) adjusting the primal neural network layer in our framework.**
>
> 1. **SVR problems have been well-studied in literature and many theoretical results for SVR have been developed to provide a good understanding of this problem** [1,2,3,4]. For example, in [5] and [6], robust SVR and SVM formulations have been proposed with theoretical guarantees that can be used to develop a new class of robust attention via our primal-dual framework.  Furthermore, the relevance vector machine (RVM) [7,8] has been developed as a probabilistic sparse kernel model identical in functional form to the SVM, which again can be employed to develop new probabilistic attention via our primal-dual framework. Thus, **changing the SVR problem is principled and not an ad-hoc approach based on intuition and heuristics**. In our paper, we propose the Attention with Scaled Heads (Attention-SH) as a simple but interesting example of how to develop new attention starting from the SVR problem in which we train multiple support vector regression problems using different amounts of training data to derive a new attention mechanism with multiscale heads. In Attention-SH, each token is allowed to attend to a group of tokens of different sizes at different attention heads.
> 2. Neural network layers have been intensively studied, especially over the last 10 years. Since the AlexNet [9] was proposed in 2012, many types of neural network layers have been developed to keep improving the state-of-the-art in a wide range of practical tasks, such as the residual layer [10], the squeeze-and-excitation layer [11], the inception layer [12], the batch normalization layer [13], and the neural tangent kernel layer [14]. **Compared to the attention layers, the neural network layers are better understood, and the experience in developing neural network layers has been accumulated in our community over many years. Our primal-dual framework allows the development of new attention to inherit from this understanding and experience in developing neural network layers.** The Batch Normalized Attention (Attention-BN) proposed in our paper is an example of how new attention can be developed from a well-known neural network layer, i.e. the batch normalization layer.

---

> > ### Author Response · Authors · 2022-11-16
> > **Response to Reviewer 2btF (2)**
> >
> > **References**
> >
> > [1] Smola, Alex J., and Bernhard Schölkopf. "A tutorial on support vector regression." Statistics and computing 14, no. 3 (2004): 199-222.
> >
> > [2] Schölkopf, Bernhard, Alexander J. Smola, and Francis Bach. Learning with kernels: support vector machines, regularization, optimization, and beyond. MIT press, 2002.
> >
> > [3] Awad, Mariette, and Rahul Khanna. "Support vector regression." In Efficient learning machines, pp. 67-80. Apress, Berkeley, CA, 2015.
> >
> > [4] Drucker, Harris, Christopher J. Burges, Linda Kaufman, Alex Smola, and Vladimir Vapnik. "Support vector regression machines." Advances in neural information processing systems 9 (1996).
> >
> > [5] Lv, Yuan, and Zhong Gan. "Robust ε-support vector regression." Mathematical Problems in Engineering (2014).
> >
> > [6] Xu, Huan, Constantine Caramanis, and Shie Mannor. "Robustness and Regularization of Support Vector Machines." Journal of machine learning research 10, no. 7 (2009).
> >
> > [7] Tipping, Michael. "The relevance vector machine." Advances in neural information processing systems 12 (1999).
> >
> > [8] Tipping, Michael E. "Sparse Bayesian learning and the relevance vector machine." Journal of machine learning research 1, no. Jun (2001): 211-244.
> >
> > [9] Krizhevsky, Alex, Ilya Sutskever, and Geoffrey E. Hinton. "Imagenet classification with deep convolutional neural networks." Communications of the ACM 60, no. 6 (2017): 84-90.
> >
> > [10] He, Kaiming, Xiangyu Zhang, Shaoqing Ren, and Jian Sun. "Deep residual learning for image recognition." In Proceedings of the IEEE conference on computer vision and pattern recognition, pp. 770-778. 2016.
> >
> > [11] Hu, Jie, Li Shen, and Gang Sun. "Squeeze-and-excitation networks." In Proceedings of the IEEE conference on computer vision and pattern recognition, pp. 7132-7141. 2018.
> >
> > [12] Szegedy, Christian, Vincent Vanhoucke, Sergey Ioffe, Jon Shlens, and Zbigniew Wojna. "Rethinking the inception architecture for computer vision." In Proceedings of the IEEE conference on computer vision and pattern recognition, pp. 2818-2826. 2016.
> >
> > [13] Ioffe, Sergey, and Christian Szegedy. "Batch normalization: Accelerating deep network training by reducing internal covariate shift." In International conference on machine learning, pp. 448-456. PMLR, 2015.
> >
> > [14] Jacot, Arthur, Franck Gabriel, and Clément Hongler. "Neural tangent kernel: Convergence and generalization in neural networks." Advances in neural information processing systems 31 (2018).

---

> > > ### Author Response · Authors · 2022-11-16
> > > **Response to Reviewer 2btF (3)**
> > >
> > > **Q2. Related to the previous comment, the new attention mechanisms, Attention-BN and Attention-SH, somewhat overlap with existing methods to be considered empirically novel. For instance, Attention-SH with sequence subsampling is similar to existing work on convolution-augmented attention [1,2]. Further utilizing the different design choices in the SVR formulation (e.g. choosing a different normalization h(x)  could bring a larger impact and increase the value of the primal-dual interpretation.**
> > >
> > > **[1] Gulati et al., 2020. Conformer: Convolution-augmented Transformer for Speech Recognition.**
> > >
> > > **[2] Wu et al., 2021. CvT: Introducing Convolutions to Vision Transformers.**
> > >
> > > **Reply:** We respectfully disagree with the reviewer’s comment that our Attention-BN and Attention-SH overlap with existing methods to be considered empirically novel. Please allow us to clear this misunderstanding by clarifying the key differences between our Attention-SH with [1] and [2] above. Since the reviewers did not point out existing work similar to our Attention-BN, we assume the reviewer agrees with us that our Attention-BN is a new attention that does not exist in other work.
> > >
> > > In [1], the encoder first subsamples the input sequence X once and then send it to a number of conformer blocks, which is the multi-head attention blocks with feedforward and convolutional layers. In contrast, our Attention-SH subsamples the input sequence X along the sequence length dimension at the beginning of each multi-head attention block to compute the X_downsampled. X_downsampled is then used to compute the key matrix K and the value matrix V while the original input X is used to compute the query matrix Q. Thus, the input subsampling in Attention-SH is different and complementary to the input subsampling in the conformer. Both methods can be combined in a way such that the input is subsampled at the beginning of the encoder and at each attention block in the encoder. We have included [1] in the Introduction section of our paper.
> > >
> > > Our Attention-SH is close to the squeezed convolutional projection in [2]. However, there are two main differences. First, we directly downsample the input sequence X along the sequence length dimension. The downsampled version of X, i.e. X_downsampled as in above, is used to compute the key matrix K and the value matrix V via linear transformations. Multiple methods can be used to downsample the input sequence X including the average-pooling, max-pooling, 1-D convolution, or K-means clustering. In our experiments, we use the average-pooling to downsample X. In the squeezed convolutional projection in [2], the strided convolutional projections are applied on the input sequence X to compute the key matrix K and the value matrix V. Second, the squeezed convolutional projection in [2] requires tokens to be reshaped into a 2D token map before the strided convolutions are applied while our Attention-SH does not need this reshaping step.
> > >
> > > In addition, **Remark 7 in Appendix G of our revised manuscript shows that the convolutional projections proposed in [2] can be derived from our primal-dual framework. In Appendix G and H of our revised manuscript, we have also derived our new Attention-Conv2D and Attention-Conv1D from the 2D-convolution and 1D-convolution layers in neural networks, respectively**. Both of these new convolutional attention layers outperform the baseline softmax attention as shown in Table 6 and Table 7 in Appendix B.2 of the revised manuscript. On the ImageNet image classification task, our Attention-Conv2D achieves the Top-1 accuracy and Top-5 accuracy of 73.18% and 91.52% respectively while the corresponding results of the baseline softmax attention are 72.23% and 91.13%. On the LRA benchmark, our Attention-Conv1D achieves an average accuracy of 59.01%  vs. 58.66% obtained by the baseline softmax attention. Note that since our new Attention-Conv2D is derived from the 2D-convolution layer, it is intuitively good for image processing tasks such as image classification. Thus, in Table 6, we compare our Attention-Conv2D with the baseline softmax attention on the ImageNet image classification task. Compared to the attention with convolutional projections in [2], our new Attention-Conv2D needs fewer parameters with competitive performance since we apply the same depth-wise 2D-convolution with identical kernel channels on the queries, the keys, and the values.
> > >
> > > We thank the reviewers for your suggestion that “Further utilizing the different design choices in the SVR formulation (e.g. choosing a different normalization h(x)  could bring a larger impact and increase the value of the primal-dual interpretation.” In our paper, we have chosen the appropriate normalization h(x) to derive the softmax and linear attention. Further investigating other forms for h(x) to derive new attention mechanisms is an interesting research direction, and we will explore it in future work.

---

> > > > ### Author Response · Authors · 2022-11-16
> > > > **Response to Reviewer 2btF (4)**
> > > >
> > > > **Q3. Experimental results are slightly lacking as it does not show results across all 30 and 5 tasks available in UEA Time-Series and Long Range Arena benchmarks, respectively. Is there a specific reason why experiments were run on subsets from the two benchmarks instead of the entire set of tasks? If not, could the comprehensive results from all tasks be shared as well?**
> > > >
> > > > **Reply:** Following your suggestion, we have included results for all 5 tasks in the Long Range Arena (LRA) benchmarks in Table 2 of our revision. Our Attention-BN still outperforms the baseline Softmax attention in the additional Image Classification and Pathfinder tasks. Our Attention-SH and Attention-BN+SH outperform the baseline Softmax attention in the Pathfinder task and achieve comparable results to the Softmax attention in the Image Classification task while being more efficient than the Softmax attention. All of our proposed attention outperform the baseline Softmax attention on the average accuracy across 5 LRA tasks.
> > > >
> > > > It is common to show experimental results for only 10 or 11 tasks in the UEA Time Series since this benchmark includes many tasks. However, following the reviewer’s suggestion, we have conducted experiments on additional 15 UEA Time Series classification tasks. We have also compared our Attention-BN and Attention-SH with the baseline Softmax attention on 6 tasks in the UCR Time Series Regression benchmark [1]. We include these new results in Table 8 and Table 9 in Appendix B.3 of our revision. Our proposed attentions still outperform the baseline Softmax attention on these additional tasks.
> > > >
> > > > **References**
> > > >
> > > > [1] Tan, Chang Wei, Christoph Bergmeir, François Petitjean, and Geoffrey I. Webb. "Time series extrinsic regression." Data Mining and Knowledge Discovery 35, no. 3 (2021): 1032-1060.
> > > >
> > > > -----
> > > > **Q4. What are the hyperparameter setups tested for 1) the mean weighting factor β and 2) downsampling scales for Attention-SH/SH+BN? For better reproducibility, what were the actual values used for the results presented?**
> > > >
> > > > **Reply:** Thanks for your suggestion. We have provided values for the mean weighting factor $\beta$ and the downsampling scales used in our experiments in Appendix J of our revised manuscript.
> > > >
> > > > -----
> > > > **Q5. Is there a reason behind having the weighting factor β as a constant given as hyperparameter? In order to have Attention-BN/BN+SH "flexibly adjust the effect of normalization", letting β be a trainable parameter similar to the scale and shift parameters in BatchNorm seems more suitable than fixing it as constant.**
> > > >
> > > > **Reply:** Following the reviewer’s suggestion, we have conducted experiments where we set $\beta$ to be a trainable parameter. The results are not improved much compared to setting $\beta$ to be a hyperparameter. For example, on the LRA retrieval task, Attention-BN and Attention-BN+SH with learnable $\beta$ achieve the accuracy of 80.77 and 81.31 compared to 81.05 and 81.20 when setting $\beta$ to be a hyperparameter, respectively. However, setting $\beta$ to be a learnable parameter as the reviewer suggested helps alleviate the time and effort required to finetune the value for $\beta$. We have added these new results with learnable $\beta$ in Table 11 in Appendix B.5 of our revised manuscript.
> > > >
> > > > -----
> > > > **Q6. Given that all experimental results are averaged over 5 runs with different seeds, could the standard deviations be added in Tables 1, 2, and 3?**
> > > >
> > > > **Reply:** Following the reviewer’s suggestion, we have added the standard deviation to our results in Tables 1, 2, and 3, as well as Tables 5, 6, 7, 8, 9, 10, and 11, in the revised manuscripts.
> > > >
> > > > -----
> > > > **Q7. How is the head redundancy measured with average L2-distance? What do the "distances between heads" exactly measure?**
> > > >
> > > > **Reply:** For a given pre-trained model, we compute the pair-wise L2-distances between the attention matrices $A=\text{softmax}(QK^{T}/\sqrt{D})$ at different attention heads in the same layer. We show the layer-average mean and variance of distances between these attention matrices in Attention-BN, Attention-SH, and Attention-BN+SH compared with those in the baseline softmax attention in Table 4 in our paper.

---

> > > > > ### Author Response · Authors · 2022-11-16
> > > > > **Response to Reviewer 2btF (5)**
> > > > >
> > > > > **Q8. In Equation 12, gathering all the D^{¼} terms in the numerator results in D^{j/4}, which doesn't lead to the D term in Equation 14. Is there an additional step that is missing?**
> > > > >
> > > > > **Reply:** Eqn. 12 in our paper is correct. In Eqn. 14, when we factor the $\sqrt{D}$  term into the product of two $\sqrt[4]{D}$  terms and apply each of these $\sqrt[4]{D}$ terms to vector $x$ and $k_{j}$, this  $\sqrt[4]{D}$  term will show up in each element of vector $x$ as in Eqn. 12. We have made this point clear in Eqn. 27, 28, 29, and 30 in Appendix D of our revised manuscript.
> > > > >
> > > > > -----
> > > > > **Q9. Duplicate indexing by j in Equation 11. Typo in Remark 1: "As a results" → "As a result". Typo after Equation 24: "... less and vice versus." → "... less and vice versa." Typo in Table 1 (Attention-BN+SH on HeartBeat): Should it be 76.26 instead of 72.26? Typo in Section 4: "... more efficiency than the baseline." → "... more efficient than the baseline." Typo in Appendix A: "... recentering keys and values alone ..." → Should it be "... recentering queries and keys alone ..."?**
> > > > >
> > > > > **Reply:** Thank you so much for pointing these out. We have fixed these typos in our revised manuscript.
> > > > >
> > > > > -----
> > > > > We hope we have cleared your concerns about our work. We have also revised our manuscript according to your comments, and we would appreciate it if we can get your further feedback at your earliest convenience.

---

> > > > > > ### Author Response · Authors · 2022-11-24
> > > > > > **Response to Reviewer 2btF - Any further questions on our current draft**
> > > > > >
> > > > > > We would like to thank you again for your thoughtful reviews and valuable feedback.
> > > > > >
> > > > > > We would appreciate it if you could let us know if our responses have addressed your concerns and whether you still have any other questions on the current draft and our rebuttal.
> > > > > >
> > > > > > We would be happy to do any follow-up discussion or address any additional comments.

---

> ### Comment · Reviewer_2btF · 2022-11-25
> **Response to Authors**
>
> Thank you authors for the detailed response with further derivations and results. The responses were very helpful in addressing my major concerns, and led to a much clearer understanding of the contribution. As a result, I increase my score to a solid accept.
>
> On a sidenote, there were several minor corrections in the manuscript that might be worth checking (listed below):
> - In the LRA benchmark paragraph of Section 3: "Attention-BN and Attention-SH both outperform... on all three tasks" &rarr; Should it be "on four out of five tasks"?
> - The Datasets and metrics paragraph in Section A.2 is missing descriptions on Image and Pathfinder tasks.
> - Row "SpokenArabicDigits" in Table 5 doesn't have the best result in bold.
> - Rows "RacketSports" and "DuckDuckGeese" in Table 9 are missing bolded results as well.
> - In Section J.2: "For for" &rarr; "For"

---

> > ### Author Response · Authors · 2022-11-25
> > **Thanks for your endorsement!**
> >
> > Thanks for your response and we appreciate your endorsement. We have fixed the typos you pointed out in our revision. We will update our submission with the latest revised manuscript when the paper revision is enabled.

---

### Official Review · Reviewer_fLYg · 2022-10-29

**Confidence:** 4
**Correctness:** 4
**Technical Novelty And Significance:** 3
**Empirical Novelty And Significance:** 3
**Recommendation:** 6

**Clarity, Quality, Novelty And Reproducibility:**

This paper is well written,  and the mathematic interpretation is of high quality. The introduction can be longer to discuss more motivation of this work.


**Strength And Weaknesses:**

Strength: This paper gave detailed mathematical proof of the relation between the attention mechanism and the support vector regression, which inspired me a lot because most previous attention methods were designed based on intuition and experiences. This paper also proposed a framework for how to develop attention that is meaningful for future research.

Weaknesses: The motivation and intuitive interpretation of Attention-BN and Attention-SH should be discussed more. I can not fully understand what inspired you to design these two novel attention methods. More insightful discussion can be provided. And one thing comes to my mind, can these two attention methods benefit some practical vision problems, such as semantic segmentation?  Can the proposed framework help develop some CNN-related attention modules, such as CBAM?


**Summary Of The Paper:**

This paper first proves that the self-attention mechanism in neural networks is a special form of support vector regression. Based on this conclusion, the authors proposed a principle to develop attention and designed the Attention-BN and the Attention-SH for improving accuracy and efficiency. Experimental results demonstrated that their methods have better performance than conventional attention methods.

**Summary Of The Review:**

This paper gives me a new perspective to understand attention. This research is meaningful to deep learning, and my score is 6.

---

> ### Author Response · Authors · 2022-11-16
> **Response to Reviewer fLYg (1)**
>
> Thank you for your thoughtful review and valuable feedback. Below we address your concerns.
>
> -----
> **Q1. The motivation and intuitive interpretation of Attention-BN and Attention-SH should be discussed more. I can not fully understand what inspired you to design these two novel attention methods. More insightful discussion can be provided.**
>
> **Reply:** Thanks for your comments. We will first explain our principled approach to designing new attention and then discuss the Attention-BN and Attention-SH as two examples of our principled design strategy for attention. Given our primal-dual framework for attention layers and neural network layers via solving a Support Vector Regression problem, **new attentions can be invented by 1) modifying the support vector regression problem and 2) adjusting the primal neural network layer in our framework.**
>
> 1. **SVR problems have been well-studied in literature and many theoretical results for SVR have been developed to provide a good understanding of this problem** [1,2,3,4]. For example, in [5] and [6], robust SVR and SVM formulations have been proposed with theoretical guarantees that can be used to develop a new class of robust attention via our primal-dual framework.  Furthermore, the relevance vector machine (RVM) [7,8] has been developed as a probabilistic sparse kernel model identical in functional form to the SVM, which again can be employed to develop new probabilistic attention via our primal-dual framework. Thus, **changing the SVR problem is principled and not an ad-hoc approach based on intuition and heuristics**. In our paper, we propose the Attention with Scaled Heads (Attention-SH) as a simple but interesting example of how to develop new attention starting from the SVR problem in which we train multiple support vector regression problems using different amounts of training data to derive a new attention mechanism with multiscale heads. In Attention-SH, each token is allowed to attend to a group of tokens of different sizes at different attention heads.
>
> 2. Neural network layers have been intensively studied, especially over the last 10 years. Since the AlexNet [9] was proposed in 2012, many types of neural network layers have been developed to keep improving the state-of-the-art in a wide range of practical tasks, such as the residual layer [10], the squeeze-and-excitation layer [11], the inception layer [12], the batch normalization layer [13], and the neural tangent kernel layer [14]. **Compared to the attention layers, the neural network layers are better understood, and the experience in developing neural network layers has been accumulated in our community over many years. Our primal-dual framework allows the development of new attention to inherit from this understanding and experience in developing neural network layers.** The Batch Normalized Attention (Attention-BN) proposed in our paper is an example of how new attention can be developed from a well-known neural network layer, i.e. the batch normalization layer. In Attention-BN, the similarity between the query $q_{i}$ and the key  $k_{j}$ is adjusted by the similarity between the key $k_{j}$ and all the keys $k_{j'}$, $j'=1,\dots,N$. In particular, if the key $k_{j}$ is too similar to other keys, the query $q_{i}$ will attend to it less and vice versa.

---

> > ### Author Response · Authors · 2022-11-16
> > **Response to Reviewer fLYg (2)**
> >
> > **References**
> >
> > [1] Smola, Alex J., and Bernhard Schölkopf. "A tutorial on support vector regression." Statistics and computing 14, no. 3 (2004): 199-222.
> >
> > [2] Schölkopf, Bernhard, Alexander J. Smola, and Francis Bach. Learning with kernels: support vector machines, regularization, optimization, and beyond. MIT press, 2002.
> >
> > [3] Awad, Mariette, and Rahul Khanna. "Support vector regression." In Efficient learning machines, pp. 67-80. Apress, Berkeley, CA, 2015.
> >
> > [4] Drucker, Harris, Christopher J. Burges, Linda Kaufman, Alex Smola, and Vladimir Vapnik. "Support vector regression machines." Advances in neural information processing systems 9 (1996).
> >
> > [5] Lv, Yuan, and Zhong Gan. "Robust ε-support vector regression." Mathematical Problems in Engineering (2014).
> >
> > [6] Xu, Huan, Constantine Caramanis, and Shie Mannor. "Robustness and Regularization of Support Vector Machines." Journal of machine learning research 10, no. 7 (2009).
> >
> > [7] Tipping, Michael. "The relevance vector machine." Advances in neural information processing systems 12 (1999).
> >
> > [8] Tipping, Michael E. "Sparse Bayesian learning and the relevance vector machine." Journal of machine learning research 1, no. Jun (2001): 211-244.
> >
> > [9] Krizhevsky, Alex, Ilya Sutskever, and Geoffrey E. Hinton. "Imagenet classification with deep convolutional neural networks." Communications of the ACM 60, no. 6 (2017): 84-90.
> >
> > [10] He, Kaiming, Xiangyu Zhang, Shaoqing Ren, and Jian Sun. "Deep residual learning for image recognition." In Proceedings of the IEEE conference on computer vision and pattern recognition, pp. 770-778. 2016.
> >
> > [11] Hu, Jie, Li Shen, and Gang Sun. "Squeeze-and-excitation networks." In Proceedings of the IEEE conference on computer vision and pattern recognition, pp. 7132-7141. 2018.
> >
> > [12] Szegedy, Christian, Vincent Vanhoucke, Sergey Ioffe, Jon Shlens, and Zbigniew Wojna. "Rethinking the inception architecture for computer vision." In Proceedings of the IEEE conference on computer vision and pattern recognition, pp. 2818-2826. 2016.
> >
> > [13] Ioffe, Sergey, and Christian Szegedy. "Batch normalization: Accelerating deep network training by reducing internal covariate shift." In International conference on machine learning, pp. 448-456. PMLR, 2015.
> >
> > [14] Jacot, Arthur, Franck Gabriel, and Clément Hongler. "Neural tangent kernel: Convergence and generalization in neural networks." Advances in neural information processing systems 31 (2018).

---

> > > ### Author Response · Authors · 2022-11-16
> > > **Response to Reviewer fLYg (3)**
> > >
> > > **Q2. And one thing comes to my mind, can these two attention methods benefit some practical vision problems, such as semantic segmentation? Can the proposed framework help develop some CNN-related attention modules, such as CBAM?**
> > >
> > > **Reply:** Thanks for your suggestion. In Appendix G and H of our revised manuscript, **we have derived the Attention-Conv2D and Attention-Conv1D from the 2D-convolution and 1D-convolution layers in neural networks, respectively**. Both of these new convolutional attention layers outperform the baseline softmax attention as shown in Table 6 and Table 7 in Appendix B.2 of the revised manuscript. On the ImageNet image classification task, our Attention-Conv2D achieves the Top-1 accuracy and Top-5 accuracy of 73.18% and 91.52% respectively while the corresponding results of the baseline softmax attention are 72.23% and 91.13%. On the LRA benchmark, our Attention-Conv1D achieves an average accuracy of 59.01%  vs. 58.66% obtained by the baseline softmax attention. Note that since our new Attention-Conv2D is derived from the 2D-convolution layer, it is intuitively good for image processing tasks such as image classification. Thus, in Table 6, we compare our Attention-Conv2D with the baseline softmax attention on the ImageNet image classification task.
> > >
> > > Also, Remark 7 in Appendix G of our revised manuscript shows that the convolutional projections used in the Convolutional vision Transformer (CvT) [1] can be derived from our primal-dual framework. Extending our framework to explain the Convolutional Block Attention Module (CBAM) [2] that computes the attention using neural network layers or convolutional neural network layers applying on the input feature is an interesting research direction, and we will explore it in future work. We have mentioned CBAM in the Concluding Remarks of our revision.
> > >
> > >
> > > Following your suggestion, we have been applying the Attention-BN and Attention-SH on the semantic segmentation task. We will update our reply with these new results and include them in the next version of our paper.
> > >
> > > **References**
> > >
> > > [1] Wu, Haiping, Bin Xiao, Noel Codella, Mengchen Liu, Xiyang Dai, Lu Yuan, and Lei Zhang. "Cvt: Introducing convolutions to vision transformers." In Proceedings of the IEEE/CVF International Conference on Computer Vision, pp. 22-31. 2021.
> > >
> > > [2] Woo, Sanghyun, Jongchan Park, Joon-Young Lee, and In So Kweon. "Cbam: Convolutional block attention module." In Proceedings of the European conference on computer vision (ECCV), pp. 3-19. 2018.
> > >
> > > -----
> > > We hope we have cleared your concerns about our work. We have also revised our manuscript according to your comments, and we would appreciate it if we can get your further feedback at your earliest convenience.

---

> > > > ### Author Response · Authors · 2022-11-24
> > > > **Response to Reviewer fLYg - Any further questions on our current draft**
> > > >
> > > > We would like to thank you again for your thoughtful reviews and valuable feedback.
> > > >
> > > > We would appreciate it if you could let us know if our responses have addressed your concerns and whether you still have any other questions on the current draft and our rebuttal.
> > > >
> > > > We would be happy to do any follow-up discussion or address any additional comments.

---

> > > > > ### Comment · Reviewer_fLYg · 2022-11-24
> > > > > **Response to authors**
> > > > >
> > > > > Your responses successfully address my questions, and I am looking forward to your future work for extending the proposed theory to explain other attention methods.

---

> > > > > > ### Author Response · Authors · 2022-11-24
> > > > > > **Thanks for your endorsement!**
> > > > > >
> > > > > > Thanks for your response and we appreciate your endorsement.

---

> > > > > > ### Author Response · Authors · 2022-11-25
> > > > > > **Semantic Segmentation Results**
> > > > > >
> > > > > > As promised in our rebuttal above, following your suggestion, we have conducted additional experiments on the ADE20K semantic segmentation task [1] to confirm the advantage of our Attention-BN/SH/BN+SH. The results in Table 1 below show that the Attention-BN/SH/BN+SH improve over the baseline softmax attention in both single-scale and multi-scale mean Intersection over Union (mIOU) while the Attention-SH/BN+SH are more efficient than the baseline.  Here, all models consist of 14 transformer layers with 3 heads per layer, and the model dimension is 192. We follow the same model configurations and training settings as in [2].
> > > > > >
> > > > > > Table 1: The single-scale and multi-scale mean Intersection over Union (mIOU) of the Attention-BN/SH/BN+SH vs. the baseline softmax attention on the ADE20K semantic segmentation task [1].
> > > > > >
> > > > > > | Model       | Single-scale mIOU        | Multi-scale mIOU   |
> > > > > > | :---        |    :----:   |    :----:   |
> > > > > > | Softmax     |   37.72 |    38.82   |
> > > > > > | Attention-BN   |    38.22   |     **39.23**   |
> > > > > > | Attention-SH     |   38.12   |     39.02   |
> > > > > > | Attention-BN+SH   |   **38.32**    |    39.21    |
> > > > > >
> > > > > >
> > > > > > **References**
> > > > > >
> > > > > > [1] Bolei Zhou, Hang Zhao, Xavier Puig, Sanja Fidler, Adela Barriuso, Antonio Torralba. "Scene Parsing Through ADE20K Dataset." CVPR (2017).
> > > > > >
> > > > > > [2] Robin Strudel, Ricardo Garcia, Ivan Laptev, Cordelia Schmid. "Segmenter: Transformer for Semantic Segmentation". ICCV (2021).

---

### Official Review · Reviewer_W46E · 2022-11-01

**Confidence:** 4
**Correctness:** 4
**Technical Novelty And Significance:** 4
**Empirical Novelty And Significance:** 3
**Recommendation:** 8

**Clarity, Quality, Novelty And Reproducibility:**

**Clarity**: The paper is written well and is easy to understand.

**Quality**: The quality of the paper is high. There is a potential for high impact.

**Novelty**: I score the paper high on technical novelty and technical significance. The empirical performance is convincing but currently, only marginally significant.

**Reproducibility**:  Should be reproducible.


**Strength And Weaknesses:**

## Strengths
- The paper makes novel connections between attention mechanisms in DNNs and SVR - support vector regression. It shows that formulating the network layer as a solution to the ridge regression convex optimization problem, the support vector expansion has the form of self-attention.
- It shows that several different attention types can be derived from different design choices in the SVR formulation thus unifying the approach to designing different types of attention.
- It then proposes two new attention mechanisms - one inspired by batch normalization, and another with scaled (multiresolution) attention heads.
- The paper has a potential for high technical impact.
- It evaluates the performance of the new attention mechanisms (and a combined one) on three benchmarks, demonstrating improvement in both accuracy and performance (efficiency). The performance evaluation is convincing.

## Weaknesses
- __(W.1)__ While it is clear that different design choices lead to different kinds of attention, the paper doesn’t provide intuition or meta-level criteria that could be used to make these design choices. It would be great if the authors provide a discussion of these.
- __(W.2.1)__ The paper mentions that the results are averaged over 5 runs. It is recommended that the spread around the mean is also added to the performance tables to allow the assessment of statistical significance of the gain in accuracy.
- __(W.2.2) UEA TMC__: In Table 1, (a) Most results (~50%) are within one percent of each other. Is this statistically significant? Kindly explain using the spread over 5/ 10 runs. (b) For the HeartBeat dataset, the Attention-BN+SH model is not only significantly worse than the Softmax attention baseline but also Attention-BN and Attention-SH which are both better than the baseline. Kindly explain the anomaly.
- __(W.2.3) LRA__: In Table 2,  only results on the Retrieval dataset seem to be significantly better (I’m assuming significant if performance improvement is > 1%). Kindly explain the results using the spread over 5/ 10 runs.
- __(W.2.4) ImageNet__: In Table 1, (a) Results are within one percent of each other. Is this statistically significant? Kindly explain using the spread over 5/ 10 runs. (b) For Top-5 accuracy, the Attention-BN+SH model is not only significantly worse (almost 20% in absolute terms) than the Softmax attention baseline but also Attention-BN and Attention-SH. Kindly explain the anomaly.
- __(W.3) Combining Attentions__: Explain the intuition behind combining Attention-BN+SH with Linear Attention and why it’s expected to perform better.


**Summary Of The Paper:**

The paper provides a primal-dual formulation to derive self-attention as a solution to the support vector regression problem having the primal formulation in the form of a neural network layer. The formulation allows the authors to derive various types of attention: linear, softmax, sparse, and multi-head attention using different design choices/ constraints. Two new attention mechanisms are then proposed in the paper. Performance is evaluated on 3 benchmarks with small (seemingly statistically significant) improvements in accuracy and impressive improvements in efficiency.


**Summary Of The Review:**

The paper self-attention as a solution to the SVR problem which unifies several attention mechanisms in a single optimization framework. The contribution is novel and has a potential for high technical impact. Two new types of attention are proposed using this framework along with combined attention which shows modest improvement in accuracy but with significant efficiency gains. The quality of submission is good. Overall, it is a good contribution to the field.

---

> ### Author Response · Authors · 2022-11-16
> **Response to Reviewer W46E (1)**
>
> Thank you for your thoughtful review and valuable feedback. Below we address your concerns.
>
> -----
>
> **Q1 (W.1). While it is clear that different design choices lead to different kinds of attention, the paper doesn’t provide intuition or meta-level criteria that could be used to make these design choices. It would be great if the authors provide a discussion of these.**
>
> **Reply:** Thanks for your suggestion. Please allow us to explain the intuition that we used to make these design choices. We will first explain our principled approach to designing new attention and then discuss the Attention-BN and Attention-SH as two examples of our principled design strategy for attention. Given our primal-dual framework for attention layers and neural network layers via solving a Support Vector Regression problem, **new attentions can be invented by 1) modifying the support vector regression problem and 2) adjusting the primal neural network layer in our framework.**
>
> 1. **SVR problems have been well-studied in literature and many theoretical results for SVR have been developed to provide a good understanding of this problem** [1,2,3,4]. For example, in [5] and [6], robust SVR and SVM formulations have been proposed with theoretical guarantees that can be used to develop a new class of robust attention via our primal-dual framework.  Furthermore, the relevance vector machine (RVM) [7,8] has been developed as a probabilistic sparse kernel model identical in functional form to the SVM, which again can be employed to develop new probabilistic attention via our primal-dual framework. Thus, **changing the SVR problem is principled and not an ad-hoc approach based on intuition and heuristics**. In our paper, we propose the Attention with Scaled Heads (Attention-SH) as a simple but interesting example of how to develop new attention starting from the SVR problem in which we train multiple support vector regression problems using different amounts of training data to derive a new attention mechanism with multiscale heads. In Attention-SH, each token is allowed to attend to a group of tokens of different sizes at different attention heads.
>
> 2. Neural network layers have been intensively studied, especially over the last 10 years. Since the AlexNet [9] was proposed in 2012, many types of neural network layers have been developed to keep improving the state-of-the-art in a wide range of practical tasks, such as the residual layer [10], the squeeze-and-excitation layer [11], the inception layer [12], the batch normalization layer [13], and the neural tangent kernel layer [14]. **Compared to the attention layers, the neural network layers are better understood, and the experience in developing neural network layers has been accumulated in our community over many years. Our primal-dual framework allows the development of new attention to inherit from this understanding and experience in developing neural network layers.** The Batch Normalized Attention (Attention-BN) proposed in our paper is an example of how new attention can be developed from a well-known neural network layer, i.e. the batch normalization layer.
>
> In Appendix G and H of our revised manuscript, **we have also derived the Attention-Conv2D and Attention-Conv1D from the 2D-convolution and 1D-convolution layers in neural networks, respectively.** Both of these new convolutional attention layers outperform the baseline softmax attention as shown in Table 6 and Table 7 in Appendix B.2 of the revised manuscript. On the ImageNet image classification task, our Attention-Conv2D achieves the Top-1 accuracy and Top-5 accuracy of 73.18% and 91.52% respectively while the corresponding results of the baseline softmax attention are 72.23% and 91.13%. On the LRA benchmark, our Attention-Conv1D achieves an average accuracy of 59.01%  vs. 58.66% obtained by the baseline softmax attention. Note that since our new Attention-Conv2D is derived from the 2D-convolution layer, it is intuitively good for image processing tasks such as image classification. Thus, in Table 6, we compare our Attention-Conv2D with the baseline softmax attention on the ImageNet image classification task.

---

> > ### Author Response · Authors · 2022-11-16
> > **Response to Reviewer W46E (2)**
> >
> > **References**
> >
> > [1] Smola, Alex J., and Bernhard Schölkopf. "A tutorial on support vector regression." Statistics and computing 14, no. 3 (2004): 199-222.
> >
> > [2] Schölkopf, Bernhard, Alexander J. Smola, and Francis Bach. Learning with kernels: support vector machines, regularization, optimization, and beyond. MIT press, 2002.
> >
> > [3] Awad, Mariette, and Rahul Khanna. "Support vector regression." In Efficient learning machines, pp. 67-80. Apress, Berkeley, CA, 2015.
> >
> > [4] Drucker, Harris, Christopher J. Burges, Linda Kaufman, Alex Smola, and Vladimir Vapnik. "Support vector regression machines." Advances in neural information processing systems 9 (1996).
> >
> > [5] Lv, Yuan, and Zhong Gan. "Robust ε-support vector regression." Mathematical Problems in Engineering (2014).
> >
> > [6] Xu, Huan, Constantine Caramanis, and Shie Mannor. "Robustness and Regularization of Support Vector Machines." Journal of machine learning research 10, no. 7 (2009).
> >
> > [7] Tipping, Michael. "The relevance vector machine." Advances in neural information processing systems 12 (1999).
> >
> > [8] Tipping, Michael E. "Sparse Bayesian learning and the relevance vector machine." Journal of machine learning research 1, no. Jun (2001): 211-244.
> >
> > [9] Krizhevsky, Alex, Ilya Sutskever, and Geoffrey E. Hinton. "Imagenet classification with deep convolutional neural networks." Communications of the ACM 60, no. 6 (2017): 84-90.
> >
> > [10] He, Kaiming, Xiangyu Zhang, Shaoqing Ren, and Jian Sun. "Deep residual learning for image recognition." In Proceedings of the IEEE conference on computer vision and pattern recognition, pp. 770-778. 2016.
> >
> > [11] Hu, Jie, Li Shen, and Gang Sun. "Squeeze-and-excitation networks." In Proceedings of the IEEE conference on computer vision and pattern recognition, pp. 7132-7141. 2018.
> >
> > [12] Szegedy, Christian, Vincent Vanhoucke, Sergey Ioffe, Jon Shlens, and Zbigniew Wojna. "Rethinking the inception architecture for computer vision." In Proceedings of the IEEE conference on computer vision and pattern recognition, pp. 2818-2826. 2016.
> >
> > [13] Ioffe, Sergey, and Christian Szegedy. "Batch normalization: Accelerating deep network training by reducing internal covariate shift." In International conference on machine learning, pp. 448-456. PMLR, 2015.
> >
> > [14] Jacot, Arthur, Franck Gabriel, and Clément Hongler. "Neural tangent kernel: Convergence and generalization in neural networks." Advances in neural information processing systems 31 (2018).

---

> > > ### Author Response · Authors · 2022-11-16
> > > **Response to Reviewer W46E (3)**
> > >
> > > ​​**Q2 (W.2.1). The paper mentions that the results are averaged over 5 runs. It is recommended that the spread around the mean is also added to the performance tables to allow the assessment of statistical significance of the gain in accuracy. (W.2.2). UEA TMC: In Table 1, (a) Most results (~50%) are within one percent of each other. Is this statistically significant? Kindly explain using the spread over 5/ 10 runs. (W.2.3). LRA: In Table 2, only results on the Retrieval dataset seem to be significantly better (I’m assuming significant if performance improvement is > 1%). Kindly explain the results using the spread over 5/ 10 runs. (W.2.4). ImageNet: In Table 1, (a) Results are within one percent of each other. Is this statistically significant? Kindly explain using the spread over 5/ 10 runs.**
> > >
> > > **Reply:** Following the reviewer’s suggestion, we have added the spread around the mean, i.e., the standard deviation, to our results in the revised manuscripts.
> > >
> > >  -----
> > > **Q3 (W.2.2) For the HeartBeat dataset, the Attention-BN+SH model is not only significantly worse than the Softmax attention baseline but also Attention-BN and Attention-SH which are both better than the baseline. Kindly explain the anomaly. (W.2.4) For Top-5 accuracy, the Attention-BN+SH model is not only significantly worse (almost 20% in absolute terms) than the Softmax attention baseline but also Attention-BN and Attention-SH. Kindly explain the anomaly.**
> > >
> > > **Reply:** Thanks for pointing out these typos. We have fixed the typo for the HeartBeat and top-5 ImageNet accuracy results in Table 1 and Table 3 of our revised manuscript, respectively. For the HeartBeat benchmark in Table 1, our Attention-BN+SH achieve the test accuracy of 76.26 +/- 1.07 instead of 72.26. For the ImageNet benchmark in Table 3, our Attention-BN+SH achieves a test accuracy of 91.14 +/- 0.12 instead of 72.14.
> > >
> > > -----
> > > **Q4 (W.3). Combining Attentions: Explain the intuition behind combining Attention-BN+SH with Linear Attention and why it’s expected to perform better.**
> > >
> > > **Reply:** The baseline Softmax Attention has the computational and memory complexity of O(N^{2}) where N is the input sequence length. In the meantime, the Linear Attention has the computational and memory complexity of O(N) [1]. Our Attention-BN+SH can be applied to many types of attention to improve their accuracy and efficiency, but not only the Softmax Attention. In particular, in Table 5 in Appendix B.1 of our paper, we show that applying Attention-BN+SH to the Linear Attention helps improve the accuracy of the Linear Attention while still maintaining the O(N) computational and memory complexity of the Linear Attention.
> > >
> > > In addition to the Linear Attention, we have also combined our Attention-BN and Attention-SH with the Sparse Attention [2] and summarized the results in comparison with the Sparse Attention in Table 10 in Appendix B.4 of our revised manuscript. Sparse Attention is another recent work that tries to improve the efficiency of the Softmax Attention via sparsifying the attention matrices. As shown in Table 10, when combined with the Sparse Attention, our Sparse Attention-BN, Sparse Attention-SH, and Sparse Attention-BN+SH outperform the Sparse Attention in terms of accuracy while our Sparse Attention-BN+SH and Attention-SH are more efficient than the baseline Sparse Attention. The efficiency advantages of the Sparse Attention-BN+SH and Sparse Attention-SH over the baseline Sparse Attention grow as the model dimension D and the sequence length N increase, as presented in Figure 3 and Figure 4 in Appendix C of our revised manuscript.
> > >
> > > **References:**
> > >
> > > [1] Katharopoulos, Angelos, Apoorv Vyas, Nikolaos Pappas, and François Fleuret. "Transformers are rnns: Fast autoregressive transformers with linear attention." In International Conference on Machine Learning, pp. 5156-5165. PMLR, 2020.
> > >
> > > [2] Child, Rewon, Scott Gray, Alec Radford, and Ilya Sutskever. "Generating long sequences with sparse transformers." arXiv preprint arXiv:1904.10509 (2019).
> > >
> > > -----
> > > We hope we have cleared your concerns about our work. We have also revised our manuscript according to your comments, and we would appreciate it if we can get your further feedback at your earliest convenience.

---

> > > > ### Author Response · Authors · 2022-11-24
> > > > **Response to Reviewer W46E - Any further questions on our current draft**
> > > >
> > > > We would like to thank you again for your thoughtful reviews and valuable feedback.
> > > >
> > > > We would appreciate it if you could let us know if our responses have addressed your concerns and whether you still have any other questions on the current draft and our rebuttal.
> > > >
> > > > We would be happy to do any follow-up discussion or address any additional comments.

---

> > > > > ### Comment · Reviewer_W46E · 2022-11-28
> > > > > **Acknowledgments**
> > > > >
> > > > > Dear Authors:
> > > > >
> > > > > Thanks a lot for your detailed response. I've factored this into my review. I have no further questions. All the best with the paper decision (and your research in general).
> > > > > Regards

---

> > > > > > ### Author Response · Authors · 2022-12-04
> > > > > > **Thanks for your endorsement!**
> > > > > >
> > > > > > Thanks for your response and we appreciate your endorsement.

---

### Official Review · Reviewer_ipYM · 2022-11-06

**Confidence:** 2
**Correctness:** 4
**Technical Novelty And Significance:** 3
**Empirical Novelty And Significance:** Not applicable
**Recommendation:** 8

**Clarity, Quality, Novelty And Reproducibility:**

The paper is clearly written and the theoretical results are novel to the best of my knowledge.

**Strength And Weaknesses:**

This paper has several strengths:

1. It is very well written: The problem is well motivated, the background on self attention is useful for optimization researchers who are not familiar with transformers, and the notation and theoretical results are clearly described.

2. The problem considered is important: Since self attention plays such an important role in the performance of transformers, new and more principled ways to discover better attention mechanisms will further improve this performance.

3. The theoretical results are novel: I am not aware of the connection between support vector expansions and transformers derived in this paper, and I believe this is a clever observation that is backed up theoretically.

However, it has a few weaknesses:

1. Is the proposed framework really principled? Deriving new attention mechanisms using the proposed framework involves changing the the SVR problem and the primal neural network layer. However, *how* to change the SVR problem and the primal neural network layers still appears to involve intuition and heuristics. In Section 2.3, the paper derives 2 new attention mechanisms, but there is nothing to foretell which of these attention mechanisms is superior to the other (without empirical evaluation). As such, I am not sure whether the proposed framework (in its current form) is really more principled than current practice.

2. The empirical evaluation of the new attention mechanisms is not conclusive: Based on the results in Table 1, it is unclear whether Attention-BN or Attention-BH is to be preferred overall, which undermines the goal of this paper of providing a principled way to design attention mechanisms. The baselines are also fairly sparse (just one attention mechanism) and arbitrary subsets of public datasets were used for evaluation (without justification).

**Summary Of The Paper:**

This paper shows that self-attention mechanisms can be derived by finding the support vector expansion of a support vector regression (SVR) problem. The key contribution of this paper is the framework proposed in Section 2, which links attention mechanisms to support vector regression problems, and provides a different way of deriving attention mechanisms. The paper re-derives softmax, sparse, and multihead attention mechanisms using their proposed framework, and derives two new attention mechanisms: batch normalized and multiresolution head attention.

**Summary Of The Review:**

Overall, I believe this paper proposes a new "interpretation" of self-attention by theoretically linking it to support vector expansion, and provides empirical evidence that new attention mechanisms derived using this link perform well. However, I believe the proposed framework still involves the usage of heuristics and intuition to derive new attention mechanisms. Nevertheless, the framework paves the way towards a more principled way of deriving new self-attention mechanisms in the future.

---

> ### Author Response · Authors · 2022-11-16
> **Response to Reviewer ipYM (1)**
>
> Thank you for your thoughtful review and valuable feedback. Below we address your concerns.
>
> -----
>
> **Q1. Is the proposed framework really principled? Deriving new attention mechanisms using the proposed framework involves changing the SVR problem and the primal neural network layer. However, how to change the SVR problem and the primal neural network layers still appears to involve intuition and heuristics. As such, I am not sure whether the proposed framework (in its current form) is really more principled than current practice.**
>
> **Reply:** Thanks for your comments. We respectfully disagree with the reviewer’s comment that changing the SVR problem involves intuition and heuristics. **SVR problems have been well-studied in literature and many theoretical results for SVR have been developed to provide a good understanding of this problem** [1,2,3,4]. For example, in [5] and [6], robust SVR and SVM formulations have been proposed with theoretical guarantees that can be used to develop a new class of robust attention via our primal-dual framework.  Furthermore, the relevance vector machine (RVM) [7,8] has been developed as a probabilistic sparse kernel model identical in functional form to the SVM, which again can be employed to develop new probabilistic attention via our primal-dual framework. Thus, **changing the SVR problem is principled and not an ad-hoc approach based on intuition and heuristics**. In our paper, we propose the Attention with Scaled Heads (Attention-SH) as a simple but interesting example of how to develop new attention starting from the SVR problem in which we train multiple support vector regression problems using different amounts of training data to derive a new attention mechanism with multiscale heads. In Attention-SH, each token is allowed to attend to a group of tokens of different sizes at different attention heads.
>
> We agree with the reviewer that changing the primal neural network layers involve intuition and heuristics. However, neural network layers have been intensively studied, especially over the last 10 years. Since the AlexNet [9] was proposed in 2012, many types of neural network layers have been developed to keep improving the state-of-the-art in a wide range of practical tasks, such as the residual layer [10], the squeeze-and-excitation layer [11], the inception layer [12], the batch normalization layer [13], and the neural tangent kernel layer [14]. **Compared to the attention layers, the neural network layers are better understood, and the experience in developing neural network layers has been accumulated in our community over many years. Our primal-dual framework allows the development of new attention to inherit from this understanding and experience in developing neural network layers**. The Batch Normalized Attention (Attention-BN) proposed in our paper is an example of how new attention can be developed from a well-known neural network layer, i.e. the batch normalization layer.
>
> In Appendix G and H of our revised manuscript, **we have also derived the Attention-Conv2D and Attention-Conv1D from the 2D-convolution and 1D-convolution layers in neural networks, respectively**. Both of these new convolutional attention layers outperform the baseline softmax attention as shown in Table 6 and Table 7 in Appendix B.2 of the revised manuscript. On the ImageNet image classification task, our Attention-Conv2D achieves the Top-1 accuracy and Top-5 accuracy of 73.18% and 91.52% respectively while the corresponding results of the baseline softmax attention are 72.23% and 91.13%. On the LRA benchmark, our Attention-Conv1D achieves an average accuracy of 59.01%  vs. 58.66% obtained by the baseline softmax attention. Note that since our new Attention-Conv2D is derived from the 2D-convolution layer, it is intuitively good for image processing tasks such as image classification. Thus, in Table 6, we compare our Attention-Conv2D with the baseline softmax attention on the ImageNet image classification task.

---

> > ### Author Response · Authors · 2022-11-16
> > **Response to Reviewer ipYM (2)**
> >
> > **References**
> >
> > [1] Smola, Alex J., and Bernhard Schölkopf. "A tutorial on support vector regression." Statistics and computing 14, no. 3 (2004): 199-222.
> >
> > [2] Schölkopf, Bernhard, Alexander J. Smola, and Francis Bach. Learning with kernels: support vector machines, regularization, optimization, and beyond. MIT press, 2002.
> >
> > [3] Awad, Mariette, and Rahul Khanna. "Support vector regression." In Efficient learning machines, pp. 67-80. Apress, Berkeley, CA, 2015.
> >
> > [4] Drucker, Harris, Christopher J. Burges, Linda Kaufman, Alex Smola, and Vladimir Vapnik. "Support vector regression machines." Advances in neural information processing systems 9 (1996).
> >
> > [5] Lv, Yuan, and Zhong Gan. "Robust ε-support vector regression." Mathematical Problems in Engineering (2014).
> >
> > [6] Xu, Huan, Constantine Caramanis, and Shie Mannor. "Robustness and Regularization of Support Vector Machines." Journal of machine learning research 10, no. 7 (2009).
> >
> > [7] Tipping, Michael. "The relevance vector machine." Advances in neural information processing systems 12 (1999).
> >
> > [8] Tipping, Michael E. "Sparse Bayesian learning and the relevance vector machine." Journal of machine learning research 1, no. Jun (2001): 211-244.
> >
> > [9] Krizhevsky, Alex, Ilya Sutskever, and Geoffrey E. Hinton. "Imagenet classification with deep convolutional neural networks." Communications of the ACM 60, no. 6 (2017): 84-90.
> >
> > [10] He, Kaiming, Xiangyu Zhang, Shaoqing Ren, and Jian Sun. "Deep residual learning for image recognition." In Proceedings of the IEEE conference on computer vision and pattern recognition, pp. 770-778. 2016.
> >
> > [11] Hu, Jie, Li Shen, and Gang Sun. "Squeeze-and-excitation networks." In Proceedings of the IEEE conference on computer vision and pattern recognition, pp. 7132-7141. 2018.
> >
> > [12] Szegedy, Christian, Vincent Vanhoucke, Sergey Ioffe, Jon Shlens, and Zbigniew Wojna. "Rethinking the inception architecture for computer vision." In Proceedings of the IEEE conference on computer vision and pattern recognition, pp. 2818-2826. 2016.
> >
> > [13] Ioffe, Sergey, and Christian Szegedy. "Batch normalization: Accelerating deep network training by reducing internal covariate shift." In International conference on machine learning, pp. 448-456. PMLR, 2015.
> >
> > [14] Jacot, Arthur, Franck Gabriel, and Clément Hongler. "Neural tangent kernel: Convergence and generalization in neural networks." Advances in neural information processing systems 31 (2018).

---

> > > ### Author Response · Authors · 2022-11-16
> > > **Response to Reviewer ipYM (3)**
> > >
> > > **Q2. In Section 2.3, the paper derives 2 new attention mechanisms, but there is nothing to foretell which of these attention mechanisms is superior to the other (without empirical evaluation). The empirical evaluation of the new attention mechanisms is not conclusive: Based on the results in Table 1, it is unclear whether Attention-BN or Attention-BH is to be preferred overall, which undermines the goal of this paper of providing a principled way to design attention mechanisms.**
> > >
> > > **Reply:**  We believe there is a misunderstanding of the advantage of the new attention mechanisms that we propose in the paper. Please allow us to clear this misunderstanding by clarifying that Attention-BN and Attention-SH are complementary to each other. The advantage of Attention-BN is to improve the accuracy of the model (see Tables 1, 2, 3, 5, 8, 9, and 10 in our paper and revised manuscript) while the advantage of Attention-SH is to enhance the efficiency of the model (see Figure 2 and 4 in Appendix C of the revised manuscript). Tables 1, 2, 3, 5, 8, 9, and 10 and Figure 1 and 3 in our paper and revised manuscript show that combining Attention-BN and Attention-SH, i.e., the Attention-BN+SH, improves both the model’s accuracy and efficiency.
> > >
> > > -----
> > > **Q3. The baselines are also fairly sparse (just one attention mechanism) and arbitrary subsets of public datasets were used for evaluation (without justification).**
> > >
> > > **Reply:**  As the reviewer suggests, we have conducted additional experiments on more benchmarks to justify the advantage of our Attention-BN/SH/BN+SH over the baseline softmax attention. These additional benchmarks include additional 15 datasets from the UEA Time Series Classification Archive [1] and a subset of 6 datasets from the UCR Time Series Regression Archive [2]. We observe that our Attention-BN and Attention-SH+BN outperform the baseline softmax transformers significantly on both of these benchmarks, while the attention-SH has comparable performance with the baseline but is more efficient. We summarize our results in Tables 8 and 9 in Appendix B.3 of the revision. In addition, we compared our methods with the softmax baseline on additional 2 tasks in the LRA benchmark and summarized the results in Table 2 in the main text of our revision. The results further confirm the advantage of our proposed attention mechanisms.
> > >
> > > Beyond the softmax attention, in Table 5 in the Appendix B.1 of our paper, we compared our Linear Attention-BN/SH/BN+SH with the Linear Attention. We have also added a comparison with the Sparse Attention [3] in Table 10 in the Appendix B.4 of our revised manuscript. The linear and sparse versions of our Attention-BN/SH/BN+SH outperform the Linear and Sparse Attention in terms of accuracy, respectively. Figure 3 and 4 in Appendix C of the revised manuscript show the efficiency advantage of our Sparse Attention-BN+SH and Sparse Attention-SH over the baseline Sparse Attention.
> > >
> > > **References**
> > >
> > > [1] Bagnall, Anthony, Hoang Anh Dau, Jason Lines, Michael Flynn, James Large, Aaron Bostrom, Paul Southam, and Eamonn Keogh. "The UEA multivariate time series classification archive, 2018." arXiv preprint arXiv:1811.00075 (2018).
> > >
> > > [2] Tan, Chang Wei, Christoph Bergmeir, François Petitjean, and Geoffrey I. Webb. "Time series extrinsic regression." Data Mining and Knowledge Discovery 35, no. 3 (2021): 1032-1060.
> > >
> > > [3] Child, Rewon, Scott Gray, Alec Radford, and Ilya Sutskever. "Generating long sequences with sparse transformers." arXiv preprint arXiv:1904.10509 (2019).
> > >
> > > -----
> > > We hope we have cleared your concerns about our work. We have also revised our manuscript according to your comments, and we would appreciate it if we can get your further feedback at your earliest convenience.

---

> > > > ### Author Response · Authors · 2022-11-24
> > > > **Response to Reviewer ipYM - Any further questions on our current draft**
> > > >
> > > > We would like to thank you again for your thoughtful reviews and valuable feedback.
> > > >
> > > > We would appreciate it if you could let us know if our responses have addressed your concerns and whether you still have any other questions on the current draft and our rebuttal.
> > > >
> > > > We would be happy to do any follow-up discussion or address any additional comments.

---

### Author Response · Authors · 2022-11-16
**General Response (1)**

Dear AC and reviewers,

Thanks for your thoughtful reviews and valuable comments, which have helped us improve the paper significantly. We are encouraged by the endorsements that: 1) The connection between self-attention and neural network layers via a primal-dual formulation of the Support Vector Regression (SVR) problem developed in our paper is novel and interesting (Reviewer ipYM, W46E, flYg, 2btF); 2) Our primal-dual framework is effective (Reviewer ZNRr), inspiring (Reviewer flYg), and has potential for high technical impact (Reviewer ipYM, W46E); 3) The performance evaluation in our paper is convincing (Reviewer W46E), in which a wide range of experimentation across multiple well-known benchmarks, covering various input modalities, and with comprehensive empirical analyses (Reviewer 2btF).  We have updated our submission based on the reviewers' feedback, and we have highlighted our revision in blue.


One of the main concerns from the reviewers is that the motivation of our framework is a bit unclear and if the proposed framework is really principled or not. We address this concern here.


We will first explain our principled approach to designing new attention and then discuss the Attention-BN and Attention-SH as two examples of our principled design strategy for attention. Given our primal-dual framework for attention layers and neural network layers via solving a Support Vector Regression problem, **new attentions can be invented by 1) modifying the support vector regression problem and 2) adjusting the primal neural network layer in our framework**.

1) **SVR problems have been well-studied in literature and many theoretical results for SVR have been developed to provide a good understanding of this problem** [1,2,3,4]. For example, in [5] and [6], robust SVR and SVM formulations have been proposed with theoretical guarantees that can be used to develop a new class of robust attention via our primal-dual framework.  Furthermore, the relevance vector machine (RVM) [7,8] has been developed as a probabilistic sparse kernel model identical in functional form to the SVM, which again can be employed to develop new probabilistic attention via our primal-dual framework. Thus, **changing the SVR problem is principled and not an ad-hoc approach based on intuition and heuristics**. In our paper, we propose the Attention with Scaled Heads (Attention-SH) as a simple but interesting example of how to develop new attention starting from the SVR problem in which we train multiple support vector regression problems using different amounts of training data to derive a new attention mechanism with multiscale heads. In Attention-SH, each token is allowed to attend to a group of tokens of different sizes at different attention heads.

2) Neural network layers have been intensively studied, especially over the last 10 years. Since the AlexNet [9] was proposed in 2012, many types of neural network layers have been developed to keep improving the state-of-the-art in a wide range of practical tasks, such as the residual layer [10], the squeeze-and-excitation layer [11], the inception layer [12], the batch normalization layer [13], and the neural tangent kernel layer [14]. **Compared to the attention layers, the neural network layers are better understood, and the experience in developing neural network layers has been accumulated in our community over many years. Our primal-dual framework allows the development of new attention to inherit from this understanding and experience in developing neural network layers**. The Batch Normalized Attention (Attention-BN) proposed in our paper is an example of how new attention can be developed from a well-known neural network layer, i.e. the batch normalization layer.

---

> ### Author Response · Authors · 2022-11-16
> **General Response (2)**
>
> In addition, Remark 7 in Appendix G of our revised manuscript shows that the convolutional projections proposed in [15] can be derived from our primal-dual framework. In Appendix G and H of our revised manuscript, **we have also derived our new Attention-Conv2D and Attention-Conv1D from the 2D-convolution and 1D-convolution layers in neural networks, respectively**. Both of these new convolutional attention layers outperform the baseline softmax attention as shown in Table 6 and Table 7 in Appendix B.2 of the revised manuscript. On the ImageNet image classification task, our Attention-Conv2D achieves the Top-1 accuracy and Top-5 accuracy of 73.18% and 91.52% respectively while the corresponding results of the baseline softmax attention are 72.23% and 91.13%. On the LRA benchmark, our Attention-Conv1D achieves an average accuracy of 59.01%  vs. 58.66% obtained by the baseline softmax attention. Note that since our new Attention-Conv2D is derived from the 2D-convolution layer, it is intuitively good for image processing tasks such as image classification. Thus, in Table 6, we compare our Attention-Conv2D with the baseline softmax attention on the ImageNet image classification task. Compared to the attention with convolutional projections in [15], our new Attention-Conv2D needs fewer parameters with competitive performance since we apply the same depth-wise 2D-convolution with identical kernel channels on the queries, the keys, and the values.
>
> **References**
>
> [1] Smola, Alex J., and Bernhard Schölkopf. "A tutorial on support vector regression." Statistics and computing 14, no. 3 (2004): 199-222.
>
> [2] Schölkopf, Bernhard, Alexander J. Smola, and Francis Bach. Learning with kernels: support vector machines, regularization, optimization, and beyond. MIT press, 2002.
>
> [3] Awad, Mariette, and Rahul Khanna. "Support vector regression." In Efficient learning machines, pp. 67-80. Apress, Berkeley, CA, 2015.
>
> [4] Drucker, Harris, Christopher J. Burges, Linda Kaufman, Alex Smola, and Vladimir Vapnik. "Support vector regression machines." Advances in neural information processing systems 9 (1996).
>
> [5] Lv, Yuan, and Zhong Gan. "Robust ε-support vector regression." Mathematical Problems in Engineering (2014).
>
> [6] Xu, Huan, Constantine Caramanis, and Shie Mannor. "Robustness and Regularization of Support Vector Machines." Journal of machine learning research 10, no. 7 (2009).
>
> [7] Tipping, Michael. "The relevance vector machine." Advances in neural information processing systems 12 (1999).
>
> [8] Tipping, Michael E. "Sparse Bayesian learning and the relevance vector machine." Journal of machine learning research 1, no. Jun (2001): 211-244.
>
> [9] Krizhevsky, Alex, Ilya Sutskever, and Geoffrey E. Hinton. "Imagenet classification with deep convolutional neural networks." Communications of the ACM 60, no. 6 (2017): 84-90.
>
> [10] He, Kaiming, Xiangyu Zhang, Shaoqing Ren, and Jian Sun. "Deep residual learning for image recognition." In Proceedings of the IEEE conference on computer vision and pattern recognition, pp. 770-778. 2016.
>
> [11] Hu, Jie, Li Shen, and Gang Sun. "Squeeze-and-excitation networks." In Proceedings of the IEEE conference on computer vision and pattern recognition, pp. 7132-7141. 2018.
>
> [12] Szegedy, Christian, Vincent Vanhoucke, Sergey Ioffe, Jon Shlens, and Zbigniew Wojna. "Rethinking the inception architecture for computer vision." In Proceedings of the IEEE conference on computer vision and pattern recognition, pp. 2818-2826. 2016.
>
> [13] Ioffe, Sergey, and Christian Szegedy. "Batch normalization: Accelerating deep network training by reducing internal covariate shift." In International conference on machine learning, pp. 448-456. PMLR, 2015.
>
> [14] Jacot, Arthur, Franck Gabriel, and Clément Hongler. "Neural tangent kernel: Convergence and generalization in neural networks." Advances in neural information processing systems 31 (2018).
>
> [15] Wu, Haiping, Bin Xiao, Noel Codella, Mengchen Liu, Xiyang Dai, Lu Yuan, and Lei Zhang. "Cvt: Introducing convolutions to vision transformers." In Proceedings of the IEEE/CVF International Conference on Computer Vision, pp. 22-31. 2021.
>
> -----
>
> We are glad to answer any further questions you have on our submission.

---

### Author Response · Authors · 2022-11-16
**Summary of Revision**

Incorporating the comments and suggestions from all reviewers, besides fixing typos and notations, we have made the following main changes in the revised paper.


1. We have conducted additional 15 datasets from the UEA Time Series Classification Archive [1] and a subset of 6 datasets from the UCR Time Series Regression Archive [2]. We summarize our results in Table 8 and 9 in Appendix B.3 of the revision.


2. We compared our methods with the softmax baseline on additional 2 tasks, Image and Pathfinder in the LRA benchmark and summarized the results in Table 2 in the main text of our revision.


3. We have added a comparison between our methods and the Sparse Attention [3] in Table 10 in the Appendix B.4 of our revised manuscript.  We also added Figure 3 and 4 in Appendix C of the revised manuscript to show the efficiency advantage of our Sparse Attention-BN+SH and Sparse Attention-SH over the baseline Sparse Attention.

4. We have added Figure 2 in Appendix C of the revised manuscript to show the efficiency advantage of our Attention-SH over the baseline Softmax Attention.


5. We have added the spread around the mean, i.e., the standard deviation, to our results in the revised manuscripts.

6. In Appendix G and H of our revised manuscript, we have derived the Attention-Conv2D and Attention-Conv1D from the 2D-convolution and 1D-convolution layers in neural networks, respectively. Both of these new convolutional attention layers outperform the baseline softmax attention as shown in Table 6 and Table 7 in Appendix B.2 of the revised manuscript. Remark 7 in Appendix G of our revised manuscript shows that the convolutional projections used in the Convolutional vision Transformer (CvT) [3] can be derived from our primal-dual framework.

7. We have provided values for the mean weighting factor $\beta$ and the downsampling scales used in our experiments in Appendix J of our revised manuscript.

8. We have conducted experiments where we set $\beta$ to be a trainable parameter and report the results in Table 11 in Appendix B.5 of our revised manuscript.

9. We have included the derivation for Attention-BN+SH in Appendix I of our revised manuscript.

**References**

[1] Bagnall, Anthony, Hoang Anh Dau, Jason Lines, Michael Flynn, James Large, Aaron Bostrom, Paul Southam, and Eamonn Keogh. "The UEA multivariate time series classification archive, 2018." arXiv preprint arXiv:1811.00075 (2018).

[2] Tan, Chang Wei, Christoph Bergmeir, François Petitjean, and Geoffrey I. Webb. "Time series extrinsic regression." Data Mining and Knowledge Discovery 35, no. 3 (2021): 1032-1060.

[3] Child, Rewon, Scott Gray, Alec Radford, and Ilya Sutskever. "Generating long sequences with sparse transformers." arXiv preprint arXiv:1904.10509 (2019).

---

### Author Response · Authors · 2022-11-17
**Any Questions from the Reviewers before the Deadline to Update Our Draft?**

Dear reviewers,

We would like to thank all reviewers again for your thoughtful reviews and valuable feedback. We have updated our manuscript and added new replies to your comments and questions with our latest experimental results. We have summarized the changes we made in the manuscript in the Summary of Revision below.

We would appreciate it if you could let us know if there are additional questions or concerns about our revision and rebuttal.

We would be happy to do any follow-up discussion or address any additional comments.

Best regards,

Authors

---

### Author Response · Authors · 2022-11-25
**New Empirical Results on the ADE20K Semantic Segmentation Task**

Dear reviewers,

We would like to thank all reviewers again for your thoughtful reviews and valuable feedback.

We have conducted additional experiments on the ADE20K semantic segmentation task [1] to further confirm the advantage of our Attention-BN/SH/BN+SH. The results in Table 1 below show that the Attention-BN/SH/BN+SH improve over the baseline softmax attention in both single-scale and multi-scale mean Intersection over Union (mIOU) while the Attention-SH/BN+SH are more efficient than the baseline.  Here, all models consist of 14 transformer layers with 3 heads per layer, and the model dimension is 192. We follow the same model configurations and training settings as in [2].

Table 1: The single-scale and multi-scale mean Intersection over Union (mIOU) of the Attention-BN/SH/BN+SH vs. the baseline softmax attention on the ADE20K semantic segmentation task [1].

| Model       | Single-scale mIOU        | Multi-scale mIOU   |
| :---        |    :----:   |    :----:   |
| Softmax     |   37.72 |    38.82   |
| Attention-BN   |    38.22   |     **39.23**   |
| Attention-SH     |   38.12   |     39.02   |
| Attention-BN+SH   |   **38.32**    |    39.21    |


**References**

[1] Bolei Zhou, Hang Zhao, Xavier Puig, Sanja Fidler, Adela Barriuso, Antonio Torralba. "Scene Parsing Through ADE20K Dataset." CVPR (2017).

[2] Robin Strudel, Ricardo Garcia, Ivan Laptev, Cordelia Schmid. "Segmenter: Transformer for Semantic Segmentation". ICCV (2021).

---

### Decision · Program_Chairs · 2023-01-20

**Decision:**

Accept: notable-top-25%

**Justification For Why Not Higher Score:**

Though the authors provide an interesting framework, its significance is unclear. In particular,  the two newly proposed attention mechanisms actually do not seem novel compared to existing work. Also its performance advantage over the existing literature is not significant enough.

**Justification For Why Not Lower Score:**

The authors provided detailed mathematical proof of the relation between the attention mechanism and the support vector regression. This is different from the intuitive design of most previous attention methods, and the approach of this paper may somehow inspire the community how to rethink the attention mechanism.

**Metareview: Summary, Strengths And Weaknesses:**

It is an interesting idea to explore the connection between self-attention mechanisms and the support vector expansion of a support vector regression (SVR) problem. Two new attention mechanisms: batch normalized and multiresolution head attention, have been developed through the authors' new softmax, sparse, and multihead attention mechanisms within their proposed framework. The authors have provided convincing experimental results.

**Note From Pc:**

if the above contains the word "oral" or "spotlight" please see: "oral" presentation means -> notable-top-5% and "spotlight" means -> notable-top-25%. As stated in our emails, we are disassociating presentation type from AC recommendations